# Interneuronal mechanisms of hippocampal theta oscillations in a full-scale model of the rodent CA1 circuit

**Marianne J Bezaire[1]\*[†], Ivan Raikov[1,2], Kelly Burk[1], Dhrumil Vyas[1], Ivan Soltesz[2]**

[1]Department of Anatomy and Neurobiology, University of California, Irvine, Irvine, United States; [2]Department of Neurosurgery, Stanford University, Stanford, United States

**Abstract** The hippocampal theta rhythm plays important roles in information processing; however, the mechanisms of its generation are not well understood. We developed a data-driven, supercomputer-based, full-scale (1:1) model of the rodent CA1 area and studied its interneurons during theta oscillations. Theta rhythm with phase-locked gamma oscillations and phase-preferential discharges of distinct interneuronal types spontaneously emerged from the isolated CA1 circuit without rhythmic inputs. Perturbation experiments identified parvalbumin-expressing interneurons and neurogliaform cells, as well as interneuronal diversity itself, as important factors in theta generation. These simulations reveal new insights into the spatiotemporal organization of the CA1 circuit during theta oscillations.

**\*For correspondence:** marianne. bezaire@gmail.com

**Present address:** [†]Department of Psychological and Brain Sciences, Boston University, Boston, United States

**Competing interests:** The authors declare that no competing interests exist.

## Introduction

The hippocampal CA1 area supports diverse cognitive tasks including learning, memory, and spatial processing (*Squire, 1992*; *Remondes and Schuman, 2004*; *Manns et al., 2007*; *Moser et al., 2008*). These cognitive tasks are thought to require coordination of neuronal activity provided by physiological network oscillations, including the theta rhythm (*Buzsáki, 2002*; *Buzsáki and Moser, 2013*). In rodents, hippocampal theta is a 5–10 Hz oscillation in the local field potential (LFP) and neuronal firing probabilities (*Soltesz and Deschênes, 1993*; *Lee et al., 1994*; *Ylinen et al., 1995*; *Klausberger and Somogyi, 2008*; *Varga et al., 2012*, *2014*), occurring during locomotion and in REM sleep (*Buzsáki, 2002*). Though several major afferents provide theta-frequency rhythmic input to the CA1 in vivo (*Soltesz and Deschênes, 1993*; *Buzsáki, 2002*; *Fuhrmann et al., 2015*), recent reports indicate the presence of spontaneous theta-frequency LFP oscillations even in the isolated whole CA1 preparation in vitro (*Goutagny et al., 2009*; *Amilhon et al., 2015*). Therefore, the latter studies suggest an intrinsic ability of the CA1 circuit to generate some form of theta waves even without rhythmic external inputs. However, the intra-CA1 mechanisms that may contribute to the generation of the theta rhythm are not well understood (*Colgin, 2013*, *2016*).

Here we investigated the ability of the CA1 to generate intrinsic theta oscillations using a uniquely biological data-driven, full-scale computer model of the isolated CA1 network. Recent advances in supercomputing power and high-quality synaptic connectivity data present the intriguing opportunity to develop full-scale models where every biological synapse and neuron is explicitly represented. In principle, such full-scale models of mammalian circuits comprising hundreds of thousands of neurons of distinct types advantageously avoid the connectivity scaling tradeoff that besets reduced-scale models: smaller models of large networks with realistic single cell electrophysiological properties (e.g., input resistance and resting membrane potential) remain silent unless synaptic strengths or numbers are arbitrarily increased beyond the biologically relevant levels to compensate

for fewer inputs to their model cells (e.g., [*Dyhrfjeld-Johnsen et al., 2007*; *Sterratt et al., 2011*]). Biological relevance may also increase as other network components are modeled in greater detail. However, full-scale models require considerable computational resources. Further, such detailed models have a large parameter space which risks being sub-optimally constrained by neurobiological properties that are only partially quantified (*Sejnowski et al., 1988*). Because the CA1 area is one of the most extensively studied brain regions, there are abundant anatomical and electrophysiological data about its organization, making it a logical choice for the development of a full-scale model. The CA1 area is also worth modeling at full-scale because of the diverse cognitive tasks it supports. These tasks likely require the simultaneous processing of thousands of incoming and outgoing signals, and full-scale network models, at least in principle, have the potential to match this in vivo processing capacity.

In this paper, we describe the development of a full-scale CA1 computational network model of unprecedented biological detail and its application to gain insights into the roles and temporal organization of CA1 interneurons during theta rhythm. The simulated full-scale CA1 circuit was able to spontaneously generate theta waves as well as phase-locked gamma oscillations. Furthermore, distinct interneuron types discharged at particular phases of theta, demonstrating that phase-preferential firing (*Klausberger et al., 2003*, *2004*, *2005*; *Ferraguti et al., 2005*; *Jinno et al., 2007*; *Fuentealba et al., 2008*; *Klausberger and Somogyi, 2008*; *Varga et al., 2012*; *Lapray et al., 2012*; *Katona et al., 2014*; *Varga et al., 2014*) originates in part within the CA1 network. Perturbation experiments revealed that parvalbumin-expressing (PV+) interneurons, neurogliaform cells, connections between CA1 pyramidal cells, and interneuronal diversity were important for theta generation. These results provide new mechanistic insights into the emergence of the theta rhythm from within the CA1 circuitry and the role of interneurons in theta oscillations.

## Results

### Development of a data-driven, full-scale model of the isolated CA1

Details of the full-scale model are described in the Methods, and the most important features are illustrated in *Figures 1* and *2* and summarized here. Briefly, CA1 model cells were evenly distributed within their respective layers in a 3-dimensional prism with realistic dimensions for the rodent hippocampal CA1 region (*Figure 1A and B*). The model network contained 338,740 cells (similar to the biological CA1 in rats, including 311,500 pyramidal cells and 27,240 interneurons) (*Figure 1D–E* and *Figure 1—figure supplement 1*). In addition, the network also incorporated 454,700 artificial stimulating cells (spiking units with random, Poisson-distributed interspike intervals) to simulate afferents to CA1; the cell type-specific distribution, dendritic position, amplitude and kinetics of the excitatory input synapses were all experimentally constrained by afferent CA3 and entorhinal cortical data. Cell type-specific connectivity data, including cell numbers (*Figure 1D*) and convergence and divergence values (*Figure 1E*; *Figure 1—figure supplement 1* and *Table 1*) were taken without alteration from our previously published, in-depth, quantitative assessment of the CA1 circuit (*Bezaire and Soltesz, 2013*). Anatomical constraints of the connectivity were implemented in the model by accounting for the distribution of the axonal boutons as a function of longitudinal and transverse distance from the presynaptic cell soma (*Figure 1—figure supplement 2*). The afferent divergence and convergence onto the cells were also anatomically patterned, maintaining the topographical arrangement seen experimentally (*Hongo et al., 2015*), for a total of 5.19 billion synaptic connections in the model network. In addition, the remaining parameters that could not be constrained by experimental data were documented, with the assumptions used to arrive at them explicitly listed in Table 2 of *Bezaire and Soltesz (2013)* and additional parameter calculations described in the Appendix of the present paper, section 'Inhibitory connectivity'. To highlight the many constraints applied in the current work and address the unconstrained model parameters, we characterized all model components (constrained and unconstrained) in experimental terms, comparing with experimental data where possible (*Figure 2*; Appendix). For a four second simulation, the full-scale model required 3–4 terabytes (TB) of RAM and four hours of execution time on a supercomputer using ~3000 processors (or up to 12 hr for simulations calculating a high-accuracy local field potential (LFP) analog). Additional details and data about model performance are available in *Table 2* and *Bezaire et al. (2016a)*.

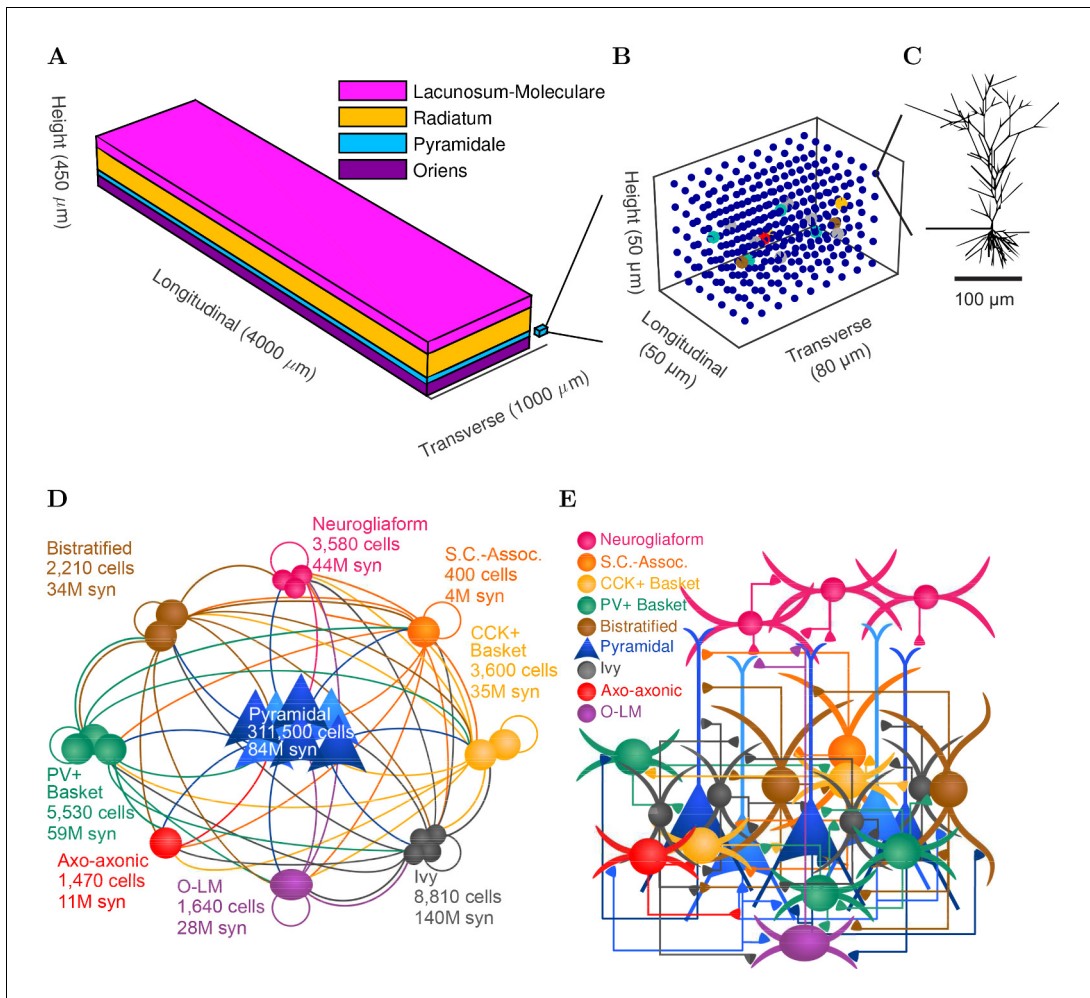

**Figure 1.** CA1 network connectivity. (**A**) The model network is arranged in a layered prism with the lengths of each dimension similar to the actual dimensions of the CA1 region and its layers. (**B**) The model cell somata within a small chunk of stratum pyramidale (as depicted in **A**) are plotted to show the regular distribution of model cells throughout the layer in which they are found. (**C**) Each pyramidal cell in the network has detailed morphology with realistic incoming synapse placement along the dendrites and soma. (**D,E**) Diagrams illustrate connectivity between types of cells. (**D**) The network includes one principal cell type (pyramidal cells) and eight interneuron types. Cell types that may connect are linked by a line colored according to the presynaptic cell type. Most cell types can connect to most other cell types. Total number of cells of each type are displayed, as are the number of local output synapses (boutons) from all cells of each type. (**E**) The number, position, and cell types of each connection are biologically constrained, as are the numbers and positions of the cells. See *Figure 1—figure supplement 1*) for details about the convergence onto each cell type. Also see *Table 1* and *Figure 1—figure supplement 2* for information about the cell-type combinations of the 5 billion connections and the axonal distributions followed by each cell type, as well as detailed connectivity results at http://doi.org/10.6080/K05H7D60.

The following figure supplements are available for figure 1:

**Figure supplement 1.** Quantitative network connectivity.

**Figure supplement 2.** Anatomically constrained connectivity.

An important set of constraints was the electrophysiology and other properties of individual cells and synapses (*Figure 2*; *Figure 2—source data 3–27*; *Tables 3* and *4*) that were based on experimental data (*Lee et al., 2016*; *Quattrocolo and Maccaferri, 2016*). Briefly, our pyramidal cell model

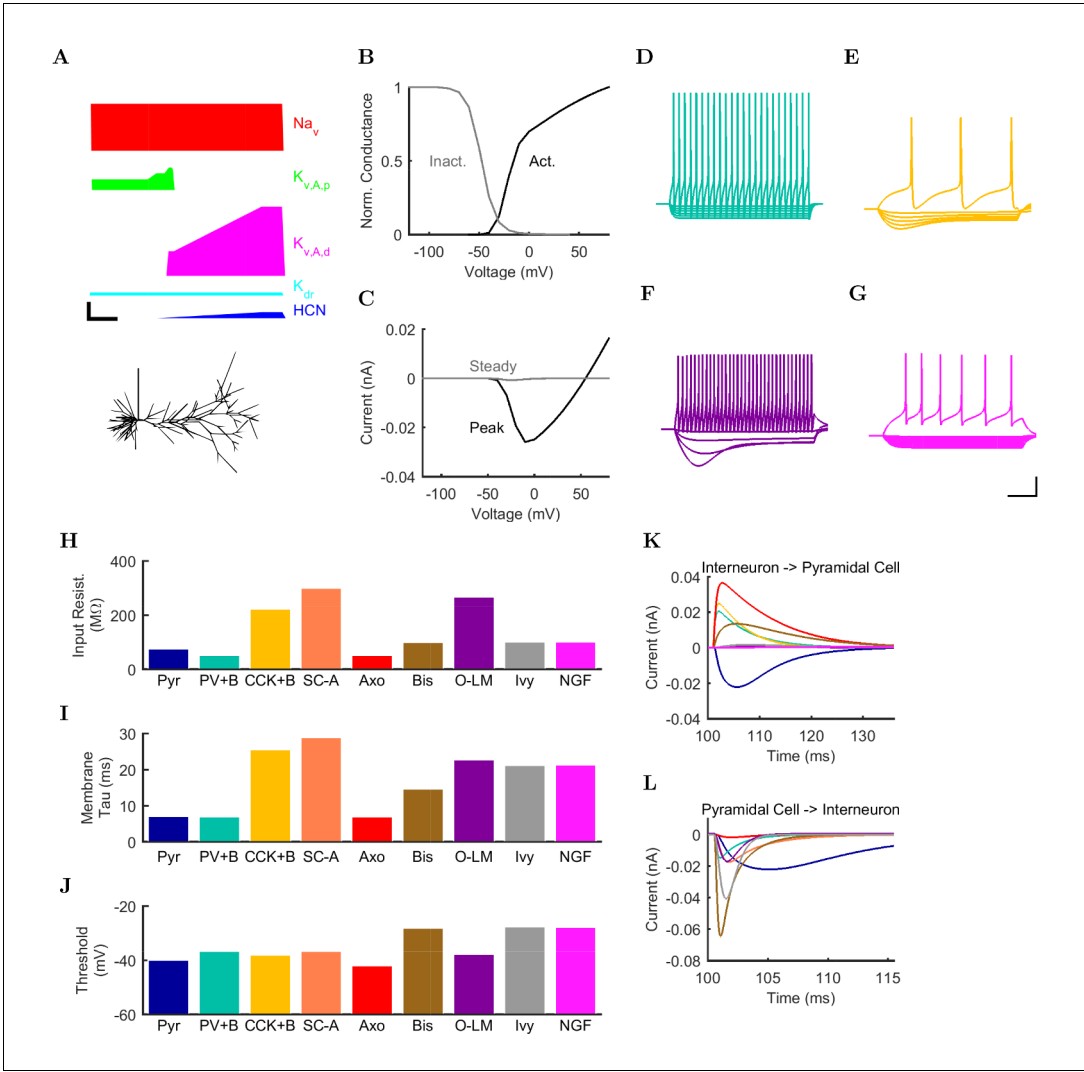

**Figure 2.** Electrophysiology of the model network components. (**A**) Ion channel densities vary as a function of location (top) in the morphologically detailed pyramidal cell model (bottom; adapted from *Poolos et al., 2002*). Scale bar: 100 $\mu m$ and 0.01 $\mu F/cm^2$. (**B–C**) The sodium channel found in the pyramidal cell soma is characterized in terms of (**B**) the activation/inactivation curves and (**C**) the current-voltage relation at peak (transient) current and steady state. (**D–G**) Current sweeps are shown for four model cell types: (**D**) PV+ basket cell, (**E**) CCK+ basket cell, (**F**) O-LM cell, and (**G**) neurogliaform cell. Scale bar: 100 ms and 20 mV. (**H–J**) Electrophysiological properties for each cell type, including (**H**) input resistance, (**I**) membrane time constant, and (**J**) action potential threshold. (**K–L**) Pyramidal cell synaptic connections are characterized as post-synaptic currents with the postsynaptic cell voltage clamped at −50 mV; (**K**) synapses made onto the pyramidal cell from all other cell types and (**L**) synapses made by the pyramidal cell onto all network cell types. Cells represented by same colors as in *Figure 1*. Source Data available for electrophysiological characterizations shown here. Additional details available in the Methods, *Table 3*, and the Appendix.

The following source data is available for figure 2:

**Source data 1.** Model sodium channel activation.
**Source data 2.** Model sodium channel inactivation.
**Source data 3.** Model axo-axonic cell current injection sweep.
**Source data 4.** Model bistratified cell current injection sweep.
*Figure 2 continued on next page*

*Figure 2 continued*

**Source data 5.** Model CCK+ basket cell current injection sweep.

**Source data 6.** Model ivy cell current injection sweep.

**Source data 7.** Model neurogliaform cell current injection sweep.

**Source data 8.** Model O-LM cell current injection sweep.

**Source data 9.** Model PV+ basket cell current injection sweep.

**Source data 10.** Model pyramidal cell current injection sweep.

**Source data 11.** Model Schaffer Collateral-Associated cell current injection sweep.

**Source data 12.** Model paired recording of an Axo-axonic cell to Pyramidal cell connection.

**Source data 13.** Model paired recording of a Bistratified cell to Pyramidal cell connection.

**Source data 14.** Model paired recording of a CA3 cell to Pyramidal cell connection.

**Source data 15.** Model paired recording of a CCK+ basket cell to Pyramidal cell connection.

**Source data 16.** Model paired recording of an ECIII cell to Pyramidal cell connection.

**Source data 17.** Model paired recording of an Ivy cell to Pyramidal cell connection.

**Source data 18.** Model paired recording of a Pyramidal cell to Pyramidal cell connection.

**Source data 19.** Model paired recording of a Neurogliaform cell to Pyramidal cell connection.

**Source data 20.** Model paired recording of an O-LM cell to Pyramidal cell connection.

**Source data 21.** Model paired recording of a PV+ basket cell to Pyramidal cell connection.

**Source data 22.** Model paired recording of a Pyramidal cell to Axo-axonic cell connection.

**Source data 23.** Model paired recording of a Pyramidal cell to Bistratified cell connection.

**Source data 24.** Model paired recording of a Pyramidal cell to Ivy cell connection.

**Source data 25.** Model paired recording of a Pyramidal cell to O-LM cell connection.

**Source data 26.** Model paired recording of a Pyramidal cell to PV+ basket cell connection.

**Source data 27.** Model paired recording of a Pyramidal cell to Schaffer Collateral-Associated cell connection.

(*Poolos et al., 2002*) contained 200 compartments in a realistic morphology and six fully characterized ion channel types with kinetics and densities based on anatomical location within the cell (*Figure 2A–C*; *Figure 2—source data 1–2*). We included eight model interneuron types (*Klausberger and Somogyi, 2008*; *Soltesz, 2006*; *Armstrong and Soltesz, 2012*): PV+ basket cells (these fast-spiking cells synapse on the somata and proximal dendrites of CA1 pyramidal cells), cholecystokinin+ (CCK+) basket cells (these regular-spiking cells also innervate the somata and proximal dendrites, but have properties and functions distinct from the PV+ basket cells), bistratified cells (these PV+ and somatostatin+ (SOM+) fast-spiking cells innervate the basal and apical dendritic trees), axo-axonic cells (these PV+ fast-spiking cells synapse only on the axon initial segments of pyramidal cells and are also known as chandelier cells), Schaffer Collateral-Associated (SC-A) cells

**Table 1.** Number of synapses between each cell type. Connections between cells generally comprise 1–10 synapses each. Presynaptic cells are listed down the first column (corresponding to each row) and postsynaptic cells are listed along the first row (corresponding to each column).

| Pre/Post | Axo | Bis | CCK+B | Ivy | NGF | O-LM | Pyr | PV+B | SC-A |
|---|---|---|---|---|---|---|---|---|---|
| Axo | 0.00e + 00 | 0.00e + 00 | 0.00e + 00 | 0.00e + 00 | 0.00e + 00 | 0.00e + 00 | 1.12e + 07 | 0.00e + 00 | 0.00e + 00 |
| Bis | 2.35e + 05 | 3.54e + 05 | 5.76e + 05 | 2.64e + 05 | 0.00e + 00 | 6.40e + 05 | 3.12e + 07 | 8.85e + 05 | 6.80e + 04 |
| CCK+B | 1.41e + 05 | 2.12e + 05 | 9.79e + 05 | 5.64e + 05 | 0.00e + 00 | 2.62e + 05 | 3.24e + 07 | 5.31e + 05 | 8.32e + 04 |
| Ivy | 3.53e + 05 | 5.30e + 05 | 3.42e + 06 | 2.11e + 06 | 1.00e + 06 | 2.23e + 06 | 1.28e + 08 | 1.33e + 06 | 4.08e + 05 |
| NGF | 0.00e + 00 | 0.00e + 00 | 0.00e + 00 | 0.00e + 00 | 6.09e + 05 | 0.00e + 00 | 4.36e + 07 | 0.00e + 00 | 0.00e + 00 |
| O-LM | 1.18e + 05 | 1.77e + 05 | 1.44e + 06 | 0.00e + 00 | 4.65e + 05 | 9.84e + 04 | 2.49e + 07 | 4.42e + 05 | 1.60e + 05 |
| Pyr | 7.19e + 05 | 2.43e + 06 | 0.00e + 00 | 2.38e + 05 | 0.00e + 00 | 1.17e + 07 | 6.14e + 07 | 7.03e + 06 | 1.26e + 05 |
| PV+B | 5.73e + 04 | 8.62e + 04 | 1.37e + 05 | 7.05e + 04 | 0.00e + 00 | 0.00e + 00 | 5.83e + 07 | 2.16e + 05 | 9.60e + 03 |
| SC-A | 8.82e + 03 | 1.33e + 04 | 1.30e + 05 | 1.06e + 05 | 0.00e + 00 | 1.97e + 04 | 3.74e + 06 | 3.32e + 04 | 1.44e + 04 |
| CA3 | 1.23e + 07 | 2.56e + 07 | 1.44e + 07 | 3.39e + 07 | 0.00e + 00 | 0.00e + 00 | 3.73e + 09 | 6.69e + 07 | 1.55e + 06 |
| ECIII | 1.43e + 06 | 1.91e + 06 | 4.02e + 06 | 0.00e + 00 | 3.75e + 06 | 0.00e + 00 | 8.09e + 08 | 0.00e + 00 | 4.58e + 05 |

(these CCK+, regular-spiking cells innervate dendrites in the stratum radiatum), oriens-lacunosum-moleculare (O-LM) cells (these SOM+ cells project to the distal dendrites in the stratum lacunosum-moleculare though their somata are located in the stratum oriens), neurogliaform cells (these cells have relatively small dendrites and a dense axonal cloud, and they innervate distal dendrites in the stratum lacunosum-moleculare), and ivy cells (these cells are similar to neurogliaform cells, but innervate proximal dendrites) (*Figure 2D–E*). Some interneurons in the model, as in the biological network, also innervated other interneurons (*Table 1*). For greater detail of model connectivity, including convergence per single cell, synaptic amplitude, and other factors, see the Appendix. These cell types collectively comprise the majority (~70%) of known CA1 interneurons (*Bezaire and Soltesz, 2013*). The remaining 30% of the interneurons were not included in the model due to paucity of quantitative data (*Bezaire and Soltesz, 2013*). We differentiated the interneurons by their electrophysiological profiles, connectivity patterns, synaptic properties, and anatomical abundance (*Gulyás et al., 1991*; *Hájos and Mody, 1997*; *Maccaferri et al., 2000*; *Megías et al., 2001*; *Lee et al., 2010*; *Krook-Magnuson et al., 2011*; *Bezaire and Soltesz, 2013*; *Lee et al., 2014*). The synaptic connections were implemented using double exponential mechanisms to better fit experimental data on rise and decay time constants. We used experimental data to constrain the synaptic kinetics, amplitudes, and locations on the postsynaptic cell (*Figure 1E*, *2K and L*). We implemented the model in parallel NEURON (*Carnevale and Hines, 2005*) and executed the simulations on

**Table 2.** Simulation time, exchange time, and load balance for simulations executed on various supercomputers and numbers of processors.

| Supercomputer | # Processors | Sim time (s) | Exchange time (s) | Load balance |
|---|---|---|---|---|
| Comet | 1680 | 2610.28 | 1.05 | 0.999 |
| Comet | 1704 | 2566.76 | 0.65 | 0.999 |
| Comet | 1728 | 2601.22 | 0.86 | 0.999 |
| Comet via NSG | 1728 | 2060.88 | 0.83 | 0.999 |
| Stampede via NSG | 2048 | 2471.64 | 1.71 | 1.000 |
| Stampede | 2048 | 2578.32 | 0.29 | 1.000 |
| Stampede | 2528 | 2189.56 | 1.78 | 0.999 |
| Stampede | 3008 | 1844.22 | 0.91 | 0.999 |
| Stampede | 3488 | 1641.91 | 0.86 | 0.999 |

**Table 3.** Electrophysiological characteristics of each model cell type. For more information about model electrophysiology, see the Appendix.

| Condition | Pyr | PV+B | CCK+B | SC-A | Axo | Bis | O-LM | Ivy | NGF |
|---|---|---|---|---|---|---|---|---|---|
| Resting Membrane Potential (mV) | −63.0 | −65.0 | −70.6 | −70.5 | −65.0 | −67.0 | −71.5 | −60.0 | −60.0 |
| Input Resistance (MΩ) | 62.2 | 52.0 | 211.0 | 272.4 | 52.0 | 98.7 | 343.8 | 100.0 | 100.0 |
| Membrane Tau (ms) | 4.8 | 6.9 | 22.6 | 24.4 | 7.0 | 14.7 | 22.4 | 21.1 | 21.1 |
| Rheobase (pA) | 250.0 | 300.0 | 60.0 | 40.0 | 200.0 | 350.0 | 50.0 | 160.0 | 170.0 |
| Threshold (mV) | 52.0 | −36.6 | −40.6 | −43.1 | −41.6 | −28.1 | 100.2 | −27.6 | −27.7 |
| Delay to 1st Spike (ms) | 12.4 | 74.6 | 166.6 | 127.7 | 43.5 | 28.4 | 8.9 | 173.3 | 119.0 |
| Half-Width (ms) | 80.7 | 0.9 | 1.9 | 1.6 | 0.6 | 0.5 | 112.9 | 0.6 | 0.6 |

several supercomputers. All model results, characterizations, and experimental comparisons are publicly available.

## Emergence of spontaneous theta and gamma oscillations in the full-scale model in the absence of rhythmic external inputs

First, we examined whether the well-constrained, biologically detailed, full-scale CA1 model could oscillate spontaneously within the physiological range. Based on reports of spontaneous theta-frequency LFP oscillations in the isolated CA1 preparation (*Goutagny et al., 2009*), we expected a sufficiently constrained CA1 model to generate spontaneous theta rhythm when given tonic, arrhythmic excitation. We varied the magnitude of arrhythmic, tonic excitation to the network (by systematically changing the mean spiking frequency of the artificial stimulating cells, see above) and identified excitation levels where the network developed a stable, spontaneous theta rhythm (5–10 Hz; *Figure 3 and 4*; *Figure 3—source data 1–3* and *Figure 4—source data 1–2*). The pyramidal cell spikes (*Figures 3C and D*) exhibited peak power around the theta frequency of 7.8 Hz (*Figure 4* and *Table 7*). Importantly, every measure of network activity showed theta oscillations, including the somatic intracellular membrane potential from individual cells (*Figure 3D*), the spike times of individual cells and all cells collectively (*Figure 3C*), and aggregate measures such as the spike density function (*Szucs, 1998*) per cell type and the LFP analog (*Figure 3A and 4*; see also *Figure 4—figure supplement 1*). In all of these measures of network activity, theta was apparent within one theta period of the simulation start. The theta oscillation was stable, maintaining a steady power level throughout the duration of the oscillation (*Figure 4A*). To our knowledge, this is the first strictly data-driven, full-scale computational network model of the CA1 that exhibits spontaneous theta rhythm without rhythmic synaptic inputs.

In addition to theta rhythm, the model network displayed gamma oscillations (25–80 Hz; *Figures 3B* and *4D*), as expected based on in vivo data (*Soltesz and Deschênes, 1993*; *Tort et al., 2009*; *Colgin and Moser, 2010*) and in vitro slice data showing 65–75 Hz gamma oscillations arising in response to theta rhythmic network stimulation (*Butler et al., 2016*). The amplitude envelope of the gamma oscillation was phase-locked to the theta rhythm (*Figures 3A,B* and *4C*), as it is in the biological CA1, representing cross-frequency coupling (*Soltesz and Deschênes, 1993*; *Bragin et al., 1995*; *Buzsáki et al., 2003*; *Jensen and Colgin, 2007*; *Belluscio et al., 2012*). The highest amplitude of the gamma oscillations in the model was observed at the theta trough (0°/360°) in the pyramidal layer LFP analog (*Figure 4C*). Because the current study focused primarily on theta oscillations and experimental data from the isolated CA1 are available only for the theta rhythm (*Goutagny et al., 2009*; *Amilhon et al., 2015*), the gamma oscillations were not examined further in the present study.

These results demonstrate that, in spite of gaps in our knowledge, our model was sufficiently well-constrained by experimental data that it generated theta and gamma oscillations on its own, without extrinsic rhythmic inputs or deliberate tuning of intrinsic parameters.

Although we generally refrained from deliberately compensating for missing parameters in this paper, it is of course possible to do so. For example, as mentioned above, no sufficiently detailed information was available for certain interneuron types. Therefore, these lesser-known interneurons

**Table 4.** Current injection levels used to characterize interneuron current sweeps in *Figure 2D–G*.

| Cell type | Hyper. (pA) | Step size (pA) | Depol. (pA) |
|---|---|---|---|
| PV+ B. | −300 | 50 | +500 |
| CCK+ B. | −100 | 20 | +80 |
| O-LM | −130 | 30 | +80 |
| NGF | −130 | 20 | +190 |

were not included in the model, which meant that inhibition received by the pyramidal cells was probably weaker than in the biological situation. Indeed, the pyramidal cells in our model described above (*Figures 3* and *4*) tended to fire more than they typically do during theta oscillations in vivo (e.g., [*Soltesz and Deschênes, 1993*; *Robbe et al., 2006*]). Is the higher firing frequency of the pyramidal cells related to the weaker inhibition? To answer the latter question, in a subset of the simulations we artificially scaled up inhibition in the model to match the inhibitory synapse numbers on CA1 pyramidal cells that were expected from electron microscopic reconstructions of pyramidal cell dendrites and somata (*Megías et al., 2001*; *Bezaire and Soltesz, 2013*). The rationale for scaling up inhibition in this way was that, as described in detail in *Bezaire and Soltesz (2013)*, the estimates of local inhibitory inputs to pyramidal cells were different when based on experimental observations of presynaptic anatomy (local boutons available for synapsing from distinct types of intracellularly filled and reconstructed interneurons) as opposed to postsynaptic anatomy (inhibitory post-synaptic densities on pyramidal cell dendrites). In simulations with the model containing this rationally scaled up inhibition, only 1% of the pyramidal cells were active, and they fired at a low rate of 1.8 Hz (data not shown), closely resembling the in vivo condition (*Soltesz and Deschênes, 1993*; *Robbe et al., 2006*). Therefore, the model was capable of reproducing the experimentally observed relatively low-firing frequencies for the principal cells during theta oscillations in vivo. However, because the source of the additional inhibition onto CA1 principal cells has not yet been experimentally identified, we used the connectivity estimates as constrained by experimental observations of axonal boutons and lengths in the full scale model (without the scaled-up inhibition) described above (*Figures 3* and *4*) in the subsequent computational experiments.

## Mechanism of theta generation and phase-preferential firing of interneurons in the full-scale model of the isolated CA1

Next, we examined the onset of the theta rhythm and the firing patterns of the various cell types in the model circuit during theta oscillations (*Figure 5*, *Table 5*, and *Figure 5—source data 1–11* ). As mentioned above, distinct interneuronal types, defined based on their selective axonal innervation patterns of the postsynaptic domains of pyramidal cells, exhibit characteristic, cell-type-specific preferred phases of firing during theta oscillations in vivo (*Klausberger et al., 2003*; *2004*, *2005*; *Ferraguti et al., 2005*; *Jinno et al., 2007*; *Fuentealba et al., 2008*; *Varga et al., 2012*; *Lapray et al., 2012*; *Katona et al., 2014*; *Varga et al., 2014*). Importantly, this fundamental property emerged spontaneously from the full-scale model, without purposeful tuning of parameters except the mean spiking frequency and synaptic strength of the artificial stimulating cells to set the incoming excitation levels from afferents (see Materials and methods for details). As expected, the numerically dominant pyramidal cells, whose intracellular membrane potential oscillations to a large extent generate and underlie the extracellular LFP signal during theta oscillations (*Buzsáki et al., 2012*), preferentially discharged around the trough $0^o/360^o$ of the LFP analog theta rhythm (*Figure 5A*).

Interneurons in the model preferentially fired at specific phases of theta oscillations, depending on the cell type. Their phase preferences fell into two broad categories (*Figure 5A*). The cells belonging to the first group, including the PV+ basket cells, bistratified cells and O-LM cells, were most likely to fire at the theta trough compared to other theta phases. Since these cells received substantial excitatory inputs from local CA1 pyramidal cells both in the biological state and in the model (*Bezaire and Soltesz, 2013*), their firing in the isolated CA1 model was probably driven by the pyramidal cell discharges around the theta trough. In contrast, the second group of cells,

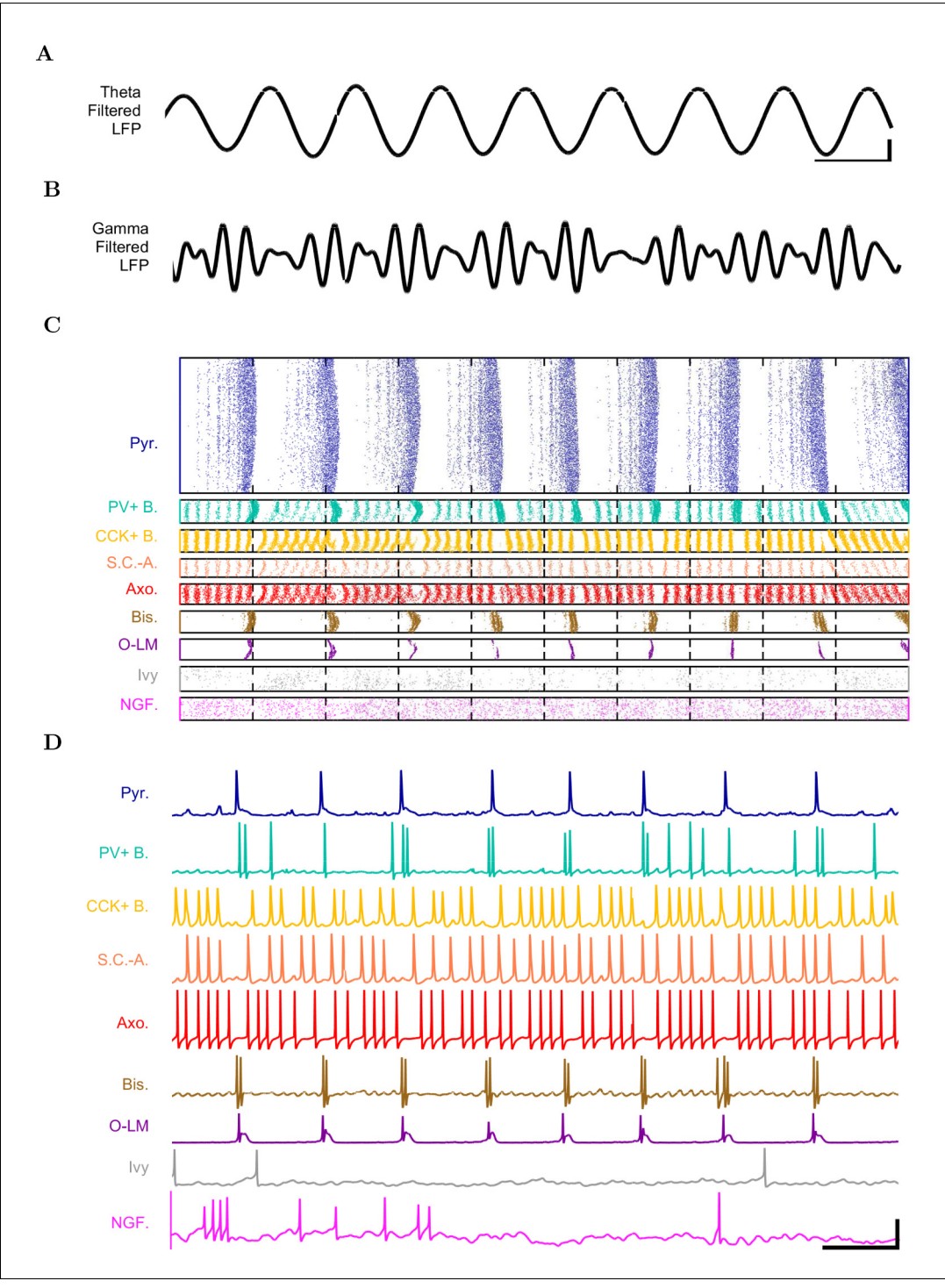

**Figure 3.** Detailed network activity. (**A–D**) One second of network activity is shown. (**A–B**) The LFP analog, filtered at (**A**) the theta range of 5–10 Hz and (**B**) the low gamma range of 25–40 Hz, shows consistent theta and gamma signals. Scale bar represents 100 ms and 0.2 mV (theta) or 0.27 mV (gamma) for filtered LFP traces. (**C**) Raster of all spikes from cells within 100 $\mu$m of the reference electrode point. (**D**) Representative intracellular somatic membrane potential traces from cells near the reference electrode point. Scale bar represents 100 ms and 50 mV for the intracellular traces.

The following source data is available for figure 3:

**Source data 1.** Filtered analog local field potential of model network.

*Figure 3 continued on next page*

*Figure 3 continued*
**Source data 2.** Spike Raster.
**Source data 3.** Somatic membrane potential recordings.

including the ivy and neurogliaform cells, the CCK+ basket cells and the axo-axonic cells, fired least around the theta trough, leading to an inverted firing probability distribution relative to the first group of interneurons (*Figure 5A*). Their differing phase preferences were most likely due to a combination of weak or non-existent excitatory inputs from local CA1 pyramidal cells and inhibition from the interneurons that prominently discharged around the theta trough. In general agreement with the first group of cells being strongly and rhythmically driven by the local pyramidal cells, there was a correlation between the phase preference and the strength of modulation (*Figure 5C*; see Materials and methods), with the cells discharging around the trough all showing strong modulation of firing.

These results were in line with recent data from the isolated CA1 preparation in vitro (*Ferguson et al., 2015*) which showed that cells belonging to the broadly defined SOM+ and PV+ classes (identified using genetic drivers) displayed phase preferences similar to the O-LM, PV+ basket and bistratified cells in our model (note that Ferguson and colleagues used LFP theta recorded in the stratum radiatum as reference, which is approximately 180 degrees out of phase with the pyramidal cell layer theta used in this paper). In addition, the interneuronal phase preferences in the model were also remarkably similar to in vivo data from anesthetized animals (*Figure 5B*; because no data are available on the phase preferential firing of morphologically identified interneurons from the isolated CA1 preparation, comparison is made here with results from anesthetized animals, from which the most complete data sets are available; see also Discussion). Specifically, the majority (71%; 5/7) of the interneuron types for which there were experimental data, including the CCK+ basket, axo-axonic, bistratified, O-LM and neurogliaform cells, showed similar preferential maxima in their firing probabilities in the model (*Figure 5A*) and in vivo (*Figure 5B*). The largest differences between the model and the in vivo phase-preferential firing occurred for the PV+ basket cells and the ivy cells, suggesting that during theta oscillations in vivo these cells may be strongly driven by CA3 afferents active during the late falling phase of the theta cycle (*Colgin and Moser, 2010*); note that PV+ cells receive a high number of excitatory inputs on their dendrites compared to other interneuron classes (*Gulyás et al., 1999*). A comparison of the model and the anesthetized in vivo data is illustrated in *Figure 5D*, where the arrows indicate the shift required for the model phase preferences (*Figure 5A*) to equal the in vivo (*Figure 5B*) phase preferences; note that the required shifts (arrows) are small for all interneuron types except PV+ basket and ivy cells. A clear majority of the interneuronal types in the model showed phase preferences similar to the in vivo condition where rhythmically discharging afferent inputs are present, indicating that theta-preferential discharges are to a large extent determined by the wiring properties of the CA1 circuit itself.

## Perturbation experiments indicate a key role for interneuronal diversity in the emergence of spontaneous theta

Importantly, the ability to generate theta oscillations, phase-locked gamma oscillations, and theta-related phase-preferential firing of distinct interneuronal subtypes was not a universal property of the model. As shown in *Figure 6A*, our strongly constrained model only exhibited spontaneous theta oscillations at certain levels of afferent excitation. The results described above (*Figures 3–5*) were obtained with an afferent excitation level of 0.65 Hz (labeled as 'Control' in *Figure 6A*), meaning that each excitatory afferent cell excited the model network with a Poisson-distributed spike train having a Poisson mean interspike interval (ISI) corresponding to a firing rate of 0.65 Hz. When the excitation level decreased below 0.65 Hz, the theta rhythm fell apart, and when the excitation level increased beyond 0.80 Hz, theta power also started to drop significantly as the oscillation frequency rose out of theta range (*Figure 6* and *Figure 6—figure supplement 1*; *Figure 6—source data 1– 2*), evolving into a beta oscillation (*Engel and Fries, 2010*). These data indicate that while synaptic-cellular organization of the CA1 circuit enables the intrinsic, within-CA1 generation of theta waves,

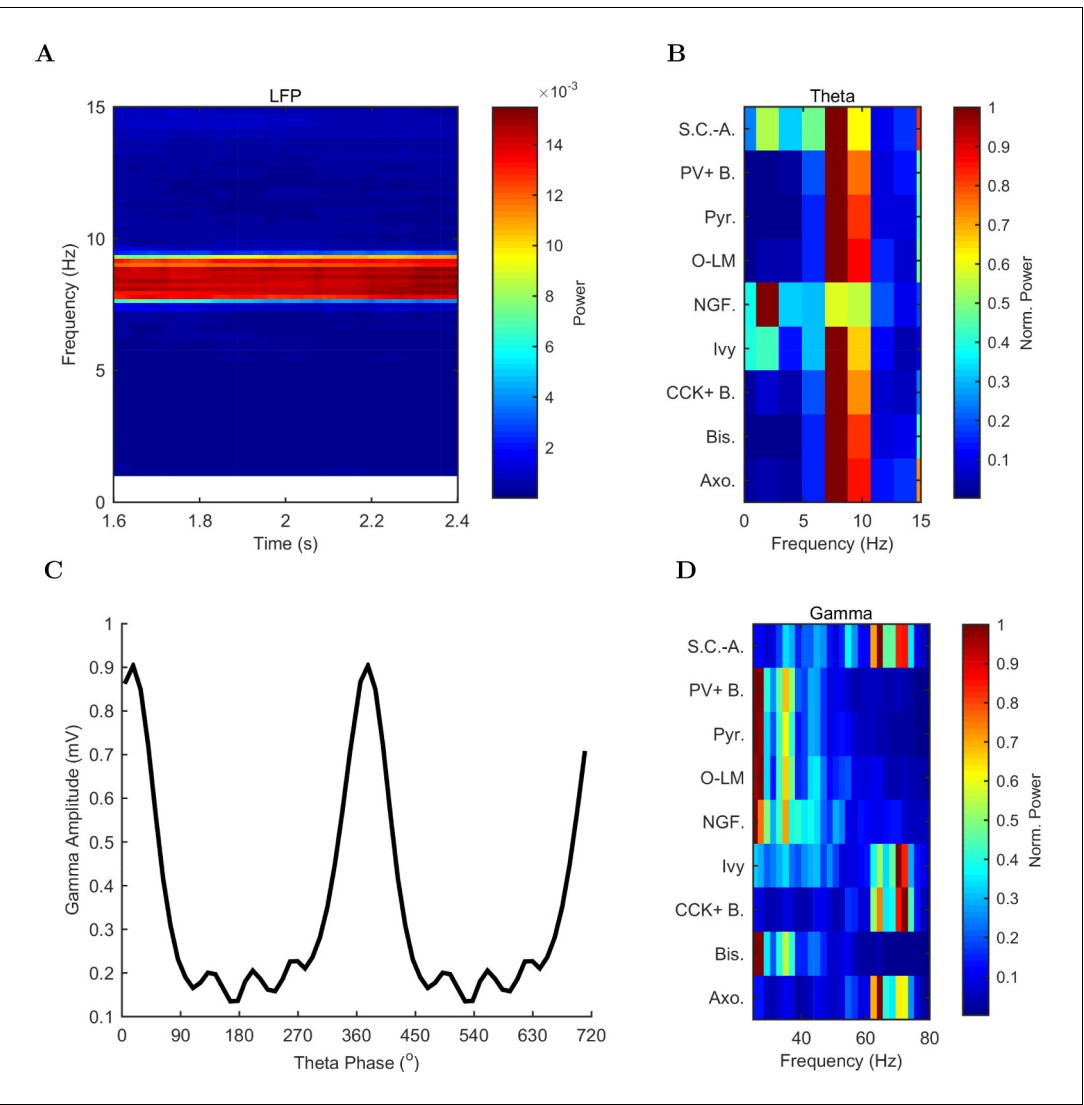

**Figure 4.** Spectral analysis of model activity. (**A**) A spectrogram of the local pyramidal-layer LFP analog (including contributions from all pyramidal cells within 100 $\mu$m of the reference electrode and 10% of pyramidal cells outside that radius) shows the stability and strength of the theta oscillation over time. The oscillation also featured strong harmonics at multiples of the theta frequency of 7.8 Hz. (**B,D**) Welch's periodogram of the spike density function for each cell type, normalized by cell type and by displayed frequency range, shows the dominant network frequencies of (**B**) theta (7.8 Hz) and (**D**) gamma (71 Hz). Power is normalized to the peak power displayed in the power spectrum for each cell type. (**C**) Cross-frequency coupling between theta and gamma components of the LFP analog shows that the gamma oscillation is theta modulated. The gamma envelope is a function of the theta phase with the largest amplitude gamma oscillations occurring at the trough of the theta oscillation. Following convention, the theta trough was designated 0°/360°; see e.g., *Varga et al. (2012)*. A graphical explanation of the relation between a spike train and its spike density function is shown in *Figure 4—figure supplement 1*.

The following source data and figure supplement are available for figure 4:

**Source data 1.** Raw analog local field potential of model network.
**Source data 2.** Spike Density Functions of each cell type in control network.
**Figure supplement 1.** Different views of cell activity.

the circuit is predisposed to exhibit theta oscillations only under particular excitatory input conditions. The observation that, under certain conditions the model network can oscillate at frequencies between 12 and 20 Hz, is in agreement with recent experimental findings that rhythmic driving of septal PV+ cells can reliably entrain the hippocampus in a 1:1 ratio up to frequencies of 20 Hz (*Dannenberg et al., 2015*).

Does the parameter sensitivity of the theta rhythm also apply to recurrent excitation from pyramidal cells and inhibition from CA1 interneurons? In order to answer the latter question, we tested whether the theta rhythm was differentially sensitive to the contribution of each inhibitory cell type (*Figure 6B*). We characterized the contribution of each local CA1 cell type to the theta rhythm by muting the output of the cell type so that its activity had no effect on the network. First, we studied the role of the recurrent collaterals of pyramidal cells, which contact mostly interneurons and, less frequently, other pyramidal cells (*Bezaire and Soltesz, 2013*). When we muted all the outputs from pyramidal cells, theta rhythm disappeared (bar labeled 'Pyr' in *Figure 6B*), indicating that the recurrent collaterals of pyramidal cells play a key role in theta oscillations.

Interestingly, muting the relatively rare CA1 pyramidal cell to pyramidal cell excitatory connections alone (each pyramidal cell contacts 197 other pyramidal cells in the CA1; *Bezaire and Soltesz, 2013*) was sufficient to collapse the theta rhythm (bar labeled 'None' in *Figure 6C*); key roles for inter-pyramidal cell excitatory synapses within CA1 have been suggested for sharp wave ripple oscillations as well (*Maier et al., 2011*). Furthermore, the parameter-sensitivity of the theta rhythm was also apparent when examining the role of pyramidal cell to pyramidal cell connections, because theta power dramatically decreased when these connections were either increased (doubled) or decreased (halved) from the biologically observed 197 (*Figure 6C*). Next, we investigated the effects of muting the output from each interneuron type. Silencing the output from any of the fast-spiking, PV family interneurons (PV+ basket, axo-axonic, or bistratified cells), CCK+ basket cells, or neurogliaform cells also strongly reduced theta power in the network (*Figure 6B*). In contrast, muting other interneuronal types (S.C.-A cells, O-LM cells, or ivy cells) had no effect on this form of theta oscillations generated by the intra-CA1 network (*Figure 6B*). In additional disinhibition studies simulating optogenetic experimental configurations, partial muting of all PV+ outputs (PV+ basket, bistratified, and axo-axonic cells together) had a larger effect than partial muting of all SOM+ outputs (O-LM and bistratified cells); see *Figure 6D*. Reassuringly, these results were in overall agreement with experimental data from the isolated CA1 preparation indicating that optogenetic silencing of PV+ cells, but not SOM+ cells such as the O-LM cells, caused a marked reduction in theta oscillations (*Amilhon et al., 2015*). The differential effects of silencing PV+ versus SOM+ cells could also be obtained in a rationally simplified model called the Network Clamp, where a single pyramidal cell was virtually extracted from the full-scale CA1 network with all of its afferent synapses intact (for further details, see *Bezaire et al., 2016a*).

Since the diverse sources of inhibition from the various interneuronal types are believed to enable networks to achieve more complex behaviors, including oscillations (*Soltesz, 2006*; *Rotstein et al., 2005*; *Kepecs and Fishell, 2014*), we next tested if reducing the diversity of interneurons in the model would affect its ability to produce spontaneous theta oscillations. Surprisingly, giving all interneurons a single electrophysiological profile appeared to create conditions that were not conducive for the appearance of spontaneous theta oscillations regardless of which interneuronal profile was used (*Figure 6E*; note that the cells still differed in the strengths, distribution, and identities of their incoming and outgoing connections after this manipulation). To probe this finding further, we focused on PV+ basket cells, which have been implicated in theta generation in vivo (*Soltesz and Deschênes, 1993*; *Buzsáki, 2002*; *Stark et al., 2013*; *Hu et al., 2014*) and exhibited strong theta power in their spiking in the control network model (*Figure 4B*). We gradually altered ('morphed') the properties of all other model interneuron types until they became PV+ basket cells, by first converging their electrophysiological profiles, then additionally their synaptic kinetics and incoming synapse weights, then also their incoming synapse numbers, and finally their outgoing synaptic weights and numbers (*Figure 6F*; *Table 7*). Theta was not apparent in any intermediate steps nor in the final network where all interneurons had become PV+ basket cells ('All PV+B' in *Figure 6F*). Furthermore, introduction of cell to cell variability in the resting membrane potential of interneurons in the 'All PV+B' configuration at the biologically observed values for PV+ basket cells also failed to restore theta ('Var PV+B' in *Figure 6F* shows results with standard deviation of (SD) = 8 mV in the resting membrane potential; SD = 5 mV and SD = 2 mV also yielded no theta; biological SD value: approximately

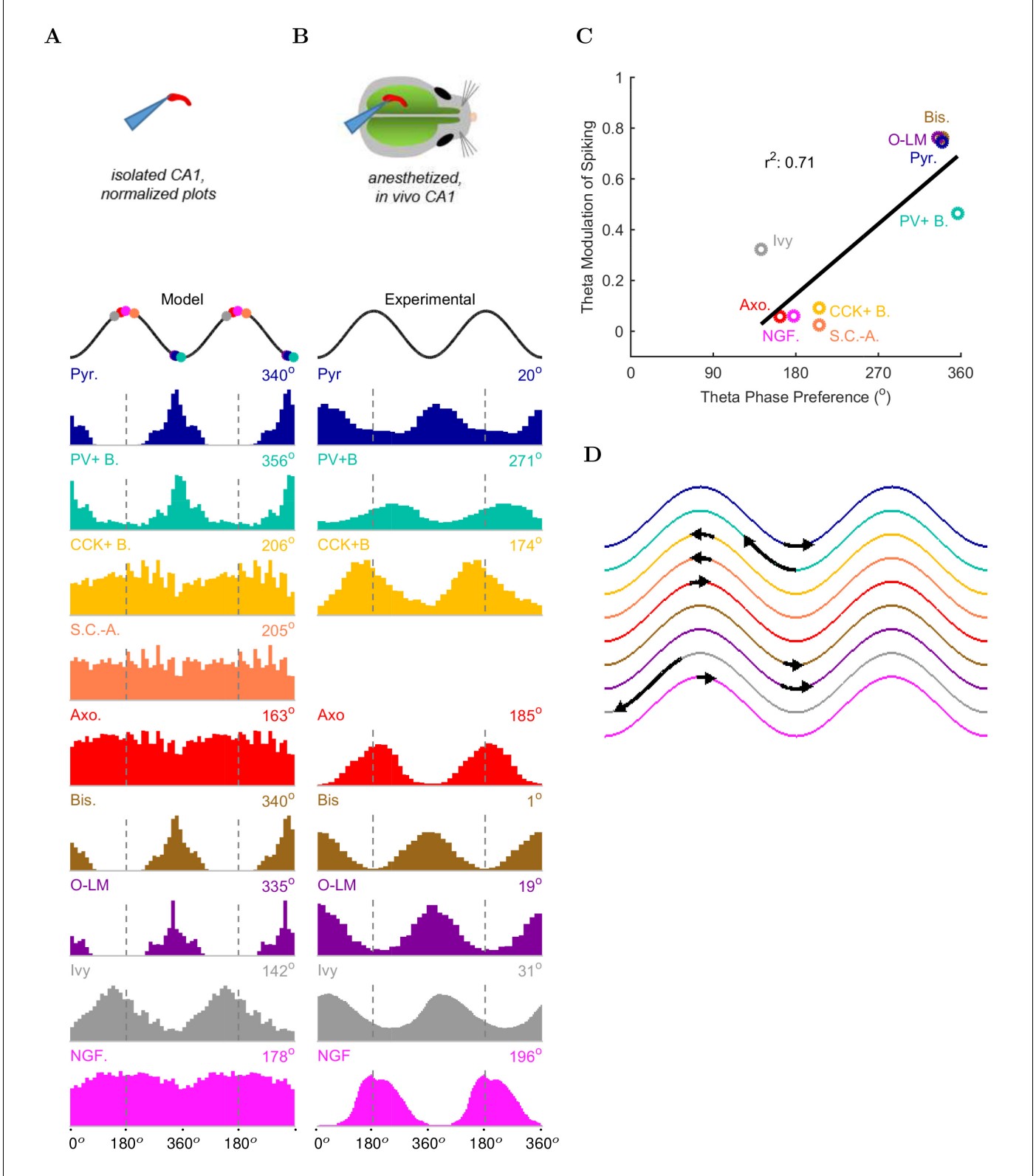

**Figure 5.** Model and experimental cell theta phases. All model results are based on the spiking of the cells within 100 $\mu$m of the reference electrode. (A–B) Firing probability by cell type as a function of theta phase for (**A**) model and (**B**) experimental cells under anesthesia (histograms adapted with permission from *Figure 2*, *Figure 5B* left, and *Figure 6F* respectively from *Klausberger and Somogyi, 2008*; *Fuentealba et al., 2008*; *Fuentealba et al., 2010*). The model histograms are normalized; see *Figure 5—figure supplement 1* for firing rates. (**C**) Theta phase preference and

*Figure 5 continued on next page*

*Figure 5 continued*

theta modulation level were correlated; better modulated cell types spiked closer to the LFP analog trough near the phase preference of pyramidal cells. (**D**) Theta phase preference plotted on an idealized LFP wave for model data (base of arrow signifies the model phase preference and head of the arrow shows the distance to anesthetized, experimental phase preference).

The following source data and figure supplements are available for figure 5:

**Source data 1.** Spike times of axo-axonic cells.
**Source data 2.** Spike times of bistratified cells.
**Source data 3.** Spike times of proximal afferent cells.
**Source data 4.** Spike times of CCK+ basket cells.
**Source data 5.** Spike times of distal afferent cells.
**Source data 6.** Spike times of ivy cells.
**Source data 7.** Spike times of neurogliaform cells.
**Source data 8.** Spike times of O-LM cells.
**Source data 9.** Spike times of PV+ basket cells.
**Source data 10.** Spike times of pyramidal cells.
**Source data 11.** Spike times of Schaffer Collateral-associated cells.
**Figure supplement 1.** Firing rates of model and experimental cells of each type.
**Figure supplement 2.** Theta phase-specific firing preferences of various biological hippocampal cell types as reported in the literature.

5 mV in *Tricoire et al. (2011)* and 2 mV in *Mercer et al. (2012)*). Therefore, although PV basket cells appear to be important for theta-generation both in the biological and the model CA1 network, endowing all interneurons with PV basket cell-like properties does not lead to a network configuration conducive to theta oscillations (*Hendrickson et al., 2015*).

To rule out the possibility that the lack of theta could be due to an inappropriate excitation level in these reduced diversity configurations, we subjected the 'All PV+ B' network to a wide range of incoming excitation levels (*Figure 6G*). Theta rhythm did not appear at any of these excitation levels. While we could not rule out a hypothetical theta regime somewhere in the parameter space of such low-diversity configurations, any theta solution space would likely be smaller and more elusive than we were able to determine in the control configuration (*Figure 6A*).

Taken together, these results indicated, for the first time, that interneuronal diversity itself is an important factor in the emergence of spontaneous theta oscillations from the CA1 network.

## Neurogliaform cell signaling and theta generation in the isolated CA1 model

In agreement with previous predictions (*Capogna, 2011*), the perturbation experiments described above suggested that neurogliaform cells were a necessary component for spontaneous theta to arise in the isolated CA1. We wondered why muting the output from neurogliaform cells, but not the closely related ivy cells, affected theta oscillations (*Figure 6B*), especially since there were fewer neurogliaform cells than ivy cells, and they were less theta modulated (*Figure 5A*). These two model interneuron groups mainly differed in that the neurogliaform cells evoked mixed GABA$_{A,B}$ postsynaptic events (*Price et al., 2005*), whereas the model ivy cells only triggered GABA$_A$ IPSPs (in agreement with a lack of evidence for ivy cell-evoked GABA$_B$ IPSPs). Could the slow kinetics of GABA$_B$

**Table 5.** Preferred theta firing phases for each model cell type.

| Cell type | Firing rate (Hz) | Modulation | | Phase (0°=trough) |
| --- | --- | --- | --- | --- |
| | | Level | p | |
| Axo. | 8.9 | 0.07 | 4.58e − 130 | 163.4 |
| Bis. | 18.0 | 0.76 | 0.00e + 00 | 340.0 |
| CCK+ B. | 54.4 | 0.10 | 0.00e + 00 | 202.8 |
| Ivy | 43.3 | 0.33 | 0.00e + 00 | 142.1 |
| NGF. | 55.1 | 0.07 | 1.46e − 32 | 176.3 |
| O-LM | 17.4 | 0.76 | 0.00e + 00 | 334.7 |
| Pyr. | 6.0 | 0.74 | 0.00e + 00 | 339.7 |
| PV+ B. | 0.9 | 0.46 | 0.00e + 00 | 356.8 |
| S.C.-A. | 5.2 | 0.03 | 1.13e − 07 | 197.9 |

IPSPs contribute to the pacing of the theta oscillations? Indeed, when we selectively removed the $GABA_B$ component of all neurogliaform cell outgoing synaptic connections, theta power was strongly reduced (*Figure 6H*). To test whether the contribution of the $GABA_B$ receptors was due to their slow kinetics, we artificially sped up the $GABA_B$ IPSPs so that they had $GABA_A$ kinetics but conserved their characteristic large charge transfer. This alteration was implemented by scaling up the $GABA_A$ synaptic conductance at neurogliaform cell output synapses to achieve a similar total charge transfer as the control $GABA_{A,B}$ mixed synapse (*Figure 6—figure supplement 2*). As shown in *Figure 6H* (green bar), theta activity was restored when the neurogliaform cell output synapses had no slow $GABA_B$ component, only a scaled up fast $GABA_A$ IPSP with a charge transfer equivalent to the mixed $GABA_{A,B}$ synapses. Therefore, muting the neurogliaform cells strongly disrupted the theta oscillations not because the theta oscillations required the slow kinetics of $GABA_B$ IPSPs specifically, but because the slow kinetics enabled a large total charge transfer.

## Discussion

### Emergence of theta oscillations from a biological data-driven, full-scale model of the CA1 network

We produced a biologically detailed, full-scale CA1 network model constrained by extensive experimental data (*Bezaire and Soltesz, 2013*). When excited with arrhythmic inputs at physiologically relevant levels (see below), the model displayed spontaneous theta (and gamma) oscillations with phase preferential firing across the nine model cell types (pyramidal cells and eight interneuron classes). Consistent with experimental results (*Goutagny et al., 2009*; *Amilhon et al., 2015*), these oscillations emerged from the network model without explicit encoding, rhythmic inputs or purposeful tuning of intra-CA1 parameters (all anatomical connectivity parameters were exactly as previously published in *Bezaire and Soltesz (2013)*). Cell type-specific perturbations of the network showed that each interneuronal type contributed uniquely to the spontaneous theta oscillation, and that the presence of diverse inhibitory dynamics was a necessary condition for sustained theta oscillations. In addition to characterizing roles for specific network components, these model results generally suggest that the presence of diverse interneuronal types and the intrinsic circuitry of the CA1 network are sufficient and necessary to enable the isolated CA1 to oscillate at spontaneous theta rhythms while supporting distinct phase preferences of each class of hippocampal neuron. These abilities may serve to maintain the stability and robustness of the theta oscillation mechanism as it operates in vivo in diverse behavioral states. The theta rhythm is thought to be important for organizing disparate memory tasks (*Lisman and Idiart, 1995*; *Hasselmo et al., 2002*; *Hasselmo, 2005*; *Lisman and Jensen, 2013*; *Siegle and Wilson, 2014*), and a CA1 network which has evolved a predisposition to oscillate at theta and gamma frequencies may enable more efficient processing of the phasic input it receives in vivo (*Akam and Kullmann, 2012*; *Fries, 2015*). In turn, phase preferential firing may aid information processing tasks by providing order and allowing multiple channels of

**Table 6.** Firing rates and theta phase preferences for various cell types in various conditions. Theta phase is relative to the LFP recorded in the pyramidal layer, where 0° and 360° are at the trough of the oscillation. non: non-theta/non-SWR state. SWR: sharp wave/ripple. u+k and x: urethane + supplemental doses of ketamine and xylazine.

| Cell type | Firing rate (Hz) | | | Theta phase (°) | State of animal | Animal | Ref. |
|---|---|---|---|---|---|---|---|
| | Theta | Non | SWR | | | | |
| ADI | 8.60 | 0.06 | 0.25 | 156 | anesth: u+k and x | rat | (*Klausberger et al., 2005*) |
| Axo-axonic | 17.10 | 3.50 | 2.95 | 185 | anesth: u+k and x | rat | (*Klausberger et al., 2003*) |
| Axo-axonic | 27 | | 27 | 251 | awake, head restraint | mouse | (*Varga et al., 2014*) |
| Bistratified | 5.90 | 0.90 | 42.80 | 1 | anesth: u+k and x | rat | (*Klausberger et al., 2004*) |
| Bistratified | 34 | | 36 | 0 | awake, head restraint | mouse | (*Varga et al., 2014*) |
| Bistratified | 30.42 | 27.65 | 35.82 | 2 | awake | rat | (*Katona et al., 2014*) |
| CCK+ Basket | 9.40 | 1.60 | 2.70 | 174 | anesth: u+k and x | rat | (*Klausberger et al., 2005*) |
| Ivy | 0.70 | 1.70 | 0.80 | 31 | anesth: u+k and x | rat | (*Fuentealba et al., 2008*) |
| Ivy | 2.80 | 2.10 | 5.20 | 46 | awake, free | rat | (*Lapray et al., 2012*) |
| Ivy | 2.40 | 3.00 | 6.70 | | awake, free | rat | (*Fuentealba et al., 2008*) |
| NGF | 6.00 | 2.65 | 2.30 | 196 | anesth: u+k and x | rat | (*Fuentealba et al., 2010*) |
| O-LM | 4.90 | 2.30 | 0.23 | 19 | anesth: u+k and x | rat | (*Klausberger et al., 2003*) |
| O-LM | 29.80 | 10.40 | 25.40 | 346 | awake, head restraint | mouse | (*Varga et al., 2012*) |
| O-LM | 17.30 | 11.88 | 18.95 | 342 | awake | rat | (*Katona et al., 2014*) |
| PPA | 5.75 | 1.95 | 1.50 | 100 | anesth: u+k and x | rat | (*Klausberger et al., 2005*) |
| PV+ Basket | 7.30 | 2.74 | 32.68 | 271 | anesth: u+k and x | rat | (*Klausberger et al., 2003*) |
| PV+ Basket | | | | 234 | anesth: u+k and x | rat | (*Klausberger et al., 2005*) |
| PV+ Basket | 21.00 | 6.50 | 122.00 | 289 | awake, free | rat | (*Lapray et al., 2012*) |
| PV+ Basket | 25.00 | 8.20 | 75.00 | 307 | awake, head restraint | mouse | (*Varga et al., 2012*) |
| PV+ Basket | 28 | | 77 | 310 | awake, head restraint | mouse | (*Varga et al., 2014*) |
| Pyramidal | | | | 20 | anesth: u+k and x | rat | (*Klausberger et al., 2003*) |
| Trilaminar | 0.20 | 0.10 | 69.00 | trough | anesth: u+k and x | rat | (*Ferraguti et al., 2005*) |
| Double Proj. | 0.90 | 0.55 | 26.93 | 77 | anesth: u+k and x | rat | (*Jinno et al., 2007*) |
| Oriens Retro. | 0.53 | 0.37 | 53.37 | 28 | anesth: u+k and x | rat | (*Jinno et al., 2007*) |
| Radiatum Retro. | 5.15 | 1.90 | 0.70 | 298 | anesth: u+k and x | rat | (*Jinno et al., 2007*) |

information to be processed in parallel (*Jensen and Lisman, 2000*; *Hasselmo et al., 2002*; *Womelsdorf et al., 2007*; *Schomburg et al., 2014*; *Jeewajee et al., 2014*; *Maris et al., 2016*).

Importantly, theta oscillations appeared only within certain levels of excitatory afferent activity, around 0.65 Hz for the average firing rate of the Poisson-distributed spike trains. When the 454,700 stimulating afferents in the model (representing the CA3 and entorhinal synapses; calculated in *Bezaire and Soltesz (2013)*) are active at a Poisson mean of 0.65 Hz, they generate approximately 37,900 incoming spikes / theta cycle, given a theta frequency of 7.8 Hz (*Equation 1*).

$$454,700 \text{ afferents} * \frac{0.65 \text{ spikes/s}}{7.8 \text{ theta cycles/s}} = 37,892 \text{ spikes/cycle} \tag{1}$$

Is the latter number of spikes in the afferents to the CA1 network within a physiologically plausible range? The biological CA1 network receives most of its input from CA3 and entorhinal cortical layer III (ECIII), and it has been estimated that about 4% of CA3 pyramidal cells fire up to four spikes per theta wave (*Gasparini and Magee, 2006*). We previously estimated 204,700 pyramidal cells in ipsilateral CA3 (*Bezaire and Soltesz, 2013*), giving an estimated 32,750 spikes from ipsilateral CA3 per theta cycle (*Equation 2*).

$$204,700 \text{ cells} * 0.04 \text{ cell fraction} * 4 \text{ spikes/cell} = 32,752 \text{ spikes} \tag{2}$$

About 250,000 principal cells from ipsilateral ECIII synapse onto the CA1 region (*Andersen et al., 2006*), and approximately 2% of these cells are active per theta cycle at a low firing rate (*Csicsvari et al., 1999*; *Mizuseki et al., 2009*). Therefore, ECIII cells could provide 5000 input spikes to ipsilateral CA1 (*Equation 3*).

$$250,000 \text{ cells} * 0.02 \text{ cell fraction} * 1 \text{ spike/cell} = 5,000 \text{ spikes} \tag{3}$$

Therefore, about 37,750 spikes per theta cycle arrive from ipsilateral CA3 and entorhinal cortex to the CA1 network in vivo, which is reassuringly close to the our modeling results indicating that robust theta emerged when the CA1 network model received approximately 37,900 afferent spikes per theta cycle. Thus, the model has the capacity to process a biologically realistic number of spike inputs per cycle while maintaining the theta rhythm.

Our results obtained using the 0.65 Hz excitation indicated that the CA1 model network exhibited phenomena that corresponded well with experimental results, for example, on the differential roles of PV+ basket cells and OLM cells. In addition, the simulations unexpectedly revealed that interneuronal diversity itself may also be important in theta generation, since conversion of all interneurons into fast spiking PV+ basket cells did not result in a network that was conducive for the emergence of theta, in spite of the key role of the PV+ basket cells in hippocampal oscillations. The modeling results also provided the interesting insight that GABA$_B$ receptors may play important roles in slow oscillations such as the theta rhythm not because their slow kinetics pace the oscillations, but because their slow kinetics enable a massive charge transfer. This insight was illuminated by the fact that slow GABA$_B$ synapses were not necessary for theta as long as their large charge was carried by the fast GABA$_A$ synapses. However, we had to increase the conductance of the GABA$_A$ synapse almost 300 times to achieve a similar charge transfer as that conveyed by the GABA$_B$ synapse. Such a large conductance is not biologically realistic, indicating that the key role for GABA$_B$ synapses may be to allow the large synaptic charge transfer via a temporal distribution. Indeed, the importance of GABA$_B$ receptors has also been indicated by a number of recent experimental studies, for example, in the modulation of theta and gamma oscillations (*Kohl and Paulsen, 2010*), setting of spike timing of neuron types during theta (*Kohl and Paulsen, 2010*), and playing a role in cortical oscillations and memory processes (*Craig and McBain, 2014*).

In addition to identifying key roles for certain inhibitory components (PV+ interneurons, neurogliaform cells, GABA$_B$, and interneuron diversity), our results also highlighted the importance of the recurrent excitatory collaterals from CA1 pyramidal cells in theta generation in the model of the isolated CA1 network. While it may be expected that isolated theta generation would require local pyramidal cells to provide rhythmic, recurrent excitation to interneurons, our simulations additionally showed that the relatively rare pyramidal cell to pyramidal cell local excitatory connections were also required.

Based on our results, we hypothesize that the inhibitory and excitatory connections within CA1 that were identified to be critical in our perturbation ('muting') simulations (*Figure 6B*) interact to generate the theta waves in the model as follows. Pyramidal cells preferentially discharge at the trough of the LFP analog, strongly recruiting especially the PV+ basket and bistratified cells (green and brown raster plots in *Figure 3C*), which, in turn, cause a silencing of the pyramidal cells (blue raster plot in *Figure 3C*) for about the first third of the rising half (i.e., from 0° to about 60°) of the LFP analog theta cycle. As the pyramidal cells begin to emerge from this period of strong inhibition, initially only a few, then progressively more and more pyramidal cells reach firing threshold, culminating in the highest firing probability at the theta trough, completing the cycle. The progressive recruitment of pyramidal cells during the theta cycle appears to be paced according to gamma (see blue raster plot in *Figure 3C*), and it is likely that the intra-CA1 collaterals of the discharging pyramidal cells play key roles in the step-wise (gamma-paced) recruitment of more and more pyramidal cells as the cycle approaches the following trough. The predicted key roles for physiological pyramidal cell to pyramidal cell connections in theta-gamma generation during running may be tested in future experiments.

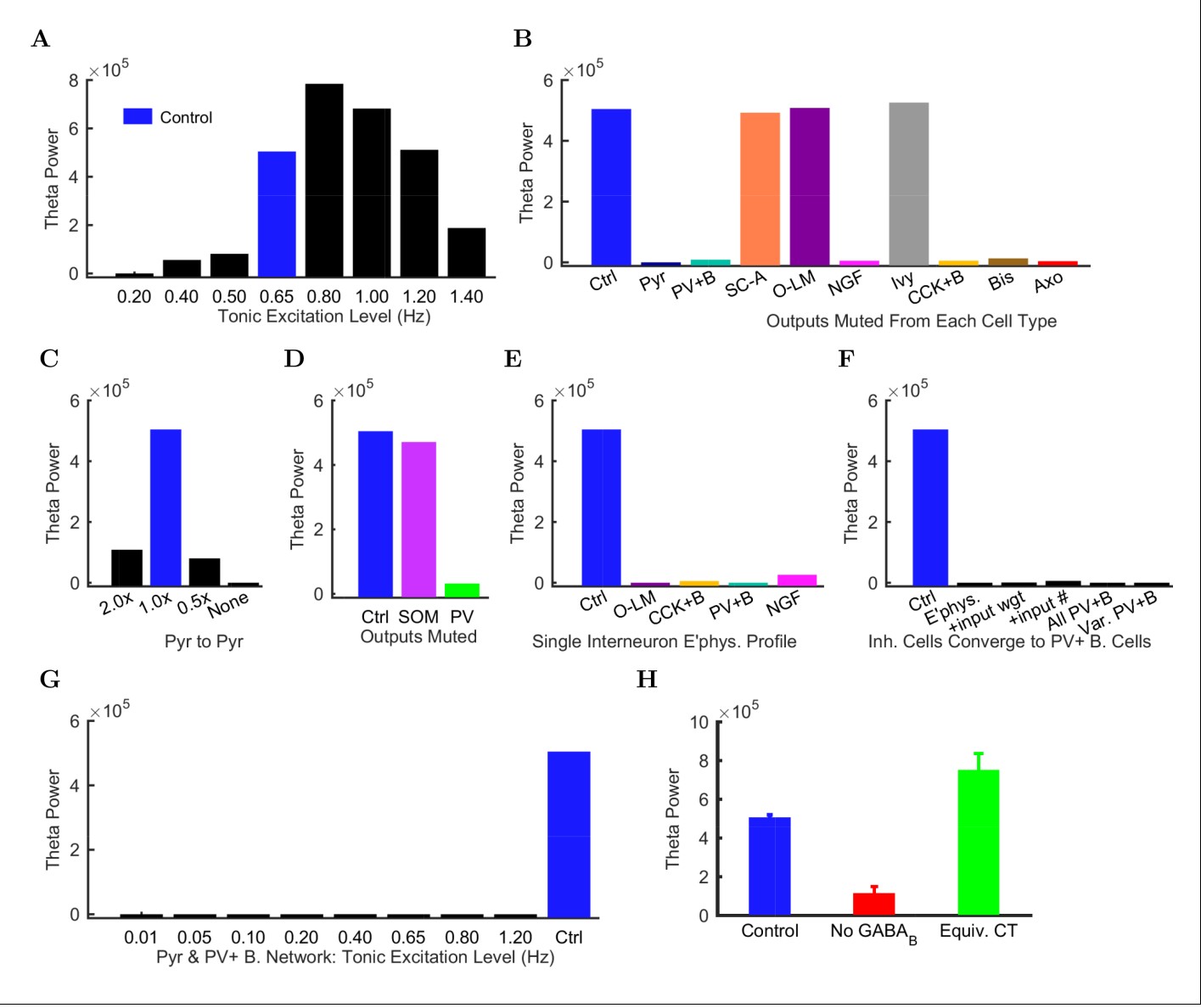

**Figure 6.** Altered network configurations. Oscillation power (in mV2²/Hz) of the spike density function (SDF) for pyramidal cells within 100 $\mu$m of the reference electrode, at the peak frequency within theta range (5–10 Hz) in altered network configurations. For corresponding peak frequencies, see *Figure 6—figure supplement 1*. (**A**) Theta is present at some excitation levels. (**B**) Muting each cell type's output caused a range of effects. (**C**) The stability and frequency of spontaneous theta in the network was sensitive to the presence and number of recurrent connections between CA1 pyramidal cells. (**D**) Partially muting the broad classes of PV+ or SOM+ cells by 50% showed that PV+ muting disrupted the network more than SOM+ muting. (**E**) Theta falls apart when all interneurons are given the same electrophysiological profile, whether it be of a PV+ basket, CCK+ basket, neurogliaform, or O-LM cell. (**F**) Gradually setting all interneuron properties to those of PV+ basket cells did not restore theta. From left to right: control network; PV+ basket cell electrophysiology; also weights of incoming synapses; also numbers of incoming synapses; then all interneurons being PV+ basket cells (with the addition of the output synapse numbers, weights, and kinetics); then variable RMP (normal distribution with standard deviation of 8 mV). (**G**) A wide range in excitation was unable to produce theta in the PV+ B. network. (**H**) Removing the GABA_B component from the neurogliaform synapses onto other neurogliaform cells and pyramidal cells showed a significant drop in theta power. Massively increasing the weight of the GABA_A component to produce a similar amount of charge transfer restored theta power (compare the IPSCs corresponding to each condition in *Figure 6—figure supplement 2*). Standard deviations (n = 3) shown; significance (p=1.8e-05).

The following source data and figure supplements are available for figure 6:

**Source data 1.** Simulation name mapping.

**Source data 2.** SDF of each network condition.

*Figure 6 continued on next page*

*Figure 6 continued*

**Figure supplement 1.** Peak frequencies of oscillations in altered networks.

**Figure supplement 2.** IPSCs from the neurogliaform to pyramidal cell synapse corresponding to the different conditions in *Figure 6H*.

## Rationale for bases of comparison between modeling results with experimental data

Because our model represented the isolated CA1 network, the modeling results were compared with experimental data from the isolated CA1 preparation when possible. Modeling results for which no corresponding experimental data were available from the isolated CA1 preparation, such as the phase preferential firing of individual interneuron types during theta oscillations, were compared with in vivo data from anesthetized animals (*Figure 5B*). Experimental results from anesthetized animals offered the most complete data set (e.g., no experimental data were available on CCK basket cells and neurogliaform cells from awake animals, see *Figure 5—figure supplement 2*). Out of the four interneuronal types for which in vivo data were available from both the awake and anesthetized conditions (*Figure 5—figure supplement 2*), the phase preference of the axo-axonic cell in the model (163°) was closer to the anesthetized phase (185°) than to the awake phase (251°), whereas the PV+ basket cells in the model displayed phase preferential firing (357°) closer to data reported from awake (289°–310°) than anesthetized animals (234°–271°); the precise reasons underlying these differences are not yet clear. In contrast, bistratified and O-LM cells fired close to the trough in the model, under anesthesia and in awake animals, potentially indicating the primary importance of pyramidal cell inputs in driving these interneurons to fire during theta oscillations under all conditions.

While our model is fundamentally a model of the rat CA1 (e.g., in terms of cell numbers and connectivity; see Table 3 in *Bezaire and Soltesz [2013]*), some of the electrophysiology data used for constructing the single cell models (Appendix) came from the mouse. In addition, the experimental data on the isolated CA1 preparation were obtained from both rat (*Goutagny et al., 2009*) and mouse (*Amilhon et al., 2015*), similar to the experimental results on the phase specific firing in vivo (e.g., awake rat: *Lapray et al., 2012*; awake mouse: *Varga et al., 2014*). Because there is no reported evidence for major, systematic differences in key parameters such as the phase specific firing of rat and mouse interneurons in vivo, we did not compare our modeling results with rat and mouse data separately.

A final point concerns the nature of the theta rhythm that emerged in our model. In general, the in vivo theta rhythm has been reported to be either atropine resistant or atropine sensitive, where the former is typically associated with walking and may not be dependent on neuromodulatory inputs, while the latter requires intact, rhythmic cholinergic inputs (*Kramis et al., 1975*). Given that our model did not explicitly represent neuromodulatory inputs, it is likely that the theta that emerged from our model most closely resembled the atropine resistant form. However, it also plausible that both forms of theta benefit from occurring in a network that is predisposed to oscillate at the theta frequency, as the model network results suggested.

## An accessible approach to modeling that balances detail, scale, flexibility and performance

Our results from the strictly data-driven, full-scale CA1 model are consistent with those of earlier models that elegantly demonstrated the basic ingredients capable of producing emergent network oscillations at a range of frequencies in microcircuits and small networks (*Rotstein et al., 2005*; *Siekmeier, 2009*; *Neymotin et al., 2011b*; *2011a*; *Ferguson et al., 2013*). In addition, our modeling approach also provides a full-scale option to advance the recent studies of network activity propagation and information processing during theta (*Cutsuridis et al., 2010*; *Cutsuridis and Hasselmo, 2012*; *Taxidis et al., 2013*; *Saudargiene et al., 2015*). Here, we demonstrated that emergent theta and gamma oscillations and theta phase preferential firing are possible even as additional interneuron types are incorporated and the network is scaled up to full size with realistic connectivity including 5 billion synapses between the 300,000-plus cells of our network model.

**Table 7.** Peak, theta and gamma frequencies and powers of the pyramidal cell spike density function using Welch's Periodogram. As in **Figure 6—figure supplement 1**, networks where no pyramidal cells spiked - resulting in zero power within the spectral analysis of the pyramidal cell spike density function - have their peak frequencies listed as 'n/a' for 'not available'.

| | Theta | | Gamma | | Overall | |
|---|---|---|---|---|---|---|
| Condition | Frequency | Power | Frequency | Power | Frequency | Power |
| Tonic excitation level (Hz) | | | | | | |
| 0.20 | n/a | 0.0e + 00 | n/a | 0.0e + 00 | n/a | 0.0e + 00 |
| 0.40 | 5.9 | 5.6e + 04 | 25.4 | 4.1e + 04 | 13.7 | 6.5e + 04 |
| 0.50 | 9.8 | 8.1e + 04 | 25.4 | 1.0e + 05 | 19.5 | 5.6e + 05 |
| 0.65 (Ctrl.) | 7.8 | 5.0e + 05 | 25.4 | 2.0e + 05 | 7.8 | 5.0e + 05 |
| 0.80 | 9.8 | 7.8e + 05 | 29.3 | 2.6e + 05 | 9.8 | 7.8e + 05 |
| 1.00 | 9.8 | 6.8e + 05 | 29.3 | 1.4e + 05 | 9.8 | 6.8e + 05 |
| 1.20 | 9.8 | 5.1e + 05 | 33.2 | 1.8e + 05 | 11.7 | 8.2e + 05 |
| 1.40 | 9.8 | 1.9e + 05 | 25.4 | 3.4e + 05 | 11.7 | 8.6e + 05 |
| Single Interneuron E'phys. Profile | | | | | | |
| Ctrl | 7.8 | 5.0e + 05 | 25.4 | 2.0e + 05 | 7.8 | 5.0e + 05 |
| O-LM | n/a | 0.0e + 00 | n/a | 0.0e + 00 | n/a | 0.0e + 00 |
| CCK+B | 9.8 | 5.7e + 03 | 62.5 | 6.9e + 05 | 62.5 | 6.9e + 05 |
| PV+B | n/a | 0.0e + 00 | n/a | 0.0e + 00 | n/a | 0.0e + 00 |
| NGF | 5.9 | 2.6e + 04 | 39.1 | 2.4e + 06 | 39.1 | 2.4e + 06 |
| Inh. Cells Converge to PV+ B. Cells | | | | | | |
| Ctrl | 7.8 | 5.0e + 05 | 25.4 | 2.0e + 05 | 7.8 | 5.0e + 05 |
| E'phys. | n/a | 0.0e + 00 | n/a | 0.0e + 00 | n/a | 0.0e + 00 |
| +input wgt | 7.8 | 6.8e + 02 | 44.9 | 1.6e + 06 | 21.5 | 3.4e + 06 |
| +input # | 9.8 | 6.1e + 03 | 31.3 | 1.1e + 06 | 15.6 | 2.0e + 06 |
| All PV+B | n/a | 0.0e + 00 | n/a | 0.0e + 00 | n/a | 0.0e + 00 |
| Var. PV+B | n/a | 0.0e + 00 | n/a | 0.0e + 00 | n/a | 0.0e + 00 |
| Outputs Muted | | | | | | |
| Ctrl | 7.8 | 5.0e + 05 | 25.4 | 2.0e + 05 | 7.8 | 5.0e + 05 |
| SOM | 7.8 | 4.7e + 05 | 27.3 | 1.4e + 05 | 7.8 | 4.7e + 05 |
| PV | 9.8 | 3.2e + 04 | 27.3 | 8.1e + 05 | 13.7 | 1.5e + 06 |
| Pyr to Pyr | | | | | | |
| 2.0x | 9.8 | 1.1e + 05 | 25.4 | 7.3e + 05 | 13.7 | 1.0e + 06 |
| 1.0x (Ctrl.) | 7.8 | 5.0e + 05 | 25.4 | 2.0e + 05 | 7.8 | 5.0e + 05 |
| 0.5x | 7.8 | 8.0e + 04 | 29.3 | 2.2e + 05 | 29.3 | 2.2e + 05 |
| None | 9.8 | 1.1e + 00 | 70.3 | 3.7e + 01 | 70.3 | 3.7e + 01 |
| Outputs Muted From Each Cell Type | | | | | | |
| Ctrl | 7.8 | 5.0e + 05 | 25.4 | 2.0e + 05 | 7.8 | 5.0e + 05 |
| Pyr | 7.8 | 1.1e + 00 | 70.3 | 3.8e + 01 | 70.3 | 3.8e + 01 |
| PV+B | 9.8 | 8.8e + 03 | 29.3 | 1.9e + 06 | 29.3 | 1.9e + 06 |
| SC-A | 9.8 | 4.9e + 05 | 27.3 | 1.8e + 05 | 9.8 | 4.9e + 05 |
| O-LM | 7.8 | 5.1e + 05 | 25.4 | 8.3e + 04 | 7.8 | 5.1e + 05 |
| NGF | 9.8 | 5.2e + 03 | 27.3 | 9.1e + 05 | 13.7 | 1.6e + 06 |
| Ivy | 7.8 | 5.3e + 05 | 25.4 | 2.0e + 05 | 7.8 | 5.3e + 05 |
| CCK+B | 5.9 | 5.5e + 03 | 25.4 | 3.3e + 03 | 3.9 | 5.7e + 03 |
| Bis | 5.9 | 1.3e + 04 | 29.3 | 1.7e + 06 | 29.3 | 1.7e + 06 |

*Table 7 continued on next page*

*Table 7 continued*

| | Theta | | Gamma | | Overall | |
|---|---|---|---|---|---|---|
| Condition | Frequency | Power | Frequency | Power | Frequency | Power |
| Axo | 7.8 | 4.0e + 03 | 33.2 | 1.2e + 06 | 15.6 | 1.9e + 06 |
| Pyr & PV+ B. Network: Tonic Excitation Level (Hz) | | | | | | |
| 0.01 | n/a | 0.0e + 00 | n/a | 0.0e + 00 | n/a | 0.0e + 00 |
| 0.05 | n/a | 0.0e + 00 | n/a | 0.0e + 00 | n/a | 0.0e + 00 |
| 0.10 | n/a | 0.0e + 00 | n/a | 0.0e + 00 | n/a | 0.0e + 00 |
| 0.20 | n/a | 0.0e + 00 | n/a | 0.0e + 00 | n/a | 0.0e + 00 |
| 0.40 | 5.9 | 2.3e + 02 | 25.4 | 1.2e + 02 | 3.9 | 2.4e + 02 |
| 0.65 | n/a | 0.0e + 00 | n/a | 0.0e + 00 | n/a | 0.0e + 00 |
| 0.80 | n/a | 0.0e + 00 | n/a | 0.0e + 00 | n/a | 0.0e + 00 |
| 1.20 | n/a | 0.0e + 00 | n/a | 0.0e + 00 | n/a | 0.0e + 00 |
| Ctrl | 7.8 | 5.0e + 05 | 25.4 | 2.0e + 05 | 7.8 | 5.0e + 05 |

This work is one step in our broader effort to build a 1:1 model of the entire temporal lobe using a hypothesis-driven model development process, where at each stage of model development the models are used to address specific questions. For example, here we employed our newly developed full-scale CA1 model to gain mechanistic insights into the ability of the intra-CA1 circuitry to generate theta oscillations (*Goutagny et al., 2009*). The current CA1 network model can be developed into a whole hippocampal or temporal lobe model by replacing the designed CA3 and entorhinal cortical afferents with biophysically detailed CA3, ECIII, and septal networks. While we design our model networks with the motivation to answer a particular question, we keep in mind their potential usage for a broad range of questions. Previously, we built a dentate gyrus model to study epileptic network dynamics (*Santhakumar et al., 2005*; *Morgan and Soltesz, 2008*) that was then used by several groups to study disparate topics including epilepsy, network mechanisms of inhibition and excitability, simulation optimization, and modeling software (*Migliore et al., 2006*; *Gleeson et al., 2007*; *Hines et al., 2008a*; *2008b*; *Hines and Carnevale, 2008*; *Thomas et al., 2009*; *Winkels et al., 2009*; *Cutsuridis et al., 2010*; *Jedlicka et al., 2010a*, *2010b*; *Thomas et al., 2010*; *Tejada and Roque, 2014*). Our previous model has demonstrated how the resource intensive process of designing a detailed, large-scale model is offset by its potential usage in numerous ways by a multitude of groups. On the other hand, future efforts will be needed to continue to incorporate experimental data obtained by the scientific community on additional, not yet represented parameters into the platform offered by our full-scale CA1 network model, e.g., on cell type-specific gap junctions and short-term plasticity, neuromodulators, diversity of pyramidal cells, glial dynamics, cell to cell variability (e.g., [*Schneider et al., 2014*]) and others.

We developed a flexible and biologically relevant model that uses computational resources efficiently, positioning the model to be used by the broader community for many future questions. Importantly, the model can be run on the Neuroscience Gateway, an online portal for accessing supercomputers that does not require technical knowledge of supercomputing (https://www.nsgportal.org/). The model is public, well documented, and also well characterized in experimentally relevant terms (See Appendix and online links given in Materials and methods). In addition, all the model configurations and simulation result data sets used in this work are available online (*Bezaire et al., 2016b*) at (http://doi.org/10.17605/OSF.IO/V4CEH) so the same simulations can easily be repeated with a future, updated model using SimTracker (*Bezaire et al., 2016a*). Mindful that this model could be used by people with a broad range of modeling experience, we have made freely available our custom software SimTracker (RRID:SCR_014735) that works with the model code to support each step of the modeling process (*Bezaire et al., 2016a*).

## Conclusion and outlook

As highlighted by the BRAIN Initiative, there is an increasing recognition in neurobiology that we must compile our collective experimental observations of the brain into something more cohesive and synergistic than what is being conveyed in individual research articles if we are to fully benefit from the knowledge that we collectively produce (*Ramaswamy et al., 2015*; *Markram et al., 2015*). By assimilating our collective knowledge into something as functional as a model, we can further probe the gaps in our experimental studies, setting goals for future experimental work. On the other hand, as powerful new tools are gathering vast quantities of neuroscience data, the extraction and organization of the data itself are becoming a challenge. At least three large programs are undertaking this challenge: the Hippocampome project (for neuroanatomical and electrophysiological data in the hippocampus of mice; [*Wheeler et al., 2015*]), the Human Brain Project (currently for neuroanatomical and electrophysiological data and models of the rat neocortex, [*Ramaswamy et al., 2015*]), and NeuroElectro (for electrophysiological data from all species and brain areas; [*Tripathy et al., 2014*]). These comprehensive databases create the opportunity to build strongly biology-inspired models of entire networks, with all the cells and synapses explicitly represented. Such models are not subject to the connectivity scaling tradeoff wherein smaller networks have unrealistically low levels of input or unrealistically high connectivity between cells. In addition, such models are usable for investigations into an almost infinite number of questions at any level from ion channels, to synapses, to cell types, to microcircuit contributions. This approach represents a new strategy in computational neuroscience, distinct from and complementary to the use of more focused models whose role is to highlight the potential mechanism of a small number of network components.

The scale, flexibility, and accessibility of our strictly data-driven, full-scale CA1 model should aid the modeling of other large scale, detailed, biologically constrained neural networks. The current CA1 network model produces results in agreement with experimental data, but also extends the results to probe the mechanisms of spontaneous theta generation. It provides specific testable predictions that enable focused design of future experiments, as well as providing an accessible resource for the broader community to explore mechanisms of spontaneous theta and gamma generation. Because the model is available at full scale, it is a relevant resource for exploring the transformation of incoming spatial and contextual information to outgoing mnemonic engrams as part of spatial and memory processing, and other pertinent network dynamics.

## Materials and methods

All results presented in this work were obtained from simulations of computational models. We implemented our CA1 model in parallel NEURON 7.4, a neural network simulator (*Carnevale and Hines, 2005*). The model simulations were run with a fixed time step between 0.01 and 0.025 ms, for a simulation duration of 2000 or 4000 ms (except for *Figure 6D* where one simulation ran for 1600 ms). We executed the simulations on several supercomputers, including Blue Waters at the National Center for Supercomputing Applications at University of Illinois, Stampede and Ranger (retired) at the Texas Advanced Computing Center (TACC), Comet and Trestles at the San Diego Supercomputing Center (SDSC), and the High Performance Computing Cluster at the University of California, Irvine. We used our MATLAB-based SimTracker software tool to design, execute, organize, and analyze the simulations (*Bezaire et al., 2016a*).

### Model development

The CA1 network model included one type of multicompartmental pyramidal cell with realistic morphology and eight types of interneurons with simplified morphology, including PV+ basket cells, CCK+ basket cells, bistratified cells, axo-axonic cells, O-LM cells, Schaffer Collateral-associated cells, neurogliaform cells, and ivy cells.

Model neurons sometimes behave much differently than expected when subjected to current sweep protocols or synaptic inputs that are outside the range of the original protocols used to construct the model. To ensure the model cells exhibited robust biophysical behavior in a wide range of network conditions, we implemented a standard, thorough characterization strategy for each cell type (Appendix).

The behavior of each cell type was characterized using a current injection sweep that matched experimental conditions reported in the literature. The published experimental data were compared side-by-side with model cell simulation results (Appendix). Model cells were connected via NEURON's double exponential synapse mechanism (Exp2Syn), with each connection comprising an experimentally observed number of synapses (see *Table 1*).

The connections between cells were determined with the following algorithm, for each postsynaptic and presynaptic cell type combination:

1. Calculate the distances between every presynaptic cell and postsynaptic cell of the respective types;
2. Compute the desired number and distance of connections, as defined by the presynaptic axonal distance distribution and total number of desired connections between these two types; the total number of incoming connections expected by each postsynaptic cell type is divided into radial distance bins and distributed among the bins according to the Gaussian axonal bouton distribution of the presynaptic cell;
3. Assign each of the possible connections determined in step 2 (connections within the axonal extent of the presynaptic cell) to their respective distance bins, and randomly select a specific number of connections from each bin (the specific number calculated to follow the axonal bouton distribution).

When determining which cells of the model to connect, we distributed all cells evenly within their respective layers in 3D space and enabled random connectivity for cell connections where the postsynaptic cell body fell within the axonal extent of the presynaptic cell (looking in the XY plane only). Each time a connection was established between two cells, the presynaptic cell innervated the experimentally observed number of synapses on the postsynaptic cell. The synapse locations were randomly chosen from all possible places on the cell where the presynaptic cell type had been experimentally observed to innervate. The random number generator used was NEURON's nrnRan4int.

## Biological constraints

The cell number and connectivity parameters were exactly as we reported previously in our in-depth quantitative assessment of anatomical data about the CA1 (*Bezaire and Soltesz, 2013*). In the latter paper that formed the data-base for the current full-scale model, we combined immunohistochemical data about laminar distribution and coexpression of markers to estimate the number of each interneuron type in CA1. We then extracted from the experimental literature bouton and input synapse counts for each cell type and multiplied these counts by our estimated number of each cell and determined the available input synapses and boutons in each layer of CA1. The number of connections each cell type was likely to make with every other cell type was based on the results of our quantitative assessment. As the quantitative assessment did not make detailed, interneuron type-specific estimates of connections between interneurons, we performed additional calculations to arrive at the numbers of connections between each type of interneuron in our model. Briefly, we determined the number of inhibitory boutons available for synapsing on interneurons within each layer of CA1. Then, we distributed these connections uniformly across the available incoming inhibitory synapses onto interneurons that we had calculated for that layer. We calculated available incoming synapses by using published experimental observations of inhibitory synapse density on interneuron dendrites by cell class and layer in CA1, which we combined with known anatomical data regarding the dendritic lengths of each interneuron type per layer. We therefore made the following assumption: All available incoming inhibitory synapses onto interneurons in a layer have an equal chance of being innervated by the available inhibitory boutons targeting interneurons in that layer. For further details of the exact calculations, please see the Appendix.

The electrophysiology of each cell was tuned using a combination of manual and optimization techniques. We first fit each cell's resting membrane potential, capacitance, time constant, and input resistance, followed by hyperpolarized properties such as the sag amplitude and time constant, followed by subthreshold depolarized properties such as a transient peak response, and finally active properties such as spike threshold, rheobase, firing rate, action potential width, height, and afterhyperpolarization. For some cells, we employed the Multiple Run Fitter tool within NEURON to simultaneously fit multiple ion channel conductances. The characterization of each cell type, as well as its comparison to experimental data from the same cell type, is included in the Appendix.

After fitting the cell model properties, we simulated paired recordings to characterize the connections between our model cells. Where experimental data existed for paired recordings, we matched the experimental holding potential and synapse reversal potential, then performed 10 different paired recordings. We characterized the average synapse properties from those 10 runs, including the synaptic amplitude, 10–90% rise time, and decay time constant. Finally, we tuned the synaptic weights and time constants to fit our averages to the experimental data.

To determine the synaptic weights and kinetics for those connections that have not yet been experimentally characterized, we used a novel modeling strategy we call Network Clamp, described in *Bezaire et al. (2016a)*. As experimental paired recording data were not available to directly constrain the synapse properties, we instead constrained the firing rate of the cell in the context of the in vivo network, for which experimental data have been published. We innervated the cell with the connections it was expected to receive in vivo, and then sent artificial spike trains through those connections, ensuring that the properties of the spike trains matched the behavior expected from each cell in vivo during theta (firing rate, level of theta modulation, preferred theta firing phase). Next, we adjusted the weight of the afferent excitatory synapses onto the cell (starting from experimentally observed values for other connections involving that cell type) until the cell achieved a realistic firing rate similar to had been experimentally observed in vivo.

## Stimulation

As none of the model neurons in our model CA1 network are spontaneously active, it was necessary to provide excitatory input to them by stimulating their CA3 and entorhinal cortex synapses. Although the model code is structured to allow the addition of detailed CA3 and cortical inputs, the stimulation patterns used in the present study were not representative of the information content thought to be carried via inputs from those areas, because the focus was on the function of the CA1 network in isolation from rhythmic extra-CA1 influences. In accordance with experimental evidence of spontaneous neurotransmitter release (*Kavalali, 2015*), we modeled the activation of CA3 and entorhinal synapses as independent Poisson stochastic processes. The model neurons were connected to a subset of these afferents, such that they received a constant level of excitatory synaptic input.

We constrained the synapse numbers and positions of the stimulating afferents using anatomical data. To constrain the afferent synapse weights, we used an iterative process to determine the combinations of synaptic weights that enabled most of the interneurons to fire similar to their observed in vivo firing rates (*Figure 5—figure supplement 1* and *Table 6*). First, we used the output of an initial full-scale simulation to run network clamp simulations on a single interneuron type, altering the incoming afferent synapse weights (but not the incoming spike trains) until the interneuron type fired at a reasonable rate. Then, we applied the synaptic weight to the afferent connections onto that interneuron type in the full-scale model. The resulting simulation then led to a new network dynamic as the constrained activity of that interneuron type caused changes in other interneuron activity. We then performed this exercise for each interneuron type as necessary until we achieved a network where all cell types participated without firing at too high of a level. CCK+ cells had a steep response to the weight of the incoming afferent synapses, remaining silent until the weight was increased significantly and then spiking at a high rate, see *Figure 5—figure supplement 1*; the particular difficulty in obtaining the in vivo observed firing rate for CCK+ cells in the model may indicate that in vivo they may be strongly regulated by extra-CA1 inhibitory inputs (e.g., from the lateral entorhinal cortex; see *Basu et al. (2016)* that are not included in the isolated CA1 model).

## Analysis of simulation results

We analyzed the results of each simulation with standard neural data analysis methods provided by our SimTracker software, RRID:SCR_014735, discussed in *Bezaire et al. (2016a)*, including the spike density function (SDF) of all pyramidal neuron spikes (*Szucs, 1998*), the periodogram of the SDF, and the spectrogram of the LFP analog (*Goutagny et al., 2009*). We determined the dominant theta and gamma frequencies for the network as the peak in the power spectral density estimate obtained by the spectrogram, and confirmed that those peaks are identical for the SDF and the LFP analog. After finding a dominant theta or gamma frequency, we then analyzed the level of modulation and preferred firing phase for each cell type. Finally, we calculated the firing rate of each cell type.

### LFP analog

We calculated an approximation of the LFP generated by the model neurons based on the method described by *Schomburg et al. (2012)*. For each pyramidal cell within 100 μm of a reference electrode location in stratum pyramidale (coordinates = longitudinal: 200 μm; transverse: 500 μm; height from base of stratum oriens: 120 μm), the contribution to extracellular potential at each point along the dendritic and axonal morphology was recorded using NEURON's extracellular mechanism and scaled in inverse proportion to the distance from the electrode. In order to reduce the computational load of the simulation, 10% of the pyramidal cells outside the 100 μm radius were randomly selected; their distance-scaled extracellular potentials were scaled up by a factor of 10 and then added to the contributions of the inner cells. We performed reference simulations and LFP analog calculations with the inner radius set to 200 μm and 500 μm and obtained results identical with those in *Figures 3* and *4* (where an inner radius of 100 μm was used), except for negligible increases in the theta oscillation power found in the LFP analog spectrogram.

### Spike density function

We calculated the spike density function (SDF) of all pyramidal cell spikes using a Gaussian kernel with a window of 3 ms and a bin size of 1 ms (*Szucs, 1998*). To see how a cell's spiking activity is related to its SDF, see *Figure 4—figure supplement 1*.

### Oscillations

To quantify the frequency and power of the oscillations of the network, we computed a one-sided Welch's Periodogram of the SDF (*Colgin et al., 2009*) using a Hamming window with 50% overlap. To characterize the stability of the theta oscillation, we ran the control network for 4 s and then computed the spectrogram of the SDF and of the LFP analog using an analysis script from *Goutagny et al. (2009)* based on the mtspecgramc function from the Chronux toolbox (http://chronux.org/).

### Spike phases and theta modulation

We calculated the preferred firing theta phases of each cell, using all the spikes of that cell type that occurred after the first 50 ms of the simulation, relative to the filtered LFP analog. The spike times were converted to theta phases, relative to the troughs of the LFP analog theta cycle in which they fired. We then subjected the spike phases to a Rayleigh test to determine the level of theta modulation of the firing of each cell type (*Varga et al., 2014*).

### Firing rates

The firing rates of the cells were calculated by cropping the first 50 ms of the simulation to remove the initial effects, and then dividing the resulting number of spikes of each cell type by the total number of cells of that type and the duration of the simulation. An alternate average firing rate was calculated by dividing by the number of active cells of that type rather than all of the cells of that type, which gave the average firing rate over all firing cells instead, to better compare with experimentally observed firing rate averages.

### Statistical comparison of theta power

For the $GABA_B$-related simulations, we ran three of each condition and then performed an ANOVA to test for significance in the difference of theta power among the conditions.

### Cross correlation of theta and gamma

To investigate whether a relationship existed between the simultaneous theta and gamma oscillations found in the LFP analog of our control simulation, we filtered the LFP analog signal within the theta range (5–10 Hz) and the gamma range (25–80 Hz). We applied a Hilbert transform to each filtered signal and then compared the phase of the theta-filtered signal with the envelope of the gamma-filtered signal to determine the extent to which theta could modulate the gamma oscillation.

### Accessibility

Our model code is available online at ModelDB, entry #187604 (https://senselab.med.yale.edu/ModelDB/showModel.cshtml?model=187604; code version used to produce results in this work) and

Open Source Brain, project nc_ca1 (http://opensourcebrain.org/projects/nc_ca1; most recent code version). Open Source Brain provides tools for users to characterize and inspect model components. The model is also characterized online at http://mariannebezaire.com/models/ca1, along with a graphical explanation of our quantitative assessment used to constrain the model connectivity *Bezaire and Soltesz (2013)*, as well as links to our model code and model results, and detailed instruction manuals for our NEURON code and SimTracker tool, RRID:SCR_014735 (*Bezaire et al., 2016a*).

For those who wish to view and analyze our simulation results without rerunning the simulation, our simulation results are available on the Open Science Framework (RRID:SCR_003238) at http://doi.org/10.17605/OSF.IO/V4CEH (*Bezaire et al., 2016b*). Our analyses of these data can be recreated using our publicly available SimTracker tool.

SimTracker is freely available online at http://mariannebezaire.com/simtracker/ and is also listed in SimToolDB, entry #153281 at https://senselab.med.yale.edu/SimToolDB/showTool.cshtml?Tool= 153281. The tool is offered both as a stand-alone, compiled version for those without access to MATLAB (for Windows, Mac OS X, and Linux operating systems), and as a collection of MATLAB scripts for those with MATLAB access.

## Acknowledgements

In addition to the direct funding provided to the authors from NIH and NSF (listed separately), the authors wish to thank several researchers and programs operating under their own grants to make this work possible. Immeasurable support was provided by NEURON developers Michael Hines and Ted Carnevale under NIH NINDS grant R01-NS11613 (to MH) and NSF grant 1458495 (to TC). This work used the Extreme Science and Engineering Discovery Environment (XSEDE), which is supported by National Science Foundation grant number ACI-1053575; the project was supported by XSEDE Research and Startup Allocations to the authors (see funding information listed separately) and via the Neuroscience Gateway with the support of NSF grants 1458840 and 1146949 (to Majumdar et al.). The authors acknowledge the Texas Advanced Computing Center (TACC) at The University of Texas at Austin for providing high performance computing resources that have contributed to the research results reported within this paper (http://www.tacc.utexas.edu). Additionally, this research is part of the Blue Waters sustained-petascale computing project, which is supported by the National Science Foundation (awards OCI-0725070 and ACI-1238993) and the state of Illinois. Blue Waters is a joint effort of the University of Illinois at Urbana-Champaign and its National Center for Supercomputing Applications (NCSA). Parallel supercomputers used in this work include: Blue Waters, owned by the University of Illinois and NCSA; Stampede and the retired Ranger, owned by the University of Texas' Texas Advanced Computing Center (TACC); Trestles and Comet, owned by the San Diego Supercomputing Center; University of California at Irvine's High Performance Computer and the retired Broadcom Distributed Unified Cluster. We would like to thank the University of Texas' Texas Advanced Computing Center team, the San Diego Supercomputing Center and Neuroscience Gateway teams (especially Glenn Lockwood, Amitava Majumdar, Subhashini Sivagnanam, Mahidhar Tatineni, and Kenneth Yoshimoto), and UC Irvine's HPC team (especially Joseph Farran and Harry Mangalam) for their excellent technical support throughout this work. We would also like to thank Padraig Gleeson, Andras Ecker, Tom Morse, and Jeff Teeters for assistance making our code and model results public, and Jesse Jackson and Sylvain Williams for the use of their spectrogram analysis script.

## Additional information

### Funding

| Funder | Grant reference number | Author |
|---|---|---|
| National Institutes of Health | F32NS090753 | Marianne J Bezaire |
| National Science Foundation | DGE-0808392 | Marianne J Bezaire |
| National Science Foundation | XSEDE Allocations - TG-IBN140007 | Marianne J Bezaire Ivan Raikov |

| | | Ivan Soltesz |
|---|---|---|
| National Science Foundation | XSEDE Allocations - TG-IBN130022 | Marianne J Bezaire<br>Ivan Raikov<br>Ivan Soltesz |
| National Science Foundation | XSEDE Allocations - TG-IBN100011 | Marianne J Bezaire<br>Ivan Raikov |
| National Institutes of Health | NS35915 | Ivan Soltesz |
| National Science Foundation | IOS-1310378 | Ivan Soltesz |
| National Institutes of Health | NS090583 | Ivan Soltesz |

The funders had no role in study design, data collection and interpretation, or the decision to submit the work for publication.

### Author contributions

MJB, IR, Conception and design, Acquisition of data, Analysis and interpretation of data, Drafting or revising the article; KB, DV, Acquisition of data, Drafting or revising the article; IS, Conception and design, Analysis and interpretation of data, Drafting or revising the article

### Author ORCIDs

Marianne J Bezaire, http://orcid.org/0000-0001-6040-3520
Ivan Raikov, http://orcid.org/0000-0002-8224-8549

# Additional files

### Major datasets

The following datasets were generated:

| Author(s) | Year | Dataset title | Dataset URL | Database, license, and accessibility information |
|---|---|---|---|---|
| Bezaire M J, Raikov I, Burk K, Vyas D, Soltesz I | 2016 | Simulation results from a network model of the isolated hippocampal CA1 subfield in rat | http://dx.doi.org/10.17605/OSF.IO/V4CEH | Publicly available at the Open Science Framework |
| Bezaire M J, Raikov I | 2016 | Hippocampal CA1 network in parallel NEURON with spontaneous theta, gamma: full scale & network clamp | https://senselab.med.yale.edu/ModelDB/showmodel.cshtml?model=187604 | Publicly available at SenseLab (accession no: 187604) |

The following previously published datasets were used:

| Author(s) | Year | Dataset title | Dataset URL | Database, license, and accessibility information |
|---|---|---|---|---|
| Quattrocolo G, Maccaferri G | 2013 | Firing pattern of O-LM cells in mouse hippocampal CA1 | http://dx.doi.org/10.17605/OSF.IO/RA8MW | Publicly available at the Open Science Framework |
| Lee SH, Krook-Magnuson E, Soltesz I | 2016 | Intracellular, in vitro somatic membrane potential recordings from whole cell patch clamped rodent hippocampal CA1 neurons | http://dx.doi.org/10.17605/OSF.IO/M5EDM | Publicly available at the Open Science Framework |

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

**Appendix 1**

## Experimental cell characterization

These experimental cells were whole cell patch clamped to record the intracellular somatic membrane potential during a range of current injections ('current sweep'). The firing rates of each cell type are plotted in separate graphs, shown in **Appendix 1—figure 1**. The electrophysiological properties of each cell type are given in **Appendix 1—table 1**. In the Appendix, section 'Model cell characterization', the model cells are compared with this experimental data using the same calculations and properties.

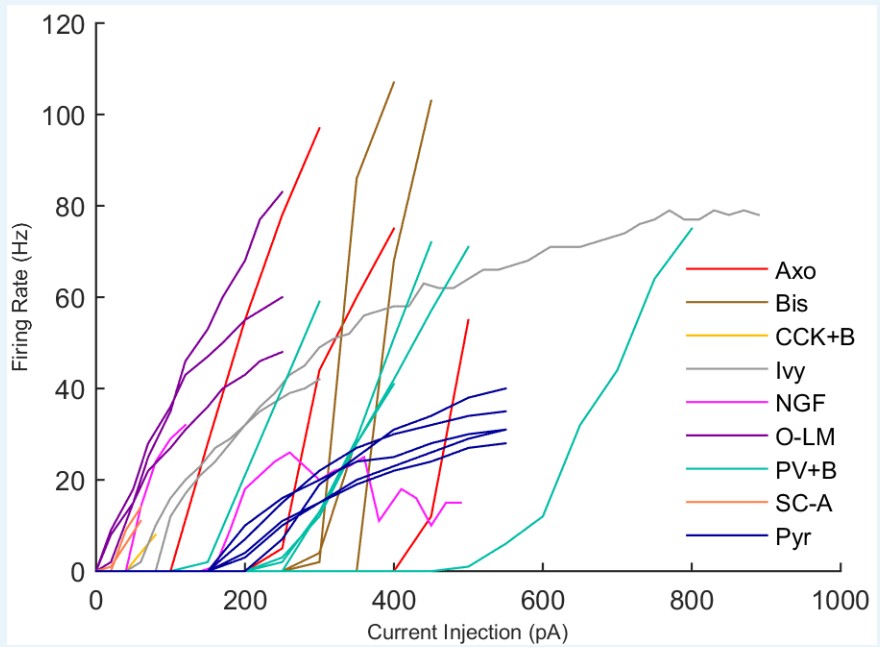

**Appendix 1—figure 1.** Firing Rates of Experimental Cells. Rebound spiking, which occurs in some O-LM cells at hyperpolarized current injection levels, is not shown in this graph.

The cell references and animals they came from (both rat and mouse (RRID:IMSR_JAX: 008069, RRID:IMSR_JAX:007905, RRID:IMSR_JAX:000664), species identified for each experimental cell) are provided here, as well as in two Open Science Framework entries online (O-LM cells: 10.17605/OSF.IO/RA8MW and other cells: 10.17605/OSF.IO/M5EDM) where the raw AxoClamp files of these experiments are also provided (**Lee et al., 2016**; **Quattrocolo and Maccaferri, 2016**). The tables of experimental conditions associated with the data entries in Open Science Framework are reproduced here for convenience, in **Appendix 1—table 2**.

The properties were calculated as follows:

### RMP

Resting membrane potential, in units of $mV$, is calculated as the average membrane potential during a current injection of 0 $pA$. If 0 was not part of the injection sweep, then the average membrane potential prior to the onset of a different current injection value is used.

**Appendix 1—table 1.** Intrinsic electrophysiological properties of experimental cells.

| Cell type | n | RMP (mV) | Input resistance (MΩ) | Sag amplitude (mV) | Sag tau (ms) | Membrane tau (ms) | Rheobase (pA) | ISI (ms) | Threshold (mV) | Spike amplitude (mV) | AHP (mV) |
|---|---|---|---|---|---|---|---|---|---|---|---|
| *From mouse* | | | | | | | | | | | |
| Pyr | 17 | −70.7 ± 1.2 | 139.5 ± 38.8 | 7.0 ± 2.2 | 34.4 ± 11.0 | 21.5 ± 8.6 | 182.4 ± 55.7 | 134.0 ± 44.0 | −36.7 ± 2.6 | 78.2 ± 7.2 | 8.6 ± 2.1 |
| Axo | 3 | −64.4 ± 4.5 | 122.0 ± 57.5 | 1.7 ± 0.6 | 45.4 ± 6.9 | 11.9 ± 2.2 | 283.3 ± 152.8 | 47.8 ± 28.5 | −31.8 ± 3.4 | 44.5 ± 6.7 | 16.6 ± 3.5 |
| Bis | 3 | −63.6 ± 4.7 | 109.1 ± 30.5 | 1.7 ± 0.6 | 62.3 ± 13.7 | 12.2 ± 0.6 | 333.3 ± 57.7 | 24.5 ± 21.8 | −31.9 ± 4.2 | 47.3 ± 6.8 | 22.6 ± 0.7 |
| O-LM | 3 | −64.8 ± 1.3 | 592.3 ± 97.0 | 10.4 ± 3.8 | 78.5 ± 22.0 | 41.4 ± 11.7 | 20.0 ± 0.0 | 101.9 ± 30.1 | −44.2 ± 2.3 | 76.3 ± 6.1 | 22.1 ± 4.7 |
| PV+B | 7 | −61.4 ± 2.0 | 65.2 ± 16.2 | 1.8 ± 0.5 | 62.9 ± 16.3 | 13.3 ± 5.4 | 307.1 ± 109.7 | 74.2 ± 36.4 | −35.3 ± 3.7 | 51.1 ± 9.0 | 18.0 ± 2.7 |
| *From rat* | | | | | | | | | | | |
| CCK+B | 1 | −61.2 | 298.1 | 2.7 | 72.1 | 56.0 | 60.0 | 261.0 | −37.7 | 63.7 | 15.5 |
| Ivy | 2 | −62.3 ± 0.3 | 267.2 ± 107.9 | 2.4 ± 2.5 | 91.1 ± 120.9 | 171.9 ± 45.6 | 80.0 ± 28.3 | 74.9 ± 20.6 | −32.8 ± 0.7 | 48.2 ± 5.1 | 20.1 ± 2.6 |
| NGF | 2 | −66.7 ± 13.4 | 260.0 ± 73.6 | 1.8 ± 1.6 | 61.7 ± 77.0 | 77.2 ± 66.2 | 110.0 ± 70.7 | 80.0 ± 28.4 | −34.0 ± 2.2 | 34.7 ± 4.9 | 16.2 ± 6.3 |
| SC-A | 2 | −57.0 ± 4.3 | 529.9 ± 2.9 | 7.9 ± 6.2 | 91.1 ± 21.7 | 74.2 ± 37.3 | 30.0 ± 14.1 | 132.4 ± 29.4 | −34.3 ± 2.2 | 58.7 ± 4.5 | 12.6 ± 2.0 |

## Input resistance

Units of MegaOhms ($M\Omega$), is the input resistance calculated from the least hyperpolarized current injection level.

## Sag amplitude

Units of $mV$, computed from the most hyperpolarized current injection level as the difference between the steady state membrane potential towards the end of the current injection and the most hyperpolarized potential achieved towards the beginning of the current injection.

## Sag tau

Units of $ms$, also computed from the most hyperpolarized current injection level, as the time constant required for an equation of the form $A * (1 - exp(-t/\tau))^4$ to best fit the potential trajectory from the most hyperpolarized point during the current injection until the trace reaches steady state.

## Membrane tau

Units of $ms$, computed from the least hyperpolarized current injection level, as the time constant required for an equation of the form $A * (exp(-t/\tau_m))$ to best fit the potential trajectory from the onset of the current injection until the trace reaches steady state.

## Rheobase

Units of $pA$, the least depolarized current injection level that resulted in the cell spiking during the current injection (i.e., not as a rebound spike after the injection ends, which can happen for certain cell types after a sufficiently hyperpolarized injection).

## ISI

Units of $ms$, the average time interval between spike threshold time points for the least depolarized current injection level where the cell spiked regularly.

## Threshold

Units of $mV$, the average threshold of the first three spikes for the least depolarized current injection level where the cell spiked regularly. For all experimental and model cells except for the experimental pyramidal cells, the threshold was calculated using CellData's method #2, where the threshold is the first point where $dV/dt$ exceeds some cutoff value (*Cooper et al., 2003*; *Metz et al., 2005*); in our case the cutoff was 28 $mV/ms$. Because calculating the threshold of the experimental pyramidal cells by this method resulted in a threshold point that was visually too depolarized given the shape of the action potential, CellData's method #1 was used instead, in which the threshold is the first point where $dV/dt > mean(dV/dt) + 2 * std(dV/dt)$, meaning the derivative of potential with time exceeds two standard deviations of the average (*Atherton and Bevan, 2005*).

## Spike amplitude

Units of $mV$, the difference between the membrane potential at the peak of the action potential and the membrane potential at the threshold of the action potential, averaged for the first three spikes of the least depolarized current injection where the cell spiked regularly.

## Slow AHP amplitude

Units of $mV$, also referred to simply as 'AHP' in the Appendix, the difference between the membrane potential at the most hyperpolarized potential following the action potential and the membrane potential at the threshold of the action potential, averaged for the first three spikes of the least depolarized current injection where the cell spiked regularly.

Further properties characterized from experimental cells (recorded and published by other labs) are available at NeuroElectro's website (http://neuroelectro.org), although the data included there are from a wide variety of conditions, animal types, and experimental protocols (and the calculations of properties may have been carried out differently).

**Appendix 1—table 2.** AxoClamp raw data files. Sch. Coll.-Assoc.: Schaffer Collateral-Associated; Super: superficial. Current sweep injection levels are reported as minimum (most hyperpolarized) : step size : maximum (depolarized) level in units of pA.

| Cell type | Lab | Cell name | Species | Current inj. Levels (pA) | Original use and methods reference |
|---|---|---|---|---|---|
| Axo-axonic | Soltesz | CA203LF57 | mouse | −200:50: +500 | unpublished |
| Axo-axonic | Soltesz | CA204LF59 | mouse | −200:50: +300 | unpublished |
| Axo-axonic | Soltesz | CA204RF59 | mouse | −200:50: +400 | unpublished |
| Bistratified | Soltesz | PV16IM | mouse | −300:50: +400 | unpublished |
| Bistratified | Soltesz | PV74 | mouse | −300:50: +350 | unpublished |
| Bistratified | Soltesz | PV27IM | mouse | −300:50: +450 | unpublished |
| PV+ Basket | Soltesz | PV34 | mouse | −300:50: +500 | *Lee et al. (2014)* |
| PV+ Basket | Soltesz | PV36 | mouse | −300:50: +800 | *Lee et al. (2014)* |
| PV+ Basket | Soltesz | PV37 | mouse | −300:50: +500 | *Lee et al. (2014)* |
| PV+ Basket | Soltesz | PV38 | mouse | −300:50: +300 | *Lee et al. (2014)* |
| PV+ Basket | Soltesz | PV72 | mouse | −300:50: +400 | *Lee et al. (2014)* |
| PV+ Basket | Soltesz | PV80 | mouse | −300:50: +450 | *Lee et al. (2014)* |
| Deep Pyramidal | Soltesz | D1_25abf | mouse | −400:50: +550 | *Lee et al. (2014)* |
| Deep Pyramidal | Soltesz | D1_45abf | mouse | −400:50: +550 | *Lee et al. (2014)* |
| Deep Pyramidal | Soltesz | D2_06abf | mouse | −400:50: +550 | *Lee et al. (2014)* |
| Deep Pyramidal | Soltesz | D2_49abf | mouse | −400:50: +550 | *Lee et al. (2014)* |
| Deep Pyramidal | Soltesz | D3_55abf | mouse | −400:50: +550 | *Lee et al. (2014)* |
| Deep Pyramidal | Soltesz | D4_11abf | mouse | −400:50: +550 | *Lee et al. (2014)* |
| Deep Pyramidal | Soltesz | D5_15abf | mouse | −400:50: +550 | *Lee et al. (2014)* |

*Appendix 1—table 2 continued on next page*

Appendix 1—table 2 continued

| Cell type | Lab | Cell name | Species | Current inj. Levels (pA) | Original use and methods reference |
|-----------|-----|-----------|---------|-----------|-------------------------------------|
| Deep Pyramidal | Soltesz | D6_19abf | mouse | −400:50:+550 | *Lee et al. (2014)* |
| Deep Pyramidal | Soltesz | D7 | mouse | −400:50:+550 | *Lee et al. (2014)* |
| Super. Pyramidal | Soltesz | S1_04abf | mouse | −400:50:+550 | *Lee et al. (2014)* |
| Super. Pyramidal | Soltesz | S1_47abf | mouse | −400:50:+550 | *Lee et al. (2014)* |
| Super. Pyramidal | Soltesz | S2_08abf | mouse | −400:50:+550 | *Lee et al. (2014)* |
| Super. Pyramidal | Soltesz | S2_31abf | mouse | −400:50:+550 | *Lee et al. (2014)* |
| Super. Pyramidal | Soltesz | S2_51abf | mouse | −400:50:+550 | *Lee et al. (2014)* |
| Super. Pyramidal | Soltesz | S3_13abf | mouse | −400:50:+550 | *Lee et al. (2014)* |
| Super. Pyramidal | Soltesz | S4 | mouse | −400:50:+550 | *Lee et al. (2014)* |
| Super. Pyramidal | Soltesz | S5_21abf | mouse | −400:50:+550 | *Lee et al. (2014)* |
| Ivy | Soltesz | 0422–1 (File 5) | rat | −100:20:+890 | *Krook-Magnuson et al. (2011)* |
| Ivy | Soltesz | 0428–1 (File 4) | rat | −100:20:+300 | *Krook-Magnuson et al. (2011)* |
| Neurogliaform | Soltesz | 09o21 (File 4) | rat | −100:20:+120 | *Krook-Magnuson et al. (2011)* |
| Neurogliaform | Soltesz | 09o27 (File 7) | rat | −100:20:+490 | *Krook-Magnuson et al. (2011)* |
| CCK+ Basket | Soltesz | sh108_BC | rat | −100:20:+80 | *Lee et al. (2010)* |
| Sch. Coll.-Assoc. | Soltesz | sh114_SCA | rat | −100:20:+60 | *Lee et al. (2010)* |
| Sch. Coll.-Assoc. | Soltesz | sh153_SCA | rat | −100:20:+60 | *Lee et al. (2010)* |
| O-LM | Maccaferri | 1May2012_P3 | mouse | −100:30:+250 | *Quattrocolo and Maccaferri (2013)* |
| O-LM | Maccaferri | 20Sept2011_P2 | mouse | −100:30:+250 | *Quattrocolo and Maccaferri (2013)* |
| O-LM | Maccaferri | 24October2012_C2 | mouse | −100:30:+250 | *Quattrocolo and Maccaferri (2013)* |

## Model cell characterization

Model cell numbers and structural connectivity are based on *Bezaire and Soltesz (2013)*.

In terms of electrophysiology, each model cell is characterized in experimental terms and compared to the experimental data presented above. A graphical summary of electrophysiological comparison is shown below in *Appendix 1—figures 2–4*, and further details of intrinsic physiology and synaptic characterization follows. The detailed information is presented as a single figure spanning two pages per cell, where the

subfigure panels may contain figures or tables, to better group and arrange the data. For each cell type, the same information is provided:

### A: Model current sweep

The somatic intracellular membrane potential recording for the model cell is shown, in response to all hyperpolarized and the most depolarized current injection

### B: Experimental current sweep

The somatic intracellular membrane potential recording for an experimental cell (one of the ones featured in the previous section) is shown, in response to all hyperpolarized and the depolarized current injection level closest to the one shown for the model cell.

### C: Model electrophysiological property table

The values of each electrophysiological property are shown for the model cell, measured or calculated in the same way as for the experimental cells in the previous section. All experimental data are taken from the cells used in section 'Experimental cell characterization' of the Appendix. More experimental data, obtained under a wider variety of conditions, are available at http://neuroelectro.org.

### D: Firing rates

The firing rate of the model cell as a function of current injection is shown for a range of currents. The firing rates of the experimental cells are shown as well (to identify specific experimental cells, see the firing rate graphs in section 'Experimental cell characterization' of the Appendix).

### E: Ion channel table

Each ion channel type present in the model cell type is listed here, along with the maximum conductance density of that channel (which may occur in the soma or in another part of the cell). Further details about the ion channels are available in the Appendix, sections 'Ion channel descriptions' and 'Ion channel equations'.

### F: Structural connectivity table

The structural basis of the model connections is provided here in terms of both connections (comprising multiple synapses) and synapses, shown for convergence onto the cell (left side) and divergence emanating from the cell (right side). Connectivity involving pyramidal cells (either pre- or postsynaptically) is based on *Bezaire and Soltesz (2013)* while connectivity between interneurons is detailed in section 'Inhibitory connectivity' in the Appendix. Blank or missing rows for specific cell types indicate that there were no connections with that cell type.

### G,H: Model paired recordings – experimental conditions table

For incoming (G) and outgoing (H) connections involving a given cell type, any experimental constraints used to fit the model connections are cited here, including the conditions of the experiment that the model reproduced (the holding potential of the cell and the reversal potential of the synapse as set by the bath and pipette solutions used in the experiment), along with the model connection characterization (amplitude, 10–90% rise time, and decay time constant) and the percentage difference from the mean experimental properties. The model results were reported as the average from 10 random connections between the two

cell types, and wherever possible were compared to an actual experimental mean (leaving out failed responses) rather than an effective experimental mean that factored in the synapse failure rate. For connections that were characterized in current clamp rather than voltage clamp, the data were colored in purple rather than black to indicate their different properties (half-width instead of decay time constant, for example). Because the experimental data did not routinely report or, in a standard way, calculate the junction potential of the experiment, we did not factor in the difference in holding potential (and hence driving force of the synapse) due to junction potential; this resulted in a slight over or underestimation of the driving force and hence the actual synaptic conductance, depending on the junction potential value.

### I: Model synapse parameters table

The model parameters used in the synapses (resulting from the experimental tuning above, or firing rate tuning if no experimental connection data were available) are listed here, with the parameters for connections onto that cell type from the other types listed on the left side, and the parameters for connections onto other cell types, from that cell type, listed on the right side.

### J: Model physiological connections table

All connections in the model were recharacterized under the same condition, as experimental data were gathered under diverse conditions where different connections could not be directly compared. In this table, all connections were recorded using the physiological reversal potential while holding the postsynaptic cell at −50 mV voltage clamp (and not accounting for any junction potential as would occur if an experiment were to replicate these model simulation conditions). The connections from other cell types to the given cell type are shown on the left side; connections from the given cell type to other cell types are shown on the right side. Blank lines or missing rows indicate there is not a connection between those cell types in the model.

### K, L: Model physiological connections

The postsynaptic current (PSC) responses for all the physiological connections as detailed in Table J above are graphed here, with connections from other cell types to the given cell type shown in panel K, and connections from the given cell type to other cells shown in panel L.

The model cell types can be further characterized as desired by using the artificially generated AxoClamp files included in the Source Data for Figure 2. These AxoClamp files were generated from our CellClamp tool within SimTracker, providing the same data format (but with generic header lines for the first ten lines), as the tab-delimited ATF file format.

Model and experimental electrophysiology

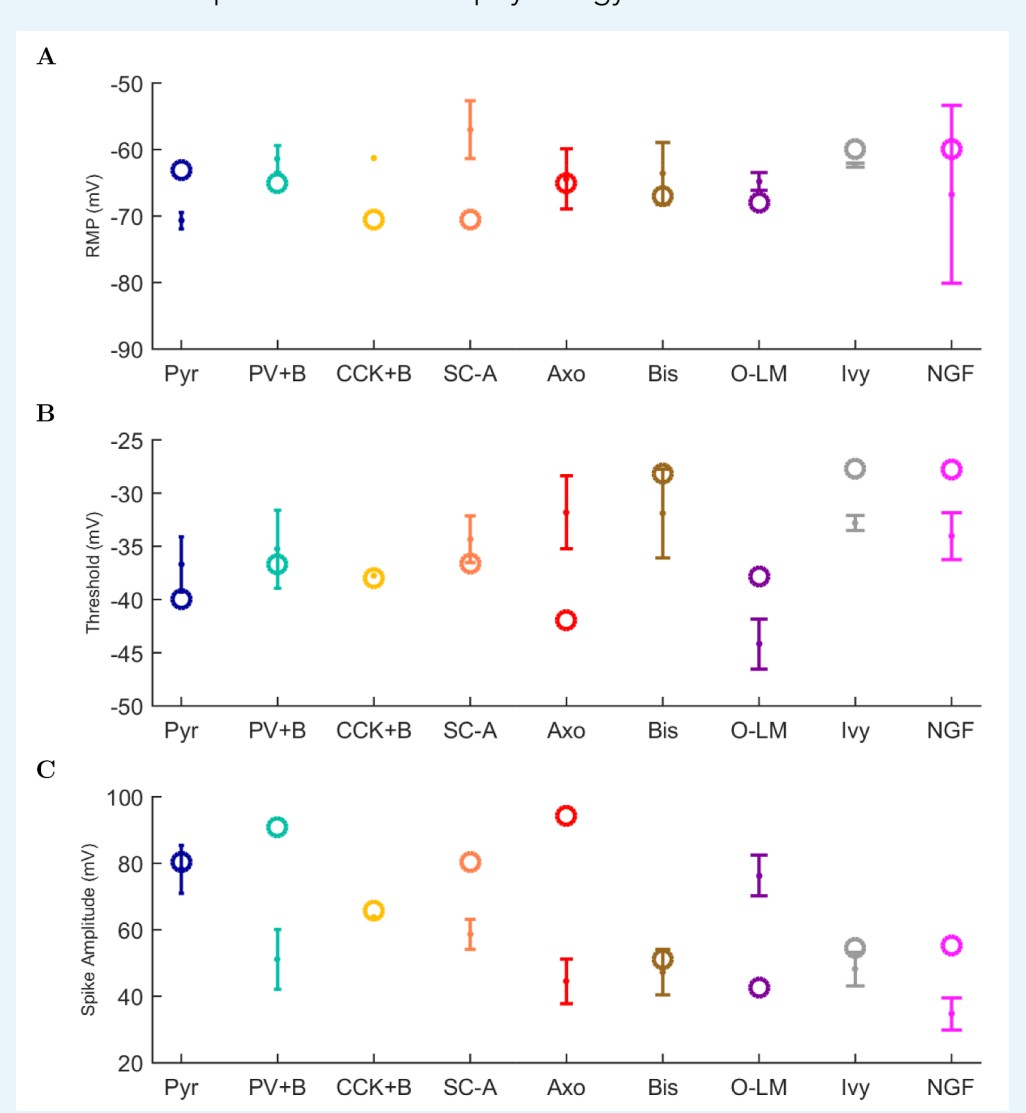

**Appendix 1—figure 2.** Physiological properties of experimental and model cells. Experimental data are shown with closed markers for the mean and error bars for cell types where n > 1. The model cell properties are plotted as open circles. Calculation of properties is explained in the text. (**A**) resting membrane potential, (**B**) threshold, and (**C**) spike amplitude.

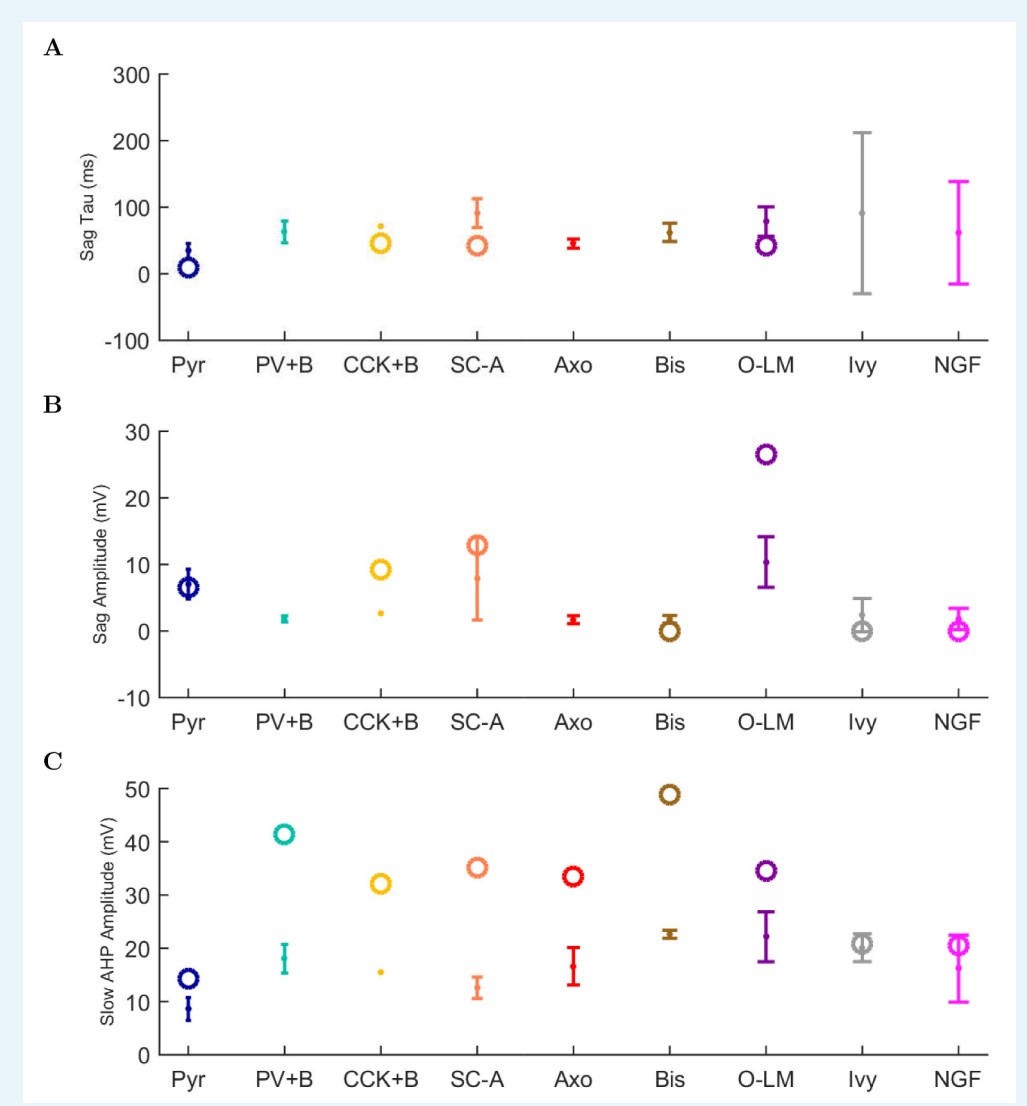

**Appendix 1—figure 3.** Physiological properties, continued. (**A**) sag time constant, (**B**) sag amplitude, and (**C**) amplitude of afterhyperpolarization (AHP).

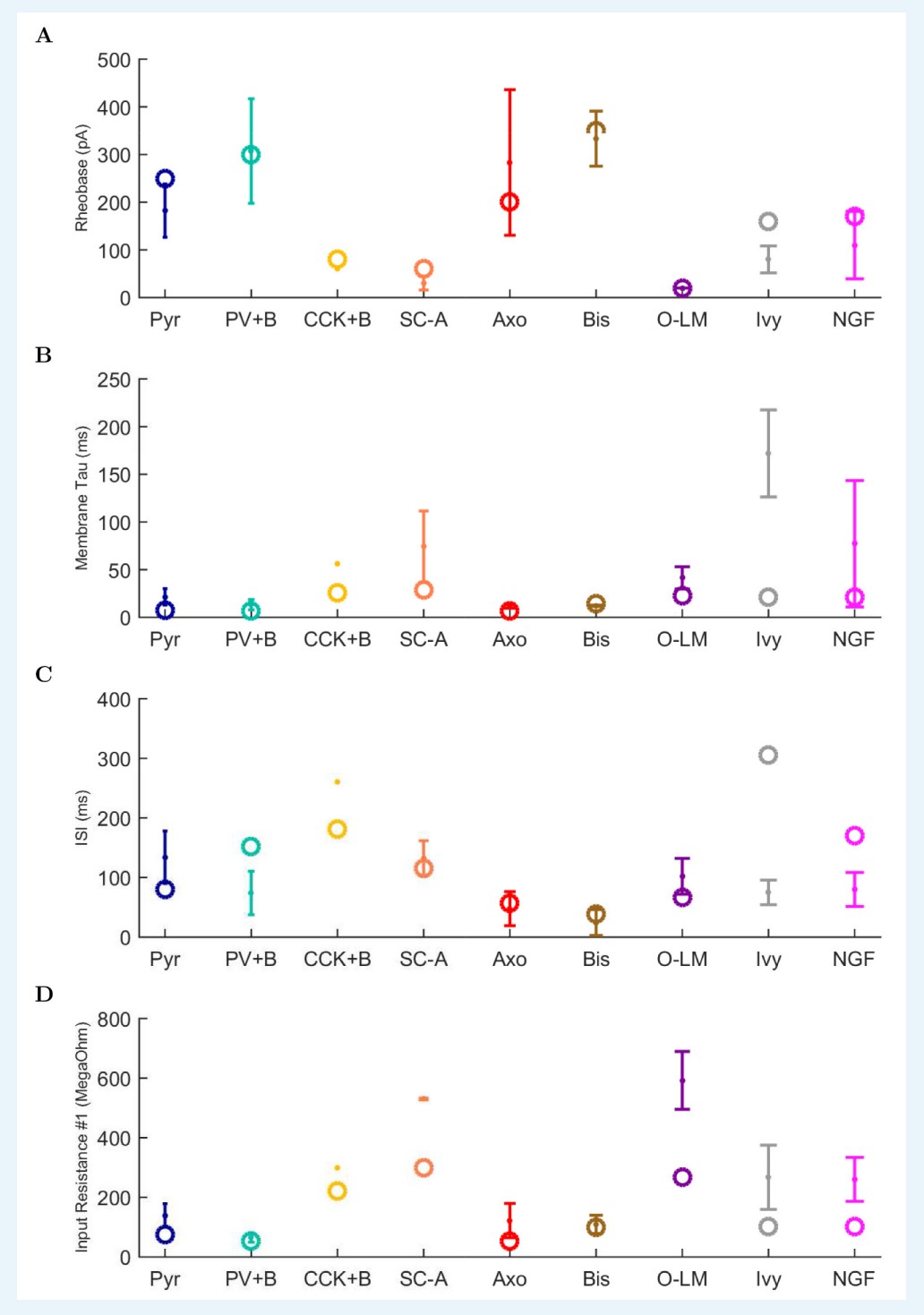

**Appendix 1—figure 4.** Physiological properties, continued. (**A**) rheobase, (**B**) membrane time constant, (**C**) interspike interval (ISI), and (**D**) input resistance.

## Pyramidal cell: principal cell (311500 Cells)

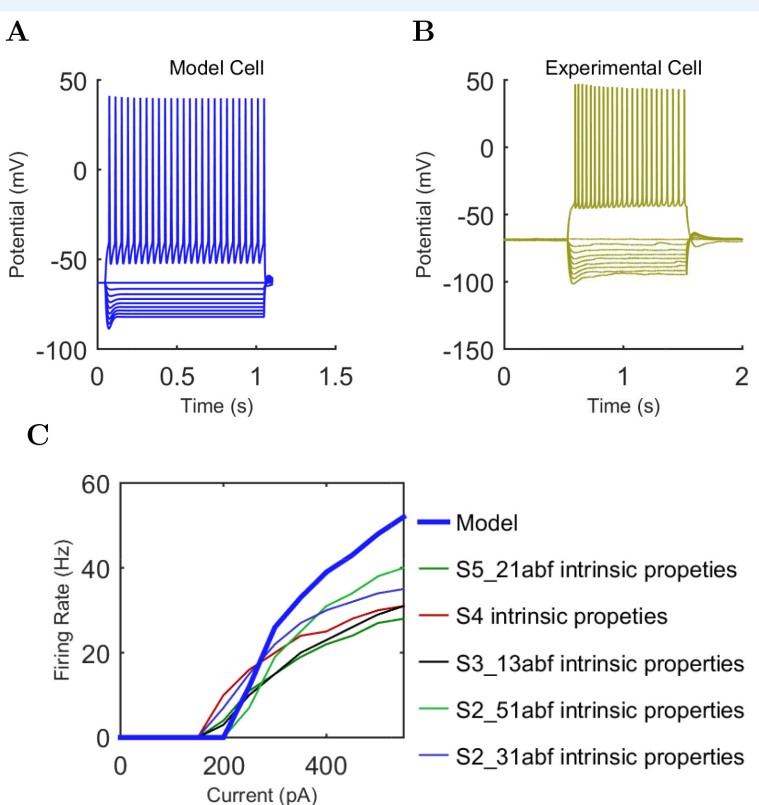

**Appendix 1—figure 5.** Pyramidal (**A**) model and (**B**) experimental current sweep. (**C**) Firing rates of model and experimental cells.

**Appendix 1—table 3.** Model Pyramidal cell electrophysiological properties.

| Property | Value |
|---|---|
| RMP | −63.0 mV |
| Input Resistance | 76.1 MΩ |
| Sag Amplitude | 6.5 mV |
| Sag Tau | 9.6 ms |
| Membrane Tau | 7.1 ms |
| Rheobase | 250.0 pA |
| ISI | 80.7 ms |
| Threshold | −39.9 mV |
| Spike Amplitude | 80.3 mV |
| Slow AHP Amplitude | 14.3 mV |

**Appendix 1—table 4.** Model Pyramidal cell ion channels and conductance at highest density location in cell.

| Channel | Highest conductance $G_{max}$ (S/cm$^2$) |
|---|---|
| HCNp | 4.968e-03 |
| Kdrp | 3.000e-03 |
| KvAdistp | 4.682e-02 |

*Appendix 1—table 4 continued on next page*

Appendix 1—table 4 continued

| Channel | Highest conductance $G_{max}$ (S/cm$^2$) |
|---|---|
| KvAproxp | 1.599e-02 |
| Navaxonp | 6.400e-02 |
| Navp | 3.200e-02 |

## Model and experimental connectivity

**Appendix 1—table 5.** Structural connection parameters for Pyramidal cells, based on *Bezaire and Soltesz (2013)*.

| | Other cell to pyr | | | | Pyr to other cell | | | |
|---|---|---|---|---|---|---|---|---|
| Other type | # Conn.s | Syn.s /Conn. | # # | Post Loc. | # Conn.s | Syn.s /Conn. | # # | Post Loc. |
| Axo | 6 | 6 | 36 | axon | 1 | 3 | 2 | apical dendrite |
| Bis | 10 | 10 | 100 | any dendrite | 3 | 3 | 7 | apical dendrite |
| CCK+B | 13 | 8 | 104 | any dendrite | | | | |
| Ivy | 42 | 10 | 420 | any dendrite | 0 | 3 | 0 | apical dendrite |
| NGF | 14 | 10 | 140 | apical dendrite | | | | |
| O-LM | 8 | 10 | 80 | apical dendrite | 13 | 3 | 37 | basal dendrite |
| Pyr | 197 | 1 | 197 | apical dendrite | 197 | 1 | 197 | apical dendrite |
| PV+B | 17 | 11 | 187 | soma | 8 | 3 | 22 | apical dendrite |
| SC-A | | | | | 0 | 3 | 0 | apical dendrite |
| CA3 | 5985 | 2 | 11970 | any dendrite | | | | |
| ECIII | 1299 | 2 | 2598 | any dendrite | | | | |

## Experimental connection constraints

**Appendix 1—table 6.** Experimental constraints for incoming connections onto Pyramidal cells (clamp: black=voltage; purple=current).

| Pre type | Exp. ref. | Hold (mV) | $E_{rev}$ (mV) | Amp. (pA,mV) | Diff. % | $t_{10-90}$ (ms) | Diff. % | $\tau_{decay}$ (ms) | Diff. % |
|---|---|---|---|---|---|---|---|---|---|
| Axo | *Maccaferri et al., 2000* | −70.0 | 7.0 | 323.78 | +5.1 | 0.83 | +3.1 | 11.20 | +0.0 |
| Bis | *Maccaferri et al., 2000* | −70.0 | 7.0 | 143.21 | −4.5 | 2.22 | +11.2 | 15.40 | −4.3 |
| CCK +B | *Lee et al., 2010* | −70.0 | −26.0 | 118.97 | +3.1 | 0.53 | −16.7 | 6.15 | −4.9 |
| Ivy | *Fuentealba et al., 2008* | −50.0 | −88.0 | 8.17 | +2.1 | 3.50 | +25.0 | 15.43 | −3.9 |
| NGF | *Price et al., 2008* | −50.0 | −89.0 | 5.25 | +7.1 | 15.48 | −3.9 | 32.73 | −34.5 |
| O-LM | *Maccaferri et al., 2000* | −70.0 | 7.0 | 24.35 | −6.3 | 4.68 | −24.6 | 18.88 | −9.3 |
| Pyr | *Deuchars and Thomson, 1996* | −67.0 | 0.0 | 0.60 | −14.5 | 6.00 | +122.2 | 20.55 | +22.3 |
| PV+B | *Szabadics et al., 2007* | −70.0 | −26.0 | 91.94 | −13.9 | 0.50 | −5.7 | 6.70 | +4.7 |
| SC-A | *Lee et al., 2010* | −70.0 | −26.0 | 52.42 | −12.9 | 1.63 | +13.6 | 8.55 | +3.0 |

**Appendix 1—table 7.** Experimental constraints for outgoing connections from Pyramidal cells (clamp: black=voltage; purple=current).

| Post type | Exp. ref. | Hold (mV) | $E_{rev}$ (mV) | Amp. (pA,mV) | Diff. % | $t_{10-90}$ (ms) | Diff. % | $\tau_{decay}$ (ms) | Diff. % |
|---|---|---|---|---|---|---|---|---|---|
| Bis | *Pawelzik et al., 2002* | −66.0 | 0.0 | 0.77 | −19.6 | 1.58 | +31.3 | 16.75 | +41.1 |
| Ivy | *Fuentealba et al., 2008* | −65.8 | −70.0 | 0.06 | −97.9 | 1.38 | −8.3 | 21.35 | +41.1 |
| Pyr | *Deuchars and Thomson, 1996* | −67.0 | 0.0 | 0.60 | −14.5 | 6.00 | +122.2 | 19.05 | +22.3 |
| PV+B | *Lee et al., 2014* | −60.0 | 0.0 | 15.09 | −67.7 | 0.28 | −72.5 | 2.00 | +22.3 |

## Model synapse parameters

**Appendix 1—table 8.** Model synaptic parameters for Pyramidal cells in the control network.

| Type | Other cell to pyr | | | | Pyr to other cell | | | |
|---|---|---|---|---|---|---|---|---|
| | $E_{rev}$ (mV) | $G_{max}$ (nS) | $\tau_{rise}$ (ms) | $\tau_{decay}$ (ms) | $E_{rev}$ (mV) | $G_{max}$ (nS) | $\tau_{rise}$ (ms) | $\tau_{decay}$ (ms) |
| Axo | −60.0 | 1.150e-03 | 0.28 | 8.40 | 0.0 | 4.000e-05 | 0.30 | 0.60 |
| Bis | −60.0 | 5.100e-04 | 0.11 | 9.70 | 0.0 | 1.900e-03 | 0.11 | 0.25 |
| CCK+B | −60.0 | 5.200e-04 | 0.20 | 4.20 | | | | |
| Ivy | −60.0 | 4.100e-05 | 1.10 | 11.00 | 0.0 | 4.050e-04 | 0.30 | 0.60 |
| NGF | −60.0 | 6.500e-05 | 9.00 | 39.00 | | | | |
| O-LM | −60.0 | 3.000e-04 | 0.13 | 11.00 | 0.0 | 2.000e-04 | 0.30 | 0.60 |
| Pyr | 0.0 | 7.000e-02 | 0.10 | 1.50 | 0.0 | 7.000e-02 | 0.10 | 1.50 |
| PV+B | −60.0 | 2.000e-04 | 0.30 | 6.20 | 0.0 | 7.000e-04 | 0.07 | 0.20 |
| SC-A | | | | | 0.0 | 4.050e-04 | 0.30 | 0.60 |
| CA3 | 0.0 | 2.000e-04 | 0.50 | 3.00 | | | | |
| ECIII | 0.0 | 2.000e-04 | 0.50 | 3.00 | | | | |

## Physiological characterization of model connections

**Appendix 1—table 9.** Model synaptic properties under voltage clamp at −50 mV with physiological reversal potentials

| Type | Other cell to pyr | | | | | Pyr to other cell | | | | |
|---|---|---|---|---|---|---|---|---|---|---|
| | Hold (mV) | $E_{rev}$ (mV) | Amp. (pA) | $t_{10-90}$ (ms) | $\tau_{decay}$ (ms) | Hold (mV) | $E_{rev}$ (mV) | Amp. (pA) | $t_{10-90}$ (ms) | $\tau_{decay}$ (ms) |
| Axo | −50.0 | −60.0 | 36.45 | 0.85 | 11.57 | −50.0 | 0.0 | 1.85 | 0.78 | 2.53 |
| Bis | −50.0 | −60.0 | 13.47 | 2.17 | 15.20 | −50.0 | 0.0 | 64.48 | 0.28 | 1.42 |
| CCK +B | −50.0 | −60.0 | 24.86 | 0.52 | 6.03 | | | | | |
| Ivy | −50.0 | −60.0 | 1.63 | 3.63 | 15.35 | −50.0 | 0.0 | 40.70 | 0.58 | 1.28 |
| NGF | −50.0 | −60.0 | 1.10 | 65.58 | 0.00 | | | | | |
| O-LM | −50.0 | −60.0 | 0.54 | 3.70 | 14.10 | −50.0 | 0.0 | 17.47 | 0.60 | 1.53 |
| Pyr | −50.0 | 0.0 | 22.13 | 2.22 | 9.65 | −50.0 | 0.0 | 22.13 | 2.22 | 9.65 |
| PV +B | −50.0 | −60.0 | 20.56 | 0.50 | 6.70 | −50.0 | 0.0 | 14.75 | 0.25 | 1.77 |
| SC-A | | | | | | −50.0 | 0.0 | 17.42 | 0.68 | 3.05 |

*Appendix 1—table 9 continued on next page*

*Appendix 1—table 9 continued*

| | Other cell to pyr | | | | | Pyr to other cell | | | | |
| Type | Hold (mV) | $E_{rev}$ (mV) | Amp. (pA) | $t_{10-90}$ (ms) | $\tau_{decay}$ (ms) | Hold (mV) | $E_{rev}$ (mV) | Amp. (pA) | $t_{10-90}$ (ms) | $\tau_{decay}$ (ms) |
|---|---|---|---|---|---|---|---|---|---|---|
| CA3 | −50.0 | 0.0 | 7.15 | 1.83 | 7.08 | | | | | |
| ECIII | −50.0 | 0.0 | 1.41 | 3.25 | 13.63 | | | | | |

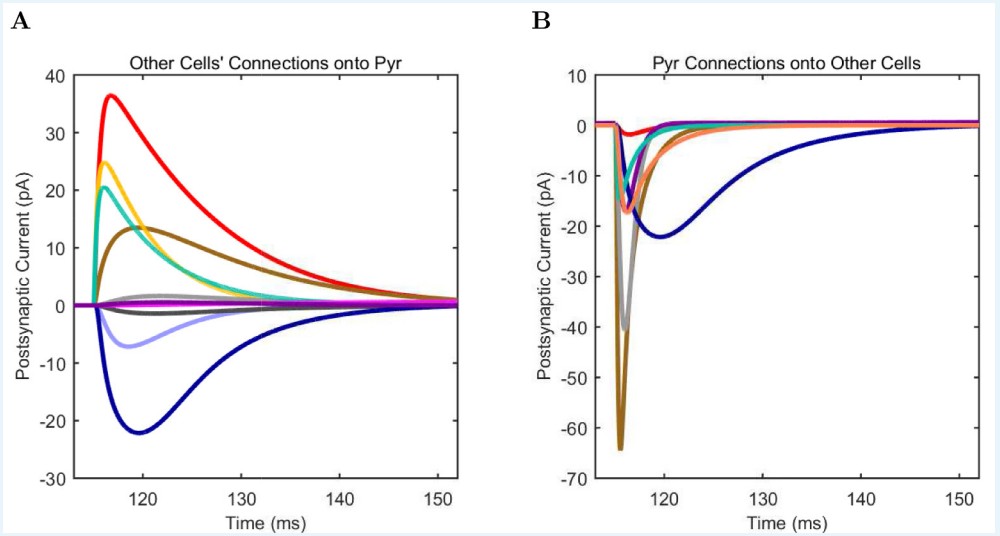

**Appendix 1—figure 6.** Connections onto (**A**) and (**B**) from model Pyramidal cells, under voltage clamp at −50 mV with physiological reversal potentials.

## Axo-axonic cell: fast-spiking axonic inhibitor (1470 Cells)

## Model and experimental electrophysiology

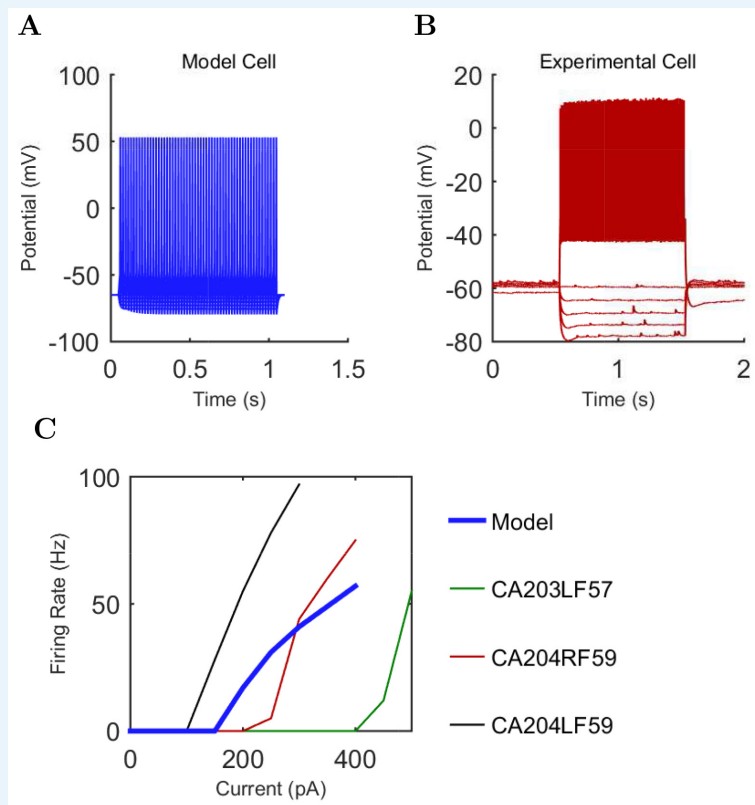

**Appendix 1—figure 7.** Axo-axonic (**A**) model and (**B**) experimental current sweep. (**C**) Firing rates of model and experimental cells.

**Appendix 1—table 10.** Model Axo-axonic cell electrophysiological properties.

| Property | Value |
| --- | --- |
| RMP | −65.0 mV |
| Input Resistance | 52.3 MΩ |
| Sag Amplitude | − |
| Sag Tau | − |
| Membrane Tau | 7.0 ms |
| Rheobase | 200.0 pA |
| ISI | 57.3 ms |
| Threshold | −42.0 mV |
| Spike Amplitude | 94.3 mV |
| Slow AHP Amplitude | 33.4 mV |

**Appendix 1—table 11.** Model Axo-axonic cell ion channels and conductance at highest density location in cell.

| Channel | $G_{max}$ (S/cm$^2$) |
| --- | --- |
| CavL | 5.000e-03 |

*Appendix 1—table 11 continued on next page*

*Appendix 1—table 11 continued*

| Channel | G$_{max}$ (S/cm$^2$) |
|---|---|
| CavN | 8.000e-04 |
| KCaS | 2.000e-06 |
| Kdrfast | 1.300e-02 |
| KvA | 1.500e-04 |
| KvCaB | 2.000e-07 |
| Nav | 1.500e-01 |
| leak | 1.800e-04 |

## Model and experimental connectivity

**Appendix 1—table 12.** Structural connection parameters for Axo-axonic cells, based on *Bezaire and Soltesz (2013)*.

| | Other cell to axo | | | | Axo to other cell | | | |
|---|---|---|---|---|---|---|---|---|
| | # | Syn.s | # | Post | # | Syn.s | # | Post |
| Other type | Conn.s | /Conn. | # | Loc. | Conn.s | /Conn. | # | Loc. |
| Bis | 16 | 10 | 160 | any dendrite | | | | |
| CCK+B | 12 | 8 | 96 | any dendrite | | | | |
| Ivy | 24 | 10 | 240 | any dendrite | | | | |
| O-LM | 8 | 10 | 80 | apical dendrite | | | | |
| Pyr | 162 | 3 | 486 | apical dendrite | 1271 | 6 | 7628 | axon |
| PV+B | 39 | 1 | 39 | soma | | | | |
| SC-A | 1 | 6 | 6 | any dendrite | | | | |
| CA3 | 4170 | 2 | 8340 | any dendrite | | | | |
| ECIII | 485 | 2 | 970 | any dendrite | | | | |

## Experimental connection constraints

Note:No experimental constraints available for incoming synapses to Axo-axonic cells.

**Appendix 1—table 13.** Experimental constraints for outgoing connections from Axo-axonic cells (clamp: black=voltage; purple=current).

| Post type | Exp. ref. | Hold (mV) | E$_{rev}$ (mV) | Amp. (pA, mV) | Diff. % | t$_{10-90}$ (ms) | Diff. % | τ$_{decay}$ (ms) | Diff. % |
|---|---|---|---|---|---|---|---|---|---|
| Pyr | *Maccaferri et al., 2000* | −70.0 | 7.0 | 323.78 | +5.1 | 0.83 | +3.1 | 11.20 | +0.0 |

## Model synapse parameters

**Appendix 1—table 14.** Model synaptic parameters for Axo-axonic cells in the control network.

| | Other cell to axo | | | | Axo to other cell | | | |
|---|---|---|---|---|---|---|---|---|
| Type | E$_{rev}$ (mV) | G$_{max}$ (nS) | τ$_{rise}$ (ms) | τ$_{decay}$ (ms) | E$_{rev}$ (mV) | G$_{max}$ (nS) | τ$_{rise}$ (ms) | τ$_{decay}$ (ms) |
| Bis | −60.0 | 6.000e-04 | 0.29 | 2.67 | | | | |
| CCK+B | −60.0 | 7.000e-04 | 0.43 | 4.49 | | | | |
| Ivy | −60.0 | 5.700e-05 | 2.90 | 3.10 | | | | |

*Appendix 1—table 14 continued on next page*

Appendix 1—table 14 continued

| Type | Other cell to axo | | | | Axo to other cell | | | |
|---|---|---|---|---|---|---|---|---|
| | $E_{rev}$ (mV) | $G_{max}$ (nS) | $\tau_{rise}$ (ms) | $\tau_{decay}$ (ms) | $E_{rev}$ (mV) | $G_{max}$ (nS) | $\tau_{rise}$ (ms) | $\tau_{decay}$ (ms) |
| O-LM | −60.0 | 1.200e-04 | 0.73 | 10.00 | | | | |
| Pyr | 0.0 | 4.000e-05 | 0.30 | 0.60 | −60.0 | 1.150e-03 | 0.28 | 8.40 |
| PV+B | −60.0 | 1.200e-04 | 0.29 | 2.67 | | | | |
| SC-A | −60.0 | 6.000e-04 | 0.42 | 4.99 | | | | |
| CA3 | 0.0 | 1.200e-04 | 2.00 | 6.30 | | | | |
| ECIII | 0.0 | 1.200e-04 | 2.00 | 6.30 | | | | |

## Physiological characterization of model connections

**Appendix 1—table 15.** Model synaptic properties under voltage clamp at −50 mV with physiological reversal potentials

| Type | Other cell to axo | | | | | Axo to other cell | | | | |
|---|---|---|---|---|---|---|---|---|---|---|
| | Hold (mV) | $E_{rev}$ (mV) | Amp. (pA) | $t_{10-90}$ (ms) | $\tau_{decay}$ (ms) | Hold (mV) | $E_{rev}$ (mV) | Amp. (pA) | $t_{10-90}$ (ms) | $\tau_{decay}$ (ms) |
| Bis | −50.0 | −60.0 | 36.77 | 0.70 | 3.70 | | | | | |
| CCK +B | −50.0 | −60.0 | 47.29 | 0.75 | 5.27 | | | | | |
| Ivy | −50.0 | −60.0 | 4.34 | 2.13 | 6.57 | | | | | |
| O-LM | −50.0 | −60.0 | 4.76 | 2.55 | 12.03 | | | | | |
| Pyr | −50.0 | 0.0 | 1.85 | 0.78 | 2.53 | −50.0 | −60.0 | 36.45 | 0.85 | 11.57 |
| PV+B | −50.0 | −60.0 | 1.08 | 0.45 | 3.13 | | | | | |
| SC-A | −50.0 | −60.0 | 24.00 | 1.00 | 6.13 | | | | | |
| CA3 | −50.0 | 0.0 | 10.85 | 2.30 | 8.80 | | | | | |
| ECIII | −50.0 | 0.0 | 8.74 | 3.08 | 9.20 | | | | | |

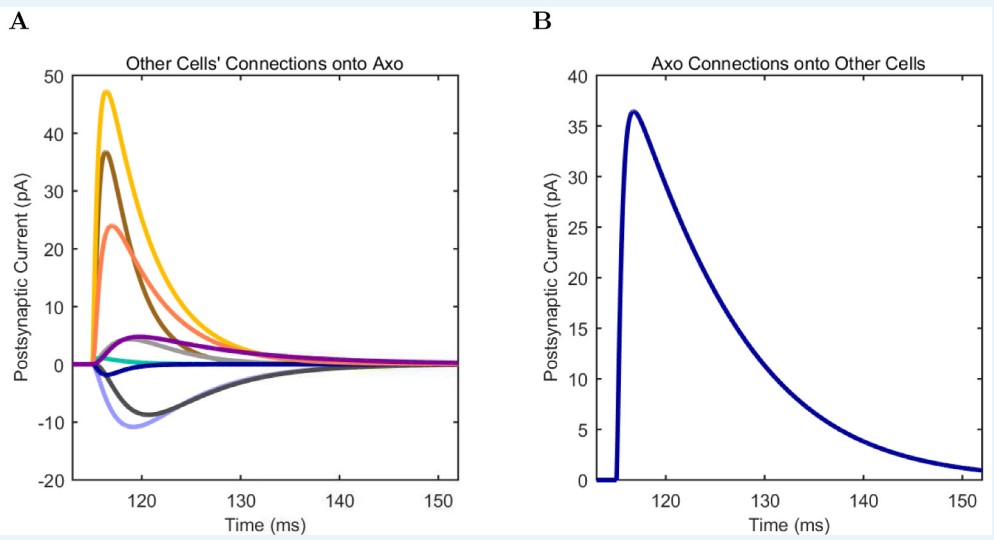

**Appendix 1—figure 8.** Connections onto (**A**) and (**B**) from model Axo-axonic cells, under

voltage clamp at −50 mV with physiological reversal potentials.

## Bistratified cell: fast-spiking dendritic inhibitor (2210 Cells)

## Model and Experimental Electrophysiology

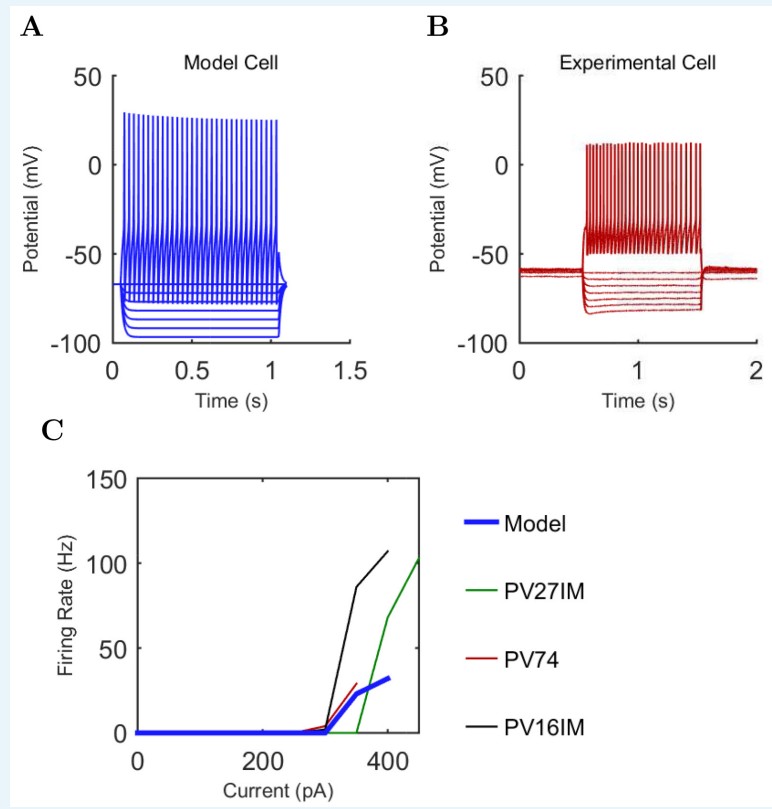

**Appendix 1—figure 9.** Bistratified (**A**) model and (**B**) experimental current sweep. (**C**) Firing rates of model and experimental cells.

**Appendix 1—table 16.** Model Bistratified cell electrophysiological properties.

| Property | Value |
| --- | --- |
| RMP | −67.0 mV |
| Input Resistance | 98.8 MΩ |
| Sag Amplitude | 0.0 mV |
| Sag Tau | − |
| Membrane Tau | 14.7 ms |
| Rheobase | 350.0 pA |
| ISI | 39.0 ms |
| Threshold | −28.1 mV |
| Spike Amplitude | 51.2 mV |
| Slow AHP Amplitude | 48.8 mV |

**Appendix 1—table 17.** Model Bistratified cell ion channels and conductance at highest density location in cell.

| Channel | $G_{max}$ (S/cm$^2$) |
|---|---|
| CavL | 4.000e-03 |
| CavN | 4.000e-04 |
| KCaS | 7.000e-07 |
| Kdrfast | 1.600e-02 |
| KvA | 5.000e-05 |
| KvCaB | 7.000e-08 |
| Navbis | 7.000e-02 |
| leak | 9.001e-05 |

## Model and experimental connectivity

**Appendix 1—table 18.** Structural connection parameters for Bistratified cells, based on *Bezaire and Soltesz (2013)*.

| | Other cell to bis | | | | Bis to other cell | | | |
|---|---|---|---|---|---|---|---|---|
| | # | Syn.s | # | Post | # | Syn.s | # | Post |
| Other type | Conn.s | /Conn. | # | Loc. | Conn.s | /Conn. | # | Loc. |
| Axo | | | | | 11 | 10 | 106 | any dendrite |
| Bis | 16 | 10 | 160 | any dendrite | 16 | 10 | 160 | any dendrite |
| CCK+B | 12 | 8 | 96 | any dendrite | 26 | 10 | 260 | any dendrite |
| Ivy | 24 | 10 | 240 | any dendrite | 12 | 10 | 119 | any dendrite |
| O-LM | 8 | 10 | 80 | apical dendrite | 29 | 10 | 289 | any dendrite |
| Pyr | 366 | 3 | 1098 | apical dendrite | 1410 | 10 | 14095 | any dendrite |
| PV+B | 39 | 1 | 39 | soma | 40 | 10 | 400 | any dendrite |
| SC-A | 1 | 6 | 6 | any dendrite | 3 | 10 | 30 | any dendrite |
| CA3 | 5782 | 2 | 11564 | any dendrite | | | | |
| ECIII | 432 | 2 | 864 | any dendrite | | | | |

## Experimental connection constraints

**Appendix 1—table 19.** Experimental constraints for incoming connections onto Bistratified cells (clamp: black=voltage; purple=current).

| Pre type | Exp. ref. | Hold (mV) | $E_{rev}$ (mV) | Amp. (pA, mV) | Diff. % | $t_{10-90}$ (ms) | Diff. % | $\tau_{decay}$ (ms) | Diff. % |
|---|---|---|---|---|---|---|---|---|---|
| Pyr | *Pawelzik et al., 2002* | −66.0 | 0.0 | 0.77 | −19.6 | 1.58 | +31.3 | 14.68 | +41.1 |
| PV+B | *Cobb et al., 1997* | −55.0 | −70.0 | 0.27 | −27.5 | 0.47 | −52.5 | 7.30 | +30.4 |

**Appendix 1—table 20.** Experimental constraints for outgoing connections from Bistratified cells (clamp: black=voltage; purple=current).

| Post type | Exp. ref. | Hold (mV) | $E_{rev}$ (mV) | Amp. (pA, mV) | Diff. % | $t_{10-90}$ (ms) | Diff. % | $\tau_{decay}$ (ms) | Diff. % |
|---|---|---|---|---|---|---|---|---|---|
| Pyr | *Maccaferri et al., 2000* | −70.0 | 7.0 | 143.21 | −4.5 | 2.22 | +11.2 | 15.40 | −4.3 |

## Model synapse parameters

**Appendix 1—table 21.** Model synaptic parameters for Bistratified cells in the control network.

| Type | Other cell to bis | | | | Bis to other cell | | | |
|------|-----------|-----------|-----------|-----------|-----------|-----------|-----------|-----------|
|      | $E_{rev}$ (mV) | $G_{max}$ (nS) | $\tau_{rise}$ (ms) | $\tau_{decay}$ (ms) | $E_{rev}$ (mV) | $G_{max}$ (nS) | $\tau_{rise}$ (ms) | $\tau_{decay}$ (ms) |
| Axo | | | | | −60.0 | 6.000e-04 | 0.29 | 2.67 |
| Bis | −60.0 | 5.100e-04 | 0.29 | 2.67 | −60.0 | 5.100e-04 | 0.29 | 2.67 |
| CCK+B | −60.0 | 7.000e-04 | 0.43 | 4.49 | −60.0 | 8.000e-04 | 0.29 | 2.67 |
| Ivy | −60.0 | 7.700e-05 | 2.90 | 3.10 | −60.0 | 5.000e-04 | 0.29 | 2.67 |
| O-LM | −60.0 | 1.100e-04 | 0.60 | 15.00 | −60.0 | 2.000e-05 | 1.00 | 8.00 |
| Pyr | 0.0 | 1.900e-03 | 0.11 | 0.25 | −60.0 | 5.100e-04 | 0.11 | 9.70 |
| PV+B | −60.0 | 2.900e-03 | 0.18 | 0.45 | −60.0 | 9.000e-03 | 0.29 | 2.67 |
| SC-A | −60.0 | 6.000e-04 | 0.42 | 4.99 | −60.0 | 8.000e-04 | 0.29 | 2.67 |
| CA3 | 0.0 | 1.500e-04 | 2.00 | 6.30 | | | | |
| ECIII | 0.0 | 1.500e-04 | 2.00 | 6.30 | | | | |

## Physiological characterization of model connections

**Appendix 1—table 22.** Model synaptic properties under voltage clamp at −50 mV with physiological reversal potentials

| Type | Other cell to bis | | | | | Bis to other cell | | | | |
|------|-----------|-----------|-----------|-----------|-----------|-----------|-----------|-----------|-----------|-----------|
|      | Hold (mV) | $E_{rev}$ (mV) | Amp. (pA) | $t_{10-90}$ (ms) | $\tau_{decay}$ (ms) | Hold (mV) | $E_{rev}$ (mV) | Amp. (pA) | $t_{10-90}$ (ms) | $\tau_{decay}$ (ms) |
| Axo | | | | | | −50.0 | −60.0 | 36.77 | 0.70 | 3.70 |
| Bis | −50.0 | −60.0 | 34.34 | 0.70 | 3.72 | −50.0 | −60.0 | 34.34 | 0.70 | 3.72 |
| CCK+B | −50.0 | −60.0 | 48.13 | 0.78 | 5.35 | −50.0 | −60.0 | 48.55 | 0.73 | 4.15 |
| Ivy | −50.0 | −60.0 | 6.39 | 2.15 | 6.63 | −50.0 | −60.0 | 43.40 | 0.60 | 3.17 |
| O-LM | −50.0 | −60.0 | 6.31 | 2.70 | 17.05 | −50.0 | −60.0 | 1.86 | 1.78 | 8.13 |
| Pyr | −50.0 | 0.0 | 64.48 | 0.28 | 1.42 | −50.0 | −60.0 | 13.47 | 2.17 | 15.20 |
| PV+B | −50.0 | −60.0 | 24.45 | 0.17 | 0.73 | −50.0 | −60.0 | 429.34 | 0.57 | 4.13 |
| SC-A | −50.0 | −60.0 | 26.43 | 1.02 | 6.20 | −50.0 | −60.0 | 50.35 | 0.70 | 4.10 |
| CA3 | −50.0 | 0.0 | 13.81 | 2.38 | 8.82 | | | | | |
| ECIII | −50.0 | 0.0 | 12.04 | 3.05 | 9.30 | | | | | |

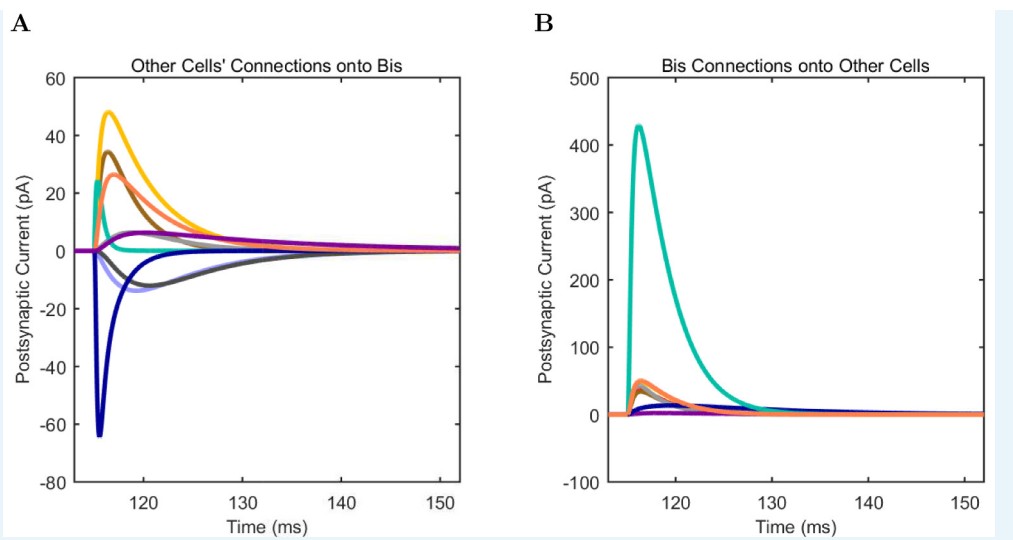

Appendix 1—figure 10. Connections onto (A) and (B) from model Bistratified cells, under voltage clamp at −50 mV with physiological reversal potentials.

## CCK+ basket cell: regular-spiking somatic inhibitor (3600 cells)

## Model and experimental electrophysiology

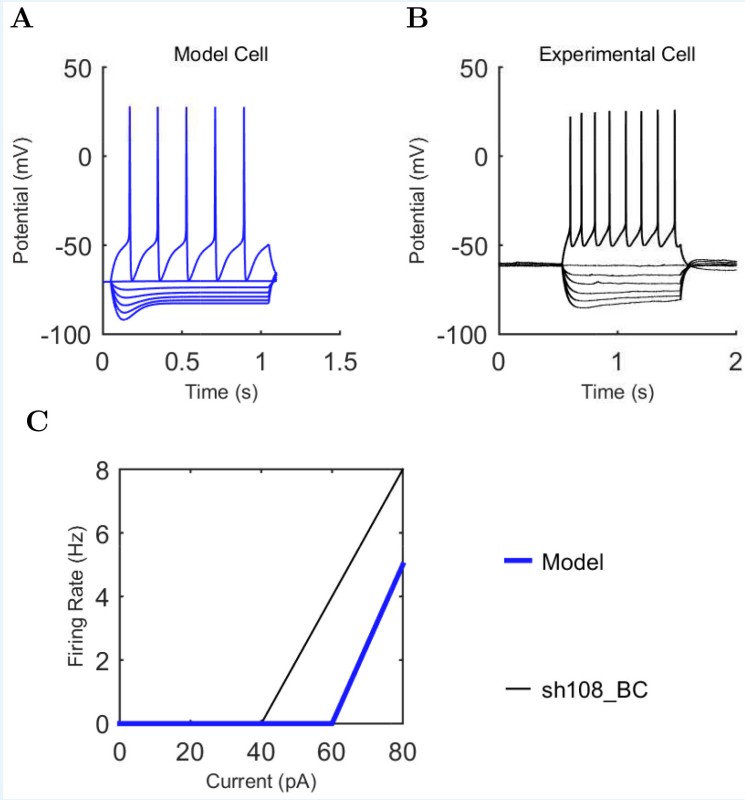

Appendix 1—figure 11. CCK+ Basket (A) model and (B) experimental current sweep. (C) Firing rates of model and experimental cells.

**Appendix 1—table 23.** Model CCK+ Basket cell electrophysiological properties.

| Property | Value |
|---|---|
| RMP | −70.6 mV |
| Input Resistance | 222.4 MΩ |
| Sag Amplitude | 9.2 mV |
| Sag Tau | 45.6 ms |
| Membrane Tau | 25.5 ms |
| Rheobase | 80.0 pA |
| ISI | 180.8 ms |
| Threshold | −38.0 mV |
| Spike Amplitude | 65.9 mV |
| Slow AHP Amplitude | 32.1 mV |

**Appendix 1—table 24.** Model CCK+ Basket cell ion channels and conductance at highest density location in cell.

| Channel | $G_{max}$ (S/cm$^2$) |
|---|---|
| CavL | 2.700e-03 |
| CavN | 2.000e-05 |
| HCN | 1.000e-04 |
| KCaS | 4.000e-06 |
| Kdrfast | 8.000e-05 |
| KvA | 4.000e-04 |
| KvCaB | 4.000e-05 |
| KvGroup | 2.600e-03 |
| Navcck | 1.800e-02 |
| leak | 3.704e-05 |

## Model and experimental connectivity

**Appendix 1—table 25.** Structural connection parameters for CCK+ Basket cells, based on *Bezaire and Soltesz (2013)*.

| Other type | Other cell to CCK+B | | | | CCK+B to other cell | | | |
|---|---|---|---|---|---|---|---|---|
| | # Conn.s | Syn.s /Conn. | # # | Post Loc. | # Conn.s | Syn.s /Conn. | # # | Post Loc. |
| Axo | | | | | 5 | 8 | 39 | any dendrite |
| Bis | 16 | 10 | 160 | any dendrite | 7 | 8 | 58 | any dendrite |
| CCK+B | 35 | 8 | 280 | any dendrite | 35 | 8 | 280 | any dendrite |
| Ivy | 96 | 10 | 960 | any dendrite | 20 | 8 | 156 | any dendrite |
| O-LM | 40 | 10 | 400 | apical dendrite | 9 | 8 | 72 | any dendrite |
| Pyr | | | | | 1125 | 8 | 8998 | any dendrite |
| PV+B | 38 | 1 | 38 | soma | 18 | 8 | 147 | any dendrite |
| SC-A | 6 | 6 | 36 | any dendrite | 3 | 8 | 24 | any dendrite |
| CA3 | 2000 | 2 | 4000 | any dendrite | | | | |
| ECIII | 559 | 2 | 1118 | any dendrite | | | | |

## Experimental connection constraints

Note:No experimental constraints available for incoming synapses to CCK+ Basket cells.

**Appendix 1—table 26.** Experimental constraints for outgoing connections from CCK+ Basket cells (clamp: black=voltage; purple=current).

| Post type | Exp. ref. | Hold (mV) | $E_{rev}$ (mV) | Amp. (pA, mV) | Diff. % | $t_{10-90}$ (ms) | Diff. % | $\tau_{decay}$ (ms) | Diff. % |
|---|---|---|---|---|---|---|---|---|---|
| Pyr | *Lee et al., 2010* | −70.0 | −26.0 | 118.97 | +3.1 | 0.53 | −16.7 | 6.15 | −4.9 |

## Model synapse parameters

**Appendix 1—table 27.** Model synaptic parameters for CCK+ Basket cells in the control network.

| Type | Other cell to CCK+B | | | | CCK+B to other cell | | | |
|---|---|---|---|---|---|---|---|---|
| | $E_{rev}$ (mV) | $G_{max}$ (nS) | $\tau_{rise}$ (ms) | $\tau_{decay}$ (ms) | $E_{rev}$ (mV) | $G_{max}$ (nS) | $\tau_{rise}$ (ms) | $\tau_{decay}$ (ms) |
| Axo | | | | | −60.0 | 7.000e-04 | 0.43 | 4.49 |
| Bis | −60.0 | 8.000e-04 | 0.29 | 2.67 | −60.0 | 7.000e-04 | 0.43 | 4.49 |
| CCK+B | −60.0 | 4.500e-04 | 0.43 | 4.49 | −60.0 | 4.500e-04 | 0.43 | 4.49 |
| Ivy | −60.0 | 3.700e-05 | 2.90 | 3.10 | −60.0 | 3.000e-04 | 0.43 | 4.49 |
| O-LM | −60.0 | 1.200e-03 | 0.73 | 20.20 | −60.0 | 7.000e-04 | 1.00 | 8.00 |
| Pyr | | | | | −60.0 | 5.200e-04 | 0.20 | 4.20 |
| PV+B | −60.0 | 1.200e-03 | 0.29 | 2.67 | −60.0 | 9.000e-03 | 0.43 | 4.49 |
| SC-A | −60.0 | 8.500e-04 | 0.42 | 4.99 | −60.0 | 7.000e-04 | 0.43 | 4.49 |
| CA3 | 0.0 | 6.500e-04 | 2.00 | 6.30 | | | | |
| ECIII | 0.0 | 6.500e-04 | 2.00 | 6.30 | | | | |

## Physiological characterization of model connections

**Appendix 1—table 28.** Model synaptic properties under voltage clamp at −50 mV with physiological reversal potentials

| Type | Other cell to CCK+B | | | | | CCK+B to other cell | | | | |
|---|---|---|---|---|---|---|---|---|---|---|
| | Hold (mV) | $E_{rev}$ (mV) | Amp. (pA) | $t_{10-90}$ (ms) | $\tau_{decay}$ (ms) | Hold (mV) | $E_{rev}$ (mV) | Amp. (pA) | $t_{10-90}$ (ms) | $\tau_{decay}$ (ms) |
| Axo | | | | | | −50.0 | −60.0 | 47.29 | 0.75 | 5.27 |
| Bis | −50.0 | −60.0 | 48.55 | 0.73 | 4.15 | −50.0 | −60.0 | 48.13 | 0.78 | 5.35 |
| CCK+B | −50.0 | −60.0 | 32.19 | 0.73 | 5.30 | −50.0 | −60.0 | 32.19 | 0.73 | 5.30 |
| Ivy | −50.0 | −60.0 | 3.00 | 2.25 | 6.95 | −50.0 | −60.0 | 22.34 | 0.80 | 5.05 |
| O-LM | −50.0 | −60.0 | 40.32 | 3.10 | 28.42 | −50.0 | −60.0 | 54.98 | 1.35 | 9.05 |
| Pyr | | | | | | −50.0 | −60.0 | 24.86 | 0.52 | 6.03 |
| PV+B | −50.0 | −60.0 | 11.31 | 0.42 | 3.08 | −50.0 | −60.0 | 523.11 | 0.68 | 5.70 |
| SC-A | −50.0 | −60.0 | 33.81 | 1.05 | 6.90 | −50.0 | −60.0 | 49.55 | 0.70 | 5.38 |
| CA3 | −50.0 | 0.0 | 55.24 | 2.53 | 9.35 | | | | | |
| ECIII | −50.0 | 0.0 | 43.27 | 3.40 | 10.87 | | | | | |

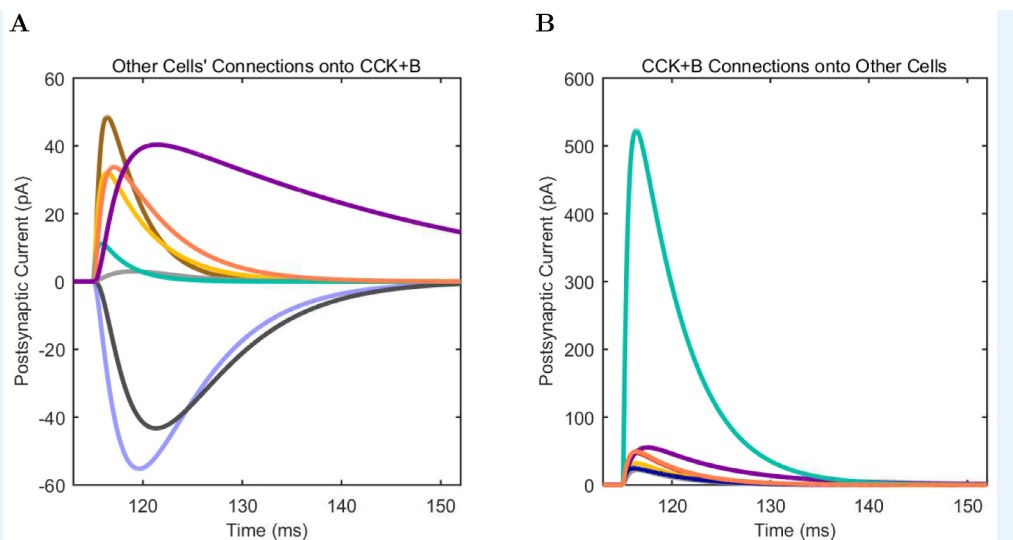

**Appendix 1—figure 12.** Connections onto (**A**) and (**B**) from model CCK+ Basket cells, under voltage clamp at −50 mV with physiological reversal potentials.

## Ivy cell: late-spiking cell (8810 cells)

## Model and experimental electrophysiology

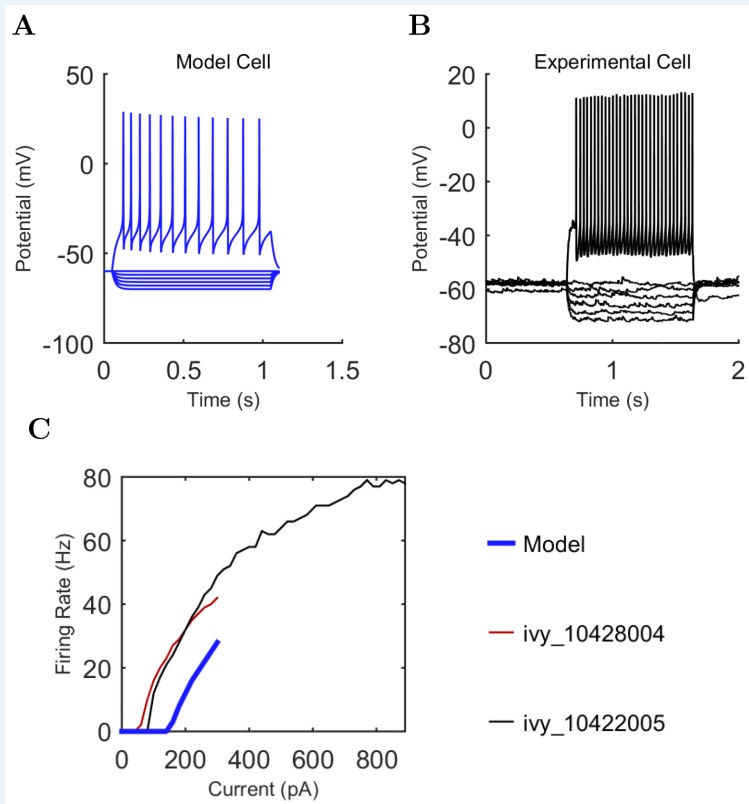

**Appendix 1—figure 13.** Ivy (**A**) model and (**B**) experimental current sweep. (fig:ivypage:firing) Firing rates of model and experimental cells.

**Appendix 1—table 29.** Model Ivy cell electrophysiological properties.

| Property | Value |
|---|---|
| RMP | −60.0 mV |
| Input Resistance | 100.7 MΩ |
| Sag Amplitude | 0.0 mV |
| Sag Tau | − |
| Membrane Tau | 21.3 ms |
| Rheobase | 160.0 pA |
| ISI | 305.5 ms |
| Threshold | −27.7 mV |
| Spike Amplitude | 54.6 mV |
| Slow AHP Amplitude | 20.9 mV |

**Appendix 1—table 30.** Model Ivy cell ion channels and conductance at highest density location in cell.

| Channel | $G_{max}$ (S/cm$^2$) |
|---|---|
| CavL | 5.611e-02 |
| CavN | 5.817e-04 |
| KCaS | 4.515e-07 |
| Kdrfastngf | 1.551e-01 |
| KvAngf | 5.220e-06 |
| KvCaB | 1.024e-06 |
| Navngf | 3.786e+00 |
| leak | 8.471e-05 |

## Model and experimental connectivity

**Appendix 1—table 31.** Structural connection parameters for Ivy cells, based on *Bezaire and Soltesz (2013)*.

| | Other cell to ivy | | | | Ivy to other cell | | | |
|---|---|---|---|---|---|---|---|---|
| | # | Syn.s | # | Post | # | Syn.s | # | Post |
| OtherType | Conn.s | /Conn. | # | Loc. | Conn.s | /Conn. | # | Loc. |
| Axo | | | | | 4 | 10 | 40 | any dendrite |
| Bis | 3 | 10 | 30 | any dendrite | 6 | 10 | 60 | any dendrite |
| CCK+B | 8 | 8 | 64 | any dendrite | 39 | 10 | 392 | any dendrite |
| Ivy | 24 | 10 | 240 | any dendrite | 24 | 10 | 240 | any dendrite |
| NGF | | | | | 11 | 10 | 113 | any dendrite |
| O-LM | | | | | 25 | 10 | 253 | any dendrite |
| Pyr | 9 | 3 | 27 | apical dendrite | 1485 | 10 | 14850 | any dendrite |
| PV+B | 8 | 1 | 8 | soma | 15 | 10 | 150 | any dendrite |
| SC-A | 2 | 6 | 12 | any dendrite | 5 | 10 | 46 | any dendrite |
| CA3 | 1923 | 2 | 3846 | any dendrite | | | | |

## Experimental connection constraints

**Appendix 1—table 32.** Experimental constraints for incoming connections onto Ivy cells (clamp: black=voltage; purple=current).

| Pre type | Exp. ref. | Hold (mV) | $E_{rev}$ (mV) | Amp. (pA, mV) | Diff. % | $t_{10-90}$ (ms) | Diff. % | $\tau_{decay}$ (ms) | Diff. % |
|---|---|---|---|---|---|---|---|---|---|
| Pyr | *Fuentealba et al., 2008* | −65.8 | −70.0 | 0.06 | −97.9 | 1.38 | −8.3 | − | −4.9 |

**Appendix 1—table 33.** Experimental constraints for outgoing connections from Ivy cells (clamp: black=voltage; purple=current).

| Post type | Exp. ref. | Hold (mV) | $E_{rev}$ (mV) | Amp. (pA,mV) | Diff. % | $t_{10-90}$ (ms) | Diff. % | $\tau_{decay}$ (ms) | Diff. % |
|---|---|---|---|---|---|---|---|---|---|
| Pyr | *Fuentealba et al., 2008* | −50.0 | −88.0 | 8.17 | +2.1 | 3.50 | +25.0 | 15.43 | −3.9 |

## Model Synapse Parameters

**Appendix 1—table 34.** Model synaptic parameters for Ivy cells in the control network.

| Type | Other cell to ivy | | | | Ivy to other cell | | | |
|---|---|---|---|---|---|---|---|---|
| | $E_{rev}$ (mV) | $G_{max}$ (nS) | $\tau_{rise}$ (ms) | $\tau_{decay}$ (ms) | $E_{rev}$ (mV) | $G_{max}$ (nS) | $\tau_{rise}$ (ms) | $\tau_{decay}$ (ms) |
| Axo | | | | | −60.0 | 5.700e-05 | 2.90 | 3.10 |
| Bis | −60.0 | 5.000e-04 | 0.29 | 2.67 | −60.0 | 7.700e-05 | 2.90 | 3.10 |
| CCK+B | −60.0 | 3.000e-04 | 0.43 | 4.49 | −60.0 | 3.700e-05 | 2.90 | 3.10 |
| Ivy | −60.0 | 5.700e-05 | 2.90 | 3.10 | −60.0 | 5.700e-05 | 2.90 | 3.10 |
| NGF | | | | | −60.0 | 5.700e-05 | 2.90 | 3.10 |
| O-LM | | | | | −60.0 | 5.700e-05 | 2.90 | 3.10 |
| Pyr | 0.0 | 4.050e-04 | 0.30 | 0.60 | −60.0 | 4.100e-05 | 1.10 | 11.00 |
| PV+B | −60.0 | 1.600e-04 | 0.29 | 2.67 | −60.0 | 7.000e-04 | 2.90 | 3.10 |
| SC-A | −60.0 | 8.500e-04 | 0.42 | 4.99 | −60.0 | 3.700e-05 | 2.90 | 3.10 |
| CA3 | 0.0 | 3.000e-04 | 2.00 | 6.30 | | | | |

## Physiological characterization of model connections

**Appendix 1—table 35.** Model synaptic properties under voltage clamp at −50 mV with physiological reversal potentials

| Type | Other cell to ivy | | | | | Ivy to other cell | | | | |
|---|---|---|---|---|---|---|---|---|---|---|
| | Hold (mV) | $E_{rev}$ (mV) | Amp. (pA) | $t_{10-90}$ (ms) | $\tau_{decay}$ (ms) | Hold (mV) | $E_{rev}$ (mV) | Amp. (pA) | $t_{10-90}$ (ms) | $\tau_{decay}$ (ms) |
| Axo | | | | | | −50.0 | −60.0 | 4.34 | 2.13 | 6.57 |
| Bis | −50.0 | −60.0 | 43.40 | 0.60 | 3.17 | −50.0 | −60.0 | 6.39 | 2.15 | 6.63 |
| CCK +B | −50.0 | −60.0 | 22.34 | 0.80 | 5.05 | −50.0 | −60.0 | 3.00 | 2.25 | 6.95 |
| Ivy | −50.0 | −60.0 | 5.48 | 1.88 | 6.42 | −50.0 | −60.0 | 5.48 | 1.88 | 6.42 |
| NGF | | | | | | −50.0 | −60.0 | 5.48 | 1.88 | 6.42 |
| O-LM | | | | | | −50.0 | −60.0 | 5.32 | 2.10 | 6.33 |
| Pyr | −50.0 | 0.0 | 40.70 | 0.58 | 1.28 | −50.0 | −60.0 | 1.63 | 3.63 | 15.35 |

*Appendix 1—table 35 continued on next page*

Appendix 1—table 35 continued

| | Other cell to ivy | | | | | Ivy to other cell | | | | |
|---|---|---|---|---|---|---|---|---|---|---|
| Type | Hold (mV) | $E_{rev}$ (mV) | Amp. (pA) | $t_{10-90}$ (ms) | $\tau_{decay}$ (ms) | Hold (mV) | $E_{rev}$ (mV) | Amp. (pA) | $t_{10-90}$ (ms) | $\tau_{decay}$ (ms) |
| PV+B | −50.0 | −60.0 | 1.44 | 0.55 | 3.13 | −50.0 | −60.0 | 51.35 | 2.05 | 6.75 |
| SC-A | −50.0 | −60.0 | 46.62 | 0.85 | 5.58 | −50.0 | −60.0 | 3.09 | 2.22 | 6.88 |
| CA3 | −50.0 | 0.0 | 29.42 | 2.05 | 8.60 | | | | | |

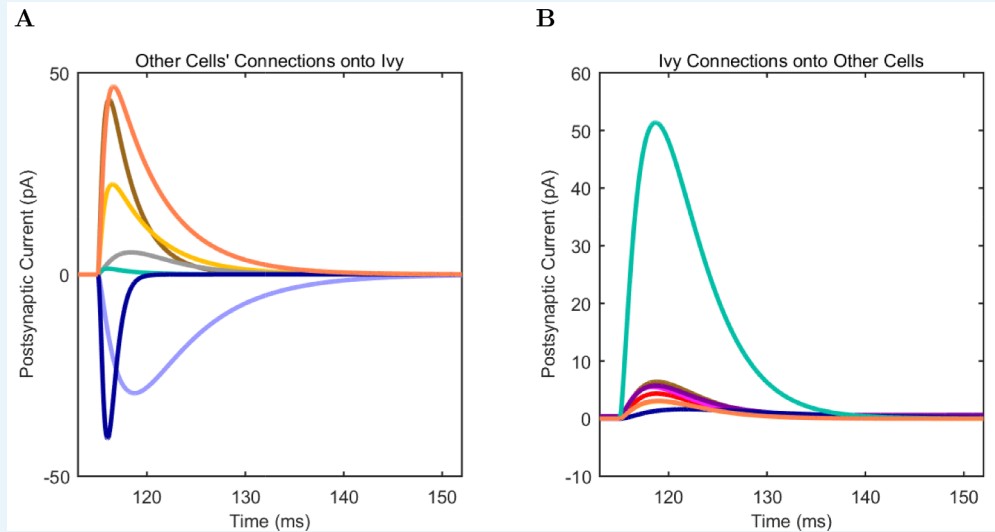

**Appendix 1—figure 14.** Connections onto (**A**) and (**B**) from model Ivy cells, under voltage clamp at −50 mV with physiological reversal potentials.

## Neurogliaform cell: late-spiking feed forward cell (3580 cells)

## Model and experimental electrophysiology

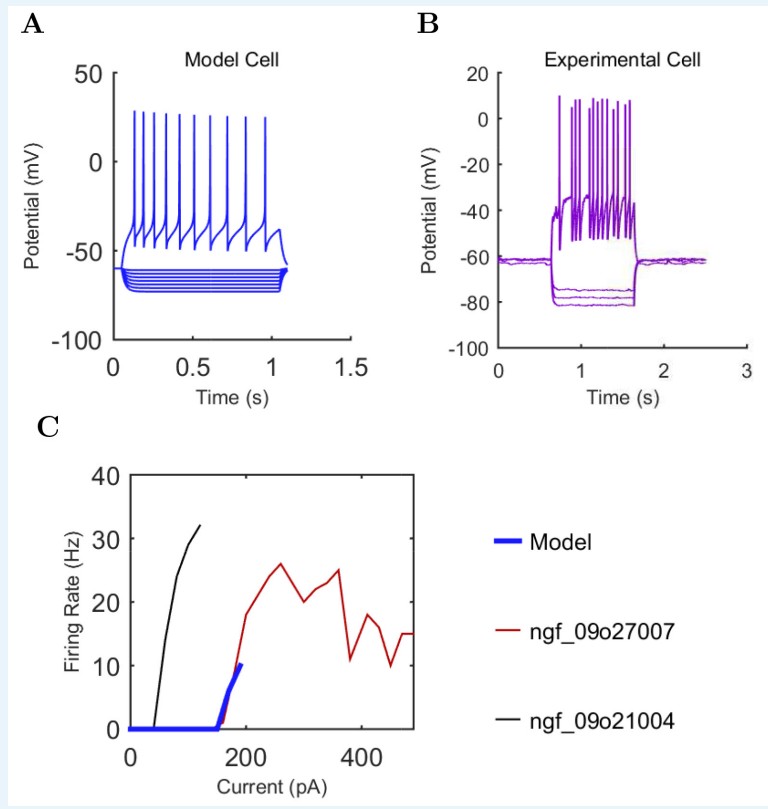

Appendix 1—figure 15. Neurogliaform (A) model and (B) experimental current sweep. (C) Firing rates of model and experimental cells.

Appendix 1—table 36. Model Neurogliaform cell electrophysiological properties.

| Property | Value |
| --- | --- |
| RMP | −60.0 mV |
| Input Resistance | 100.8 MΩ |
| Sag Amplitude | 0.0 mV |
| Sag Tau | − |
| Membrane Tau | 21.3 ms |
| Rheobase | 170.0 pA |
| ISI | 170.3 ms |
| Threshold | −27.8 mV |
| Spike Amplitude | 55.2 mV |
| Slow AHP Amplitude | 20.6 mV |

Appendix 1—table 37. Model Neurogliaform cell ion channels and conductance at highest density location in cell.

| Channel | $G_{max}$ (S/cm$^2$) |
| --- | --- |
| CavL | 5.611e-02 |

*Appendix 1—table 37 continued on next page*

*Appendix 1—table 37 continued*

| Channel | G$_{max}$ (S/cm$^2$) |
|---|---|
| CavN | 5.817e-04 |
| KCaS | 4.515e-07 |
| Kdrfastngf | 1.551e-01 |
| KvAngf | 5.220e-06 |
| KvCaB | 1.024e-06 |
| Navngf | 3.786e+00 |
| leak | 8.471e-05 |

## Model and experimental connectivity

**Appendix 1—table 38.** Structural connection parameters for Neurogliaform cells, based on *Bezaire and Soltesz (2013)*.

| | Other cell to NGF | | | | NGF to other cell | | | |
|---|---|---|---|---|---|---|---|---|
| Other type | # Conn.s | Syn.s /Conn. | # # | Post Loc. | # Conn.s | Syn.s /Conn. | # # | Post Loc. |
| Ivy | 28 | 10 | 280 | any dendrite | | | | |
| NGF | 17 | 10 | 170 | apical dendrite | 17 | 10 | 170 | apical dendrite |
| O-LM | 13 | 10 | 130 | apical dendrite | | | | |
| Pyr | | | | | 1218 | 10 | 12181 | apical dendrite |
| ECIII | 523 | 2 | 1046 | any dendrite | | | | |

## Experimental connection constraints

**Appendix 1—table 39.** Experimental constraints for incoming connections onto Neurogliaform cells (clamp: black=voltage; purple=current).

| Pre type | Exp. ref. | Hold (mV) | E$_{rev}$ (mV) | Amp. (pA, mV) | Diff. % | t$_{10-90}$ (ms) | Diff. % | $\tau_{decay}$ (ms) | Diff. % |
|---|---|---|---|---|---|---|---|---|---|
| NGF | *Karayannis et al., 2010* | −65.0 | −11.0 | 85.50 | +0.2 | 4.83 | −1.7 | 32.03 | −46.9 |
| O-LM | *Elfant et al., 2007* | −50.0 | −70.0 | 18.43 | −4.0 | 1.98 | −10.2 | 11.63 | +7.6 |

**Appendix 1—table 40.** Experimental constraints for outgoing connections from Neurogliaform cells (clamp: black=voltage; purple=current).

| Post type | Exp. ref. | Hold (mV) | E$_{rev}$ (mV) | Amp. (pA,mV) | Diff. % | t$_{10-90}$ (ms) | Diff. % | $\tau_{decay}$ (ms) | Diff. % |
|---|---|---|---|---|---|---|---|---|---|
| NGF | *Karayannis et al., 2010* | −65.0 | −11.0 | 85.50 | +0.2 | 4.83 | −1.7 | 32.03 | −46.9 |
| Pyr | *Price et al., 2008* | −50.0 | −89.0 | 5.25 | +7.1 | 15.48 | −3.9 | 32.73 | −34.5 |

## Model synapse parameters

**Appendix 1—table 41.** Model synaptic parameters for Neurogliaform cells in the control network.

| | Other cell to NGF | | | | NGF to other cell | | | |
|---|---|---|---|---|---|---|---|---|
| Type | $E_{rev}$ (mV) | $G_{max}$ (nS) | $\tau_{rise}$ (ms) | $\tau_{decay}$ (ms) | $E_{rev}$ (mV) | $G_{max}$ (nS) | $\tau_{rise}$ (ms) | $\tau_{decay}$ (ms) |
| Ivy | −60.0 | 5.700e-05 | 2.90 | 3.10 | | | | |
| NGF | −60.0 | 1.600e-04 | 3.10 | 42.00 | −60.0 | 1.600e-04 | 3.10 | 42.00 |
| O-LM | −60.0 | 9.800e-05 | 1.30 | 10.20 | | | | |
| Pyr | | | | | −60.0 | 6.500e-05 | 9.00 | 39.00 |
| ECIII | 0.0 | 3.500e-03 | 2.00 | 6.30 | | | | |

## Physiological characterization of model connections

**Appendix 1—table 42.** Model synaptic properties under voltage clamp at −50 mV with physiological reversal potentials

| | Other cell to NGF | | | | | NGF to other cell | | | | |
|---|---|---|---|---|---|---|---|---|---|---|
| Type | Hold (mV) | $E_{rev}$ (mV) | Amp. (pA) | $t_{10-90}$ (ms) | $\tau_{decay}$ (ms) | Hold (mV) | $E_{rev}$ (mV) | Amp. (pA) | $t_{10-90}$ (ms) | $\tau_{decay}$ (ms) |
| Ivy | −50.0 | −60.0 | 5.48 | 1.88 | 6.42 | | | | | |
| NGF | −50.0 | −60.0 | 17.52 | 5.67 | 14.32 | −50.0 | −60.0 | 17.52 | 5.67 | 14.32 |
| O-LM | −50.0 | −60.0 | 9.14 | 1.98 | 11.63 | | | | | |
| Pyr | | | | | | −50.0 | −60.0 | 1.10 | 65.58 | 0.00 |
| ECIII | −50.0 | 0.0 | 324.35 | 2.13 | 8.80 | | | | | |

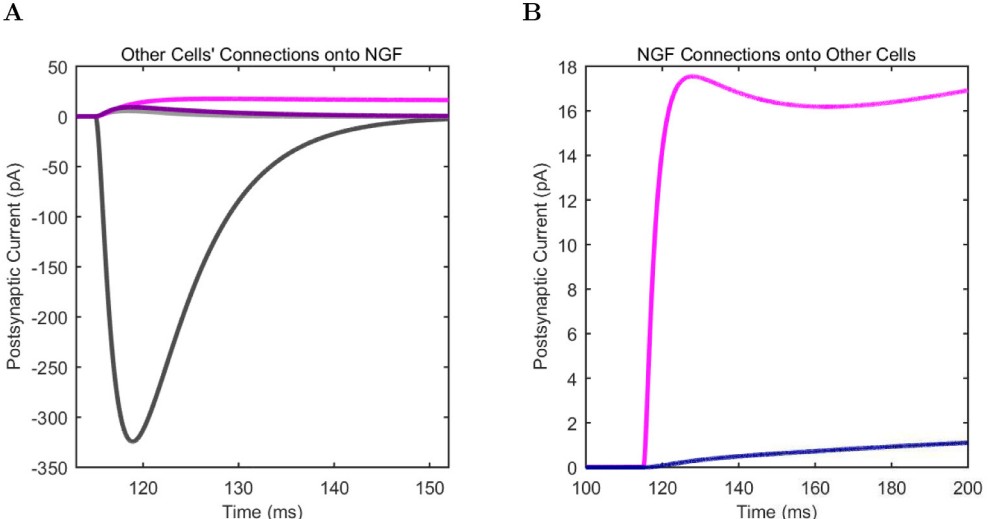

**Appendix 1—figure 16.** Connections onto (**A**) and (**B**) from model Neurogliaform cells, under voltage clamp at −50 mV with physiological reversal potentials.

## O-LM cell: feed back cell (1640 cells)

Model and experimental electrophysiology

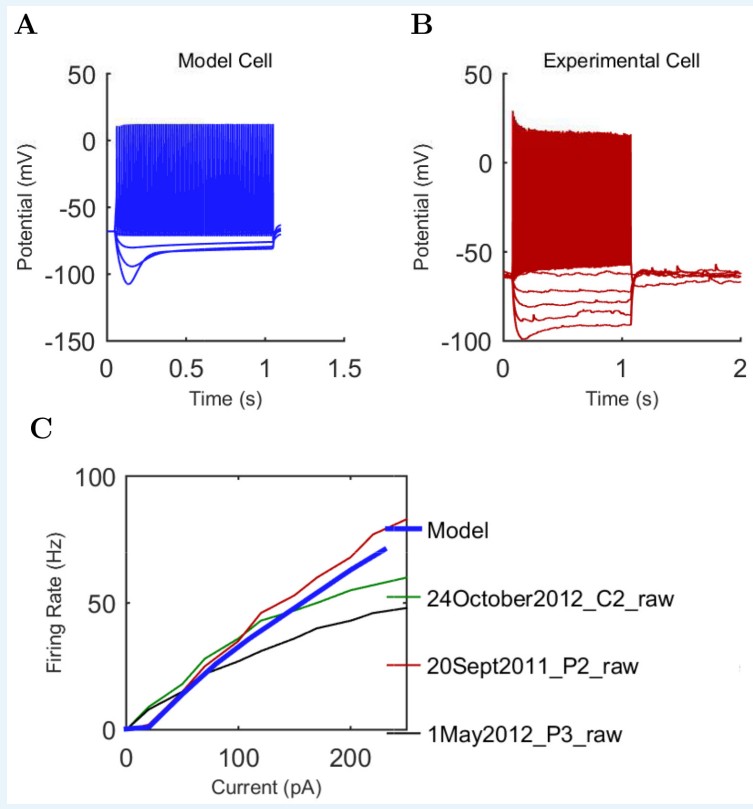

**Appendix 1—figure 17.** O-LM (**A**) model and (**B**) experimental current sweep. (**C**) Firing rates of model and experimental cells.

**Appendix 1—table 43.** Model O-LM cell electrophysiological properties.

| Property | Value |
| --- | --- |
| RMP | −68.0 mV |
| Input Resistance | 267.7 MΩ |
| Sag Amplitude | 26.5 mV |
| Sag Tau | 42.5 ms |
| Membrane Tau | 22.7 ms |
| Rheobase | 20.0 pA |
| ISI | 66.9 ms |
| Threshold | −37.8 mV |
| Spike Amplitude | 42.6 mV |
| Slow AHP Amplitude | 34.6 mV |

**Appendix 1—table 44.** Model O-LM cell ion channels and conductance at highest density location in cell.

| Channel | $G_{max}$ (S/cm$^2$) |
| --- | --- |
| HCNolm | 5.000e-04 |

*Appendix 1—table 44 continued on next page*

*Appendix 1—table 44 continued*

| Channel | G$_{max}$ (S/cm$^2$) |
|---|---|
| Kdrfast | 1.174e-01 |
| KvAolm | 4.950e-03 |
| Nav | 2.340e-02 |
| leak | 1.000e-05 |

## Model and experimental connectivity

**Appendix 1—table 45.** Structural connection parameters for O-LM cells, based on *Bezaire and Soltesz (2013)*.

| | Other cell to O-LM | | | | O-LM to other cell | | | |
|---|---|---|---|---|---|---|---|---|
| Other type | # Conn.s | Syn.s /Conn. | # # | Post Loc. | # Conn.s | Syn.s /Conn. | # # | Post Loc. |
| Axo | | | | | 7 | 10 | 71 | apical dendrite |
| Bis | 39 | 10 | 390 | any dendrite | 11 | 10 | 107 | apical dendrite |
| CCK+B | 20 | 8 | 160 | any dendrite | 88 | 10 | 878 | apical dendrite |
| Ivy | 136 | 10 | 1360 | any dendrite | | | | |
| NGF | | | | | 28 | 10 | 283 | apical dendrite |
| O-LM | 6 | 10 | 60 | basal dendrite | 6 | 10 | 60 | basal dendrite |
| Pyr | 2379 | 3 | 7137 | basal dendrite | 1520 | 10 | 15195 | apical dendrite |
| PV+B | | | | | 27 | 10 | 269 | apical dendrite |
| SC-A | | | | | 10 | 10 | 97 | apical dendrite |

## Experimental connection constraints

Note:No experimental constraints available for incoming synapses to O-LM cells.

**Appendix 1—table 46.** Experimental constraints for outgoing connections from O-LM cells (clamp: black=voltage; purple=current).

| Post type | Exp. ref. | Hold (mV) | $E_{rev}$ (mV) | Amp. (pA,mV) | Diff. % | $t_{10-90}$ (ms) | Diff. % | $\tau_{decay}$ (ms) | Diff. % |
|---|---|---|---|---|---|---|---|---|---|
| NGF | *Elfant et al., 2007* | −50.0 | −70.0 | 18.43 | −4.0 | 1.98 | −10.2 | 11.63 | +7.6 |
| Pyr | *Maccaferri et al., 2000* | −70.0 | 7.0 | 24.35 | −6.3 | 4.68 | −24.6 | 18.88 | −9.3 |
| SC-A | *Elfant et al., 2007* | −50.0 | −70.0 | 17.06 | −12.5 | 4.07 | +114.5 | 30.08 | −3.6 |

## Model synapse parameters

**Appendix 1—table 47.** Model synaptic parameters for O-LM cells in the control network.

| | Other cell to O-LM | | | | O-LM to other cell | | | |
|---|---|---|---|---|---|---|---|---|
| Type | $E_{rev}$ (mV) | $G_{max}$ (nS) | $\tau_{rise}$ (ms) | $\tau_{decay}$ (ms) | $E_{rev}$ (mV) | $G_{max}$ (nS) | $\tau_{rise}$ (ms) | $\tau_{decay}$ (ms) |
| Axo | | | | | −60.0 | 1.200e-04 | 0.73 | 10.00 |
| Bis | −60.0 | 2.000e-05 | 1.00 | 8.00 | −60.0 | 1.100e-04 | 0.60 | 15.00 |
| CCK+B | −60.0 | 7.000e-04 | 1.00 | 8.00 | −60.0 | 1.200e-03 | 0.73 | 20.20 |
| Ivy | −60.0 | 5.700e-05 | 2.90 | 3.10 | | | | |
| NGF | | | | | −60.0 | 9.800e-05 | 1.30 | 10.20 |

*Appendix 1—table 47 continued on next page*

*Appendix 1—table 47 continued*

| | Other cell to O-LM | | | | O-LM to other cell | | | |
|---|---|---|---|---|---|---|---|---|
| Type | $E_{rev}$ (mV) | $G_{max}$ (nS) | $\tau_{rise}$ (ms) | $\tau_{decay}$ (ms) | $E_{rev}$ (mV) | $G_{max}$ (nS) | $\tau_{rise}$ (ms) | $\tau_{decay}$ (ms) |
| O-LM | −60.0 | 1.200e-03 | 0.25 | 7.50 | −60.0 | 1.200e-03 | 0.25 | 7.50 |
| Pyr | 0.0 | 2.000e-04 | 0.30 | 0.60 | −60.0 | 3.000e-04 | 0.13 | 11.00 |
| PV+B | | | | | −60.0 | 1.100e-03 | 0.25 | 7.50 |
| SC-A | | | | | −60.0 | 1.500e-04 | 0.07 | 29.00 |

## Physiological characterization of model connections

**Appendix 1—table 48.** Model synaptic properties under voltage clamp at −50 mV with physiological reversal potentials

| | Other cell to O-LM | | | | | O-LM to other cell | | | | |
|---|---|---|---|---|---|---|---|---|---|---|
| Type | Hold (mV) | $E_{rev}$ (mV) | Amp. (pA) | $t_{10-90}$ (ms) | $\tau_{decay}$ (ms) | Hold (mV) | $E_{rev}$ (mV) | Amp. (pA) | $t_{10-90}$ (ms) | $\tau_{decay}$ (ms) |
| Axo | | | | | | −50.0 | −60.0 | 4.76 | 2.55 | 12.03 |
| Bis | −50.0 | −60.0 | 1.86 | 1.78 | 8.13 | −50.0 | −60.0 | 6.31 | 2.70 | 17.05 |
| CCK +B | −50.0 | −60.0 | 54.98 | 1.35 | 9.05 | −50.0 | −60.0 | 40.32 | 3.10 | 28.42 |
| Ivy | −50.0 | −60.0 | 5.32 | 2.10 | 6.33 | | | | | |
| NGF | | | | | | −50.0 | −60.0 | 9.14 | 1.98 | 11.63 |
| O-LM | −50.0 | −60.0 | 78.69 | 1.05 | 9.30 | −50.0 | −60.0 | 78.69 | 1.05 | 9.30 |
| Pyr | −50.0 | 0.0 | 17.47 | 0.60 | 1.53 | −50.0 | −60.0 | 0.54 | 3.70 | 14.10 |
| PV+B | | | | | | −50.0 | −60.0 | 35.53 | 1.65 | 10.18 |
| SC-A | | | | | | −50.0 | −60.0 | 7.91 | 3.90 | 29.83 |

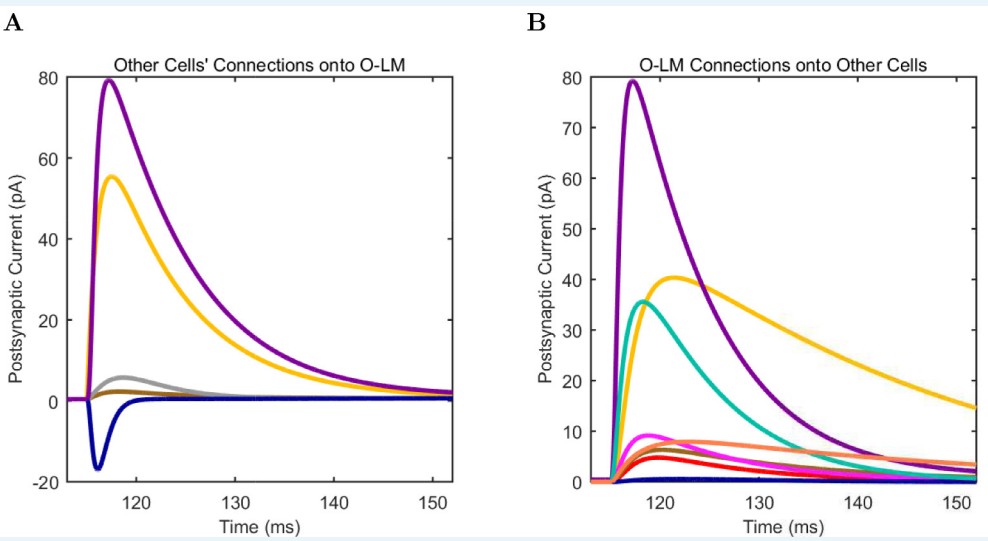

**Appendix 1—figure 18.** Connections onto (**A**) and (**B**) from model O-LM cells, under voltage clamp at −50 mV with physiological reversal potentials.

## PV+ basket cell: fast-spiking somatic inhibitor (5530 cells)

### Model and experimental electrophysiology

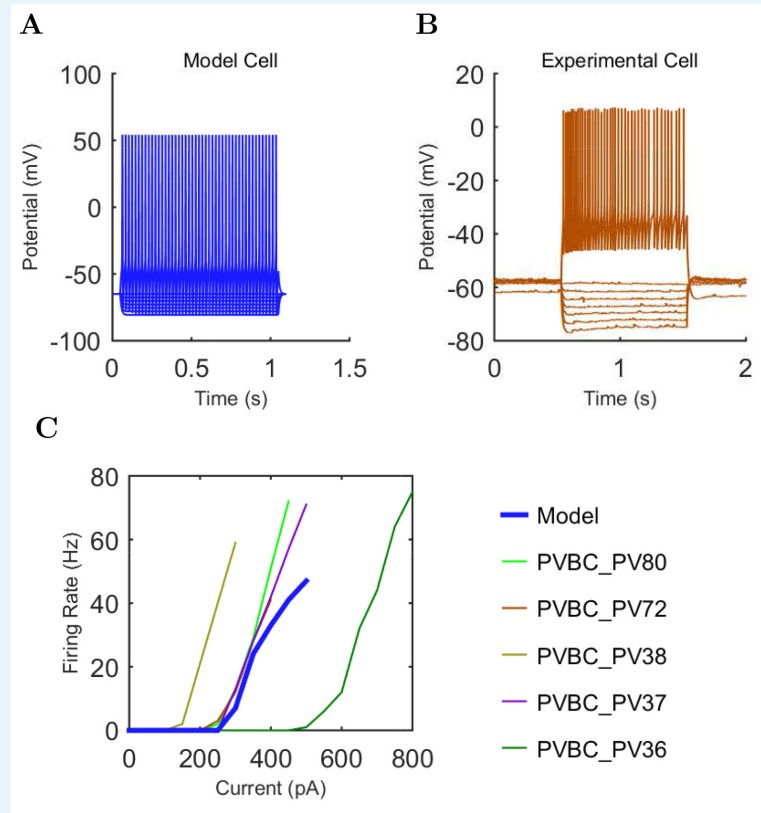

**Appendix 1—figure 19.** PV+ Basket (**A**) model and (**B**) experimental current sweep. (**C**) Firing rates of model and experimental cells.

**Appendix 1—table 49.** Model PV+ Basket cell electrophysiological properties.

| Property | Value |
|---|---|
| RMP | −65.0 mV |
| Input Resistance | 52.1 MΩ |
| Sag Amplitude | − |
| Sag Tau | − |
| Membrane Tau | 7.0 ms |
| Rheobase | 300.0 pA |
| ISI | 151.4 ms |
| Threshold | −36.7 mV |
| Spike Amplitude | 90.7 mV |
| Slow AHP Amplitude | 41.4 mV |

**Appendix 1—table 50.** Model PV+ Basket cell ion channels and conductance at highest density location in cell.

| Channel | $G_{max}$ (S/cm$^2$) |
|---|---|
| CavL | 5.000e-03 |

*Appendix 1—table 50 continued on next page*

*Appendix 1—table 50 continued*

| Channel | Gmax (S/cm²) |
|---|---|
| CavN | 8.000e-04 |
| KCaS | 2.000e-06 |
| Kdrfast | 1.300e-02 |
| KvA | 1.500e-04 |
| KvCaB | 2.000e-07 |
| Navaxonp | 1.500e-01 |
| leak | 1.800e-04 |

## Model and experimental connectivity

**Appendix 1—table 51.** Structural connection parameters for PV+ Basket cells, based on *Bezaire and Soltesz (2013)*.

| | Other cell to PV+B | | | | PV+B to other cell | | | |
|---|---|---|---|---|---|---|---|---|
| Other type | # Conn.s | Syn.s /Conn. | # # | Post Loc. | # Conn.s | Syn.s /Conn. | # # | Post Loc. |
| Axo | | | | | 10 | 1 | 10 | soma |
| Bis | 16 | 10 | 160 | any dendrite | 16 | 1 | 15 | soma |
| CCK+B | 12 | 8 | 96 | any dendrite | 25 | 1 | 24 | soma |
| Ivy | 24 | 10 | 240 | any dendrite | 13 | 1 | 12 | soma |
| O-LM | 8 | 10 | 80 | apical dendrite | | | | |
| Pyr | 424 | 3 | 1272 | apical dendrite | 958 | 11 | 10533 | soma |
| PV+B | 39 | 1 | 39 | soma | 39 | 1 | 39 | soma |
| SC-A | | | | | 2 | 1 | 1 | soma |
| CA3 | 6047 | 2 | 12094 | any dendrite | | | | |

## Experimental connection constraints

**Appendix 1—table 52.** Experimental constraints for incoming connections onto PV+ Basket cells (clamp: black=voltage; purple=current).

| Pre type | Exp. ref. | Hold (mV) | $E_{rev}$ (mV) | Amp. (pA, mV) | Diff. % | $t_{10-90}$ (ms) | Diff. % | $\tau_{decay}$ (ms) | Diff. % |
|---|---|---|---|---|---|---|---|---|---|
| Pyr | *Lee et al., 2014* | −60.0 | 0.0 | 15.09 | −67.7 | 0.28 | −72.5 | 1.83 | −55.7 |
| PV+B | *Cobb et al., 1997* | −59.0 | −70.0 | 0.29 | +14.9 | 2.67 | +105.8 | 14.72 | −45.5 |

**Appendix 1—table 53.** Experimental constraints for outgoing connections from PV+ Basket cells (clamp: black=voltage; purple=current).

| Post type | Exp. ref. | Hold (mV) | $E_{rev}$ (mV) | Amp. (pA,mV) | Diff. % | $t_{10-90}$ (ms) | Diff. % | $\tau_{decay}$ (ms) | Diff. % |
|---|---|---|---|---|---|---|---|---|---|
| Bis | *Cobb et al., 1997* | −55.0 | −70.0 | 0.27 | −27.5 | 0.47 | −52.5 | 11.85 | +30.4 |
| Pyr | *Szabadics et al., 2007* | −70.0 | −26.0 | 91.94 | −13.9 | 0.50 | −5.7 | 6.70 | +4.7 |
| PV+B | *Cobb et al., 1997* | −59.0 | −70.0 | 0.29 | +14.9 | 2.67 | +105.8 | 13.45 | −45.5 |

## Model synapse parameters

**Appendix 1—table 54.** Model synaptic parameters for PV+ Basket cells in the control network.

| Type | Other cell to PV+B | | | | PV+B to other cell | | | |
|------|-----------|-----------|-----------|-----------|-----------|-----------|-----------|-----------|
| | $E_{rev}$ (mV) | $G_{max}$ (nS) | $\tau_{rise}$ (ms) | $\tau_{decay}$ (ms) | $E_{rev}$ (mV) | $G_{max}$ (nS) | $\tau_{rise}$ (ms) | $\tau_{decay}$ (ms) |
| Axo | | | | | −60.0 | 1.200e-04 | 0.29 | 2.67 |
| Bis | −60.0 | 9.000e-03 | 0.29 | 2.67 | −60.0 | 2.900e-03 | 0.18 | 0.45 |
| CCK+B | −60.0 | 9.000e-03 | 0.43 | 4.49 | −60.0 | 1.200e-03 | 0.29 | 2.67 |
| Ivy | −60.0 | 7.000e-04 | 2.90 | 3.10 | −60.0 | 1.600e-04 | 0.29 | 2.67 |
| O-LM | −60.0 | 1.100e-03 | 0.25 | 7.50 | | | | |
| Pyr | 0.0 | 7.000e-04 | 0.07 | 0.20 | −60.0 | 2.000e-04 | 0.30 | 6.20 |
| PV+B | −60.0 | 1.600e-03 | 0.08 | 4.80 | −60.0 | 1.600e-03 | 0.08 | 4.80 |
| SC-A | | | | | −60.0 | 6.000e-04 | 0.29 | 2.67 |
| CA3 | 0.0 | 2.200e-04 | 2.00 | 6.30 | | | | |

## Physiological characterization of model connections

**Appendix 1—table 55.** Model synaptic properties under voltage clamp at −50 mV with physiological reversal potentials

| Type | Other cell to PV+B | | | | | PV+B to other cell | | | | |
|------|------------|-----------|-----------|-----------|-----------|------------|-----------|-----------|-----------|-----------|
| | Hold (mV) | $E_{rev}$ (mV) | Amp. (pA) | $t_{10-90}$ (ms) | $\tau_{decay}$ (ms) | Hold (mV) | $E_{rev}$ (mV) | Amp. (pA) | $t_{10-90}$ (ms) | $\tau_{decay}$ (ms) |
| Axo | | | | | | −50.0 | −60.0 | 1.08 | 0.45 | 3.13 |
| Bis | −50.0 | −60.0 | 429.34 | 0.57 | 4.13 | −50.0 | −60.0 | 24.45 | 0.17 | 0.73 |
| CCK+B | −50.0 | −60.0 | 523.11 | 0.68 | 5.70 | −50.0 | −60.0 | 11.31 | 0.42 | 3.08 |
| Ivy | −50.0 | −60.0 | 51.35 | 2.05 | 6.75 | −50.0 | −60.0 | 1.44 | 0.55 | 3.13 |
| O-LM | −50.0 | −60.0 | 35.53 | 1.65 | 10.18 | | | | | |
| Pyr | −50.0 | 0.0 | 14.75 | 0.25 | 1.77 | −50.0 | −60.0 | 20.56 | 0.50 | 6.70 |
| PV+B | −50.0 | −60.0 | 13.94 | 0.23 | 5.25 | −50.0 | −60.0 | 13.94 | 0.23 | 5.25 |
| SC-A | | | | | | −50.0 | −60.0 | 5.71 | 0.42 | 3.08 |
| CA3 | −50.0 | 0.0 | 19.71 | 2.38 | 8.78 | | | | | |

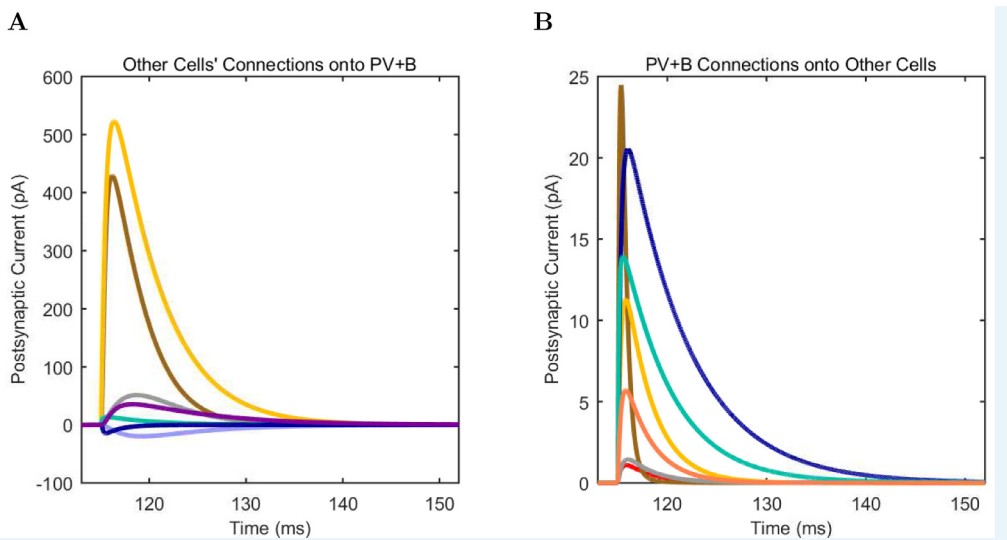

**Appendix 1—figure 20.** Connections onto (**A**) and (**B**) from model PV+ Basket cells, under voltage clamp at −50 mV with physiological reversal potentials.

## Schaffer collateral-associated cell: regular-spiking dendritic inhibitor (400 cells)

### Model and experimental electrophysiology

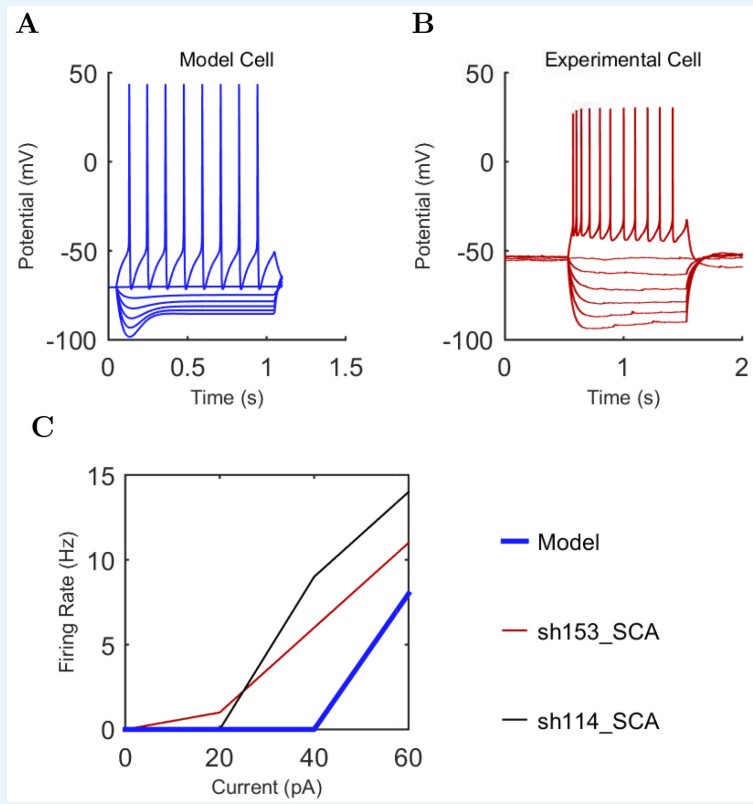

**Appendix 1—figure 21.** Schaffer Collateral-Associated (**A**) model and (**B**) experimental current sweep. (**C**) Firing rates of model and experimental cells.

**Appendix 1—table 56.** Model Schaffer Collateral-Associated cell electrophysiological properties.

| Property | Value |
|---|---|
| RMP | −70.5 mV |
| Input Resistance | 300.0 MΩ |
| Sag Amplitude | 12.9 mV |
| Sag Tau | 41.7 ms |
| Membrane Tau | 28.9 ms |
| Rheobase | 60.0 pA |
| ISI | 115.9 ms |
| Threshold | −36.6 mV |
| Spike Amplitude | 80.3 mV |
| Slow AHP Amplitude | 35.2 mV |

**Appendix 1—table 57.** Model Schaffer Collateral-Associated cell ion channels and conductance at highest density location in cell.

| Channel | $G_{max}$ (S/cm$^2$) |
|---|---|
| CavL | 1.000e-03 |
| CavN | 2.000e-05 |
| HCN | 7.000e-05 |
| KCaS | 1.000e-06 |
| Kdrfast | 6.000e-05 |
| KvA | 1.000e-04 |
| KvCaB | 7.000e-06 |
| KvGroup | 2.200e-03 |
| Navcck | 4.000e-02 |
| leak | 2.857e-05 |

## Model and experimental connectivity

**Appendix 1—table 58.** Structural connection parameters for Schaffer Collateral-Associated cells, based on *Bezaire and Soltesz (2013)*.

| | Other cell to SC-A | | | | SC-A to other cell | | | |
|---|---|---|---|---|---|---|---|---|
| Other type | # Conn.s | Syn.s /Conn. | # # | Post Loc. | # Conn.s | Syn.s /Conn. | # # | Post Loc. |
| Axo | | | | | 4 | 6 | 22 | any dendrite |
| Bis | 17 | 10 | 170 | any dendrite | 6 | 6 | 33 | any dendrite |
| CCK+B | 27 | 8 | 216 | any dendrite | 54 | 6 | 324 | any dendrite |
| Ivy | 102 | 10 | 1020 | any dendrite | 44 | 6 | 264 | any dendrite |
| O-LM | 40 | 10 | 400 | apical dendrite | | | | |
| Pyr | 105 | 3 | 315 | apical dendrite | | | | |
| PV+B | 24 | 1 | 24 | soma | | | | |
| CA3 | 1940 | 2 | 3880 | any dendrite | | | | |
| ECIII | 573 | 2 | 1146 | any dendrite | | | | |

## Experimental connection constraints

**Appendix 1—table 59.** Experimental constraints for incoming connections onto Schaffer Collateral-Associated cells (clamp: black=voltage; purple=current).

| Pre type | Exp. ref. | Hold (mV) | $E_{rev}$ (mV) | Amp. (pA,mV) | Diff. % | $t_{10-90}$ (ms) | Diff. % | $\tau_{decay}$ (ms) | Diff. % |
|---|---|---|---|---|---|---|---|---|---|
| O-LM | *Elfant et al., 2007* | −50.0 | −70.0 | 17.06 | −12.5 | 4.07 | +114.5 | 30.08 | −3.6 |
| SC-A | *Pawelzik et al., 2002* | −58.0 | −70.0 | 1.53 | −405.6 | 2.35 | −41.3 | 22.90 | −33.2 |

**Appendix 1—table 60.** Experimental constraints for outgoing connections from Schaffer Collateral-Associated cells (clamp: black=voltage; purple=current).

| Post type | Exp. ref. | Hold (mV) | $E_{rev}$ (mV) | Amp. (pA, mV) | Diff. % | $t_{10-90}$ (ms) | Diff. % | $\tau_{decay}$ (ms) | Diff. % |
|---|---|---|---|---|---|---|---|---|---|
| Pyr | *Lee et al., 2010* | −70.0 | −26.0 | 52.42 | −12.9 | 1.63 | +13.6 | 8.55 | +3.0 |
| SC-A | *Pawelzik et al., 2002* | −58.0 | −70.0 | 1.53 | −405.6 | 2.35 | −41.3 | 27.98 | −33.2 |

## Model synapse parameters

**Appendix 1—table 61.** Model synaptic parameters for Schaffer Collateral-Associated cells in the control network.

| | Other cell to SC-A | | | | SC-A to other cell | | | |
|---|---|---|---|---|---|---|---|---|
| Type | $E_{rev}$ (mV) | $G_{max}$ (nS) | $\tau_{rise}$ (ms) | $\tau_{decay}$ (ms) | $E_{rev}$ (mV) | $G_{max}$ (nS) | $\tau_{rise}$ (ms) | $\tau_{decay}$ (ms) |
| Axo | | | | | −60.0 | 6.000e-04 | 0.42 | 4.99 |
| Bis | −60.0 | 8.000e-04 | 0.29 | 2.67 | −60.0 | 6.000e-04 | 0.42 | 4.99 |
| CCK+B | −60.0 | 7.000e-04 | 0.43 | 4.49 | −60.0 | 8.500e-04 | 0.42 | 4.99 |
| Ivy | −60.0 | 3.700e-05 | 2.90 | 3.10 | −60.0 | 8.500e-04 | 0.42 | 4.99 |
| O-LM | −60.0 | 1.500e-04 | 0.07 | 29.00 | | | | |
| Pyr | 0.0 | 4.050e-04 | 0.30 | 0.60 | | | | |
| PV+B | −60.0 | 6.000e-04 | 0.29 | 2.67 | | | | |
| CA3 | 0.0 | 3.000e-04 | 2.00 | 6.30 | | | | |
| ECIII | 0.0 | 4.500e-04 | 2.00 | 6.30 | | | | |

## Physiological characterization of model connections

**Appendix 1—table 62.** Model synaptic properties under voltage clamp at −50 mV with physiological reversal potentials

| | Other cell to SC-A | | | | | SC-A to other cell | | | | |
|---|---|---|---|---|---|---|---|---|---|---|
| Type | Hold (mV) | $E_{rev}$ (mV) | Amp. (pA) | $t_{10-90}$ (ms) | $\tau_{decay}$ (ms) | Hold (mV) | $E_{rev}$ (mV) | Amp. (pA) | $t_{10-90}$ (ms) | $\tau_{decay}$ (ms) |
| Axo | | | | | | −50.0 | −60.0 | 24.00 | 1.00 | 6.13 |
| Bis | −50.0 | −60.0 | 50.35 | 0.70 | 4.10 | −50.0 | −60.0 | 26.43 | 1.02 | 6.20 |
| CCK+B | −50.0 | −60.0 | 49.55 | 0.70 | 5.38 | −50.0 | −60.0 | 33.81 | 1.05 | 6.90 |

*Appendix 1—table 62 continued on next page*

*Appendix 1—table 62 continued*

| | Other cell to SC-A | | | | | SC-A to other cell | | | | |
| Type | Hold (mV) | $E_{rev}$ (mV) | Amp. (pA) | $t_{10-90}$ (ms) | $\tau_{decay}$ (ms) | Hold (mV) | $E_{rev}$ (mV) | Amp. (pA) | $t_{10-90}$ (ms) | $\tau_{decay}$ (ms) |
|---|---|---|---|---|---|---|---|---|---|---|
| Ivy | −50.0 | −60.0 | 3.09 | 2.22 | 6.88 | −50.0 | −60.0 | 46.62 | 0.85 | 5.58 |
| O-LM | −50.0 | −60.0 | 7.91 | 3.90 | 29.83 | | | | | |
| Pyr | −50.0 | 0.0 | 17.42 | 0.68 | 3.05 | | | | | |
| PV+B | −50.0 | −60.0 | 5.71 | 0.42 | 3.08 | | | | | |
| CA3 | −50.0 | 0.0 | 27.10 | 2.35 | 9.13 | | | | | |
| ECIII | −50.0 | 0.0 | 31.82 | 3.38 | 10.47 | | | | | |

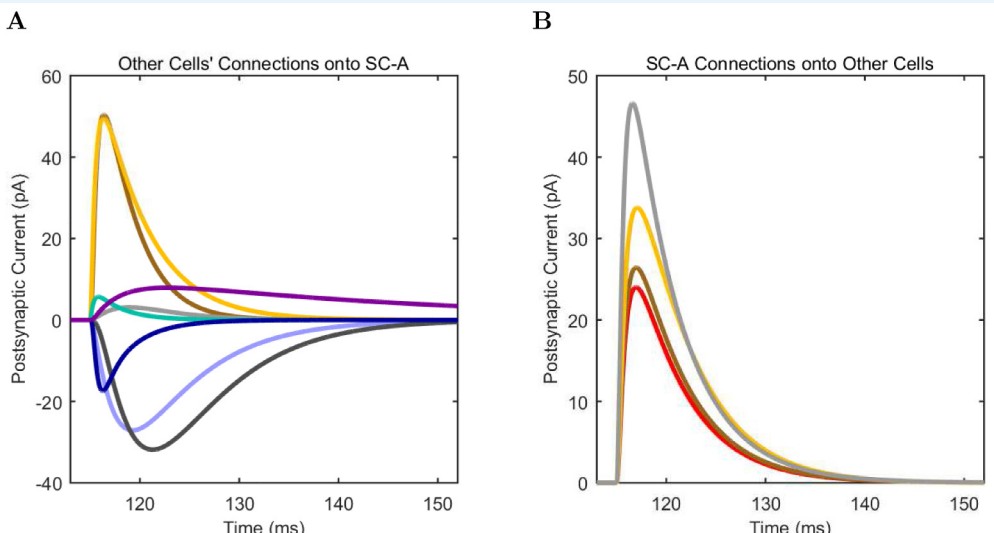

**Appendix 1—figure 22.** Connections onto (**A**) and (**B**) from model Schaffer Collateral-Associated cells, under voltage clamp at −50 mV with physiological reversal potentials.

## Inhibitory connectivity

Although the connectivity of the hippocampal CA1 network was assessed in *Bezaire and Soltesz (2013)*, detailed connectivity estimates were only made for pyramidal cells, while the convergence onto inhibitory cells and especially the inhibitory-inhibitory connections were only estimated at a high level due to lack of sufficiently specific experimental data. Here, we performed additional calculations and made use of additional data to arrive at specific estimates for each interneuron type.

First, we gathered previously published morphology data about each interneuron type. As there were no data available for ivy and neurogliaform cells, we performed the necessary experiments in our lab by filling ivy and neurogliaform cells in hippocampal CA1 slices from Wistar rats and then measuring their somatic area and dendrites. While we performed this experiment in slices, these cell types have a relatively compact morphology, allowing us to characterize a significant amount of their dendritic extent. The data for ivy and neurogliaform cells are available in *Appendix 1—table 63*. The somatic and dendritic lengths used for each interneuron type in this work are given in *Appendix 1—table 64*.

Next, we gathered data about the somatic and dendritic synaptic densities for each cell class (*Appendix 1—table 65*) and then multiplied the areas and lengths by their respective synaptic densities to arrive at the estimated excitatory and inhibitory synapses on the somata and dendrites of each cell types (*Appendix 1—table 66*). Finally, using our previous estimate of the excitatory and inhibitory boutons available for synapsing on interneurons in CA1 *Bezaire and Soltesz (2013)*, we evenly distributed the boutons available for synapsing on each neurite type in each layer across the available postsynaptic densities of that neurite type and transmitter type, so that each interneuron received approximately the same level of coverage of its incoming synapses, while respecting specific observations about interneuron connectivity, such as that CCK+ basket cells have never been observed to receive direct monosynaptic excitement from local CA1 pyramidal cells *Lee et al. (2010)* and O-LM cells receive almost all of their excitatory connections from local collaterals *Blasco-Ibáñez and Freund (1995)*. The final connectivity between each cell type, including interneurons, is given in the manuscript, *Table 1*.

**Appendix 1—table 63.** Measured dendritic lengths and somatic diameters for ivy and neurogliaform cells from the hippocampal CA1 area in Wistar rats, with calculation of somatic surface area included. Cells were characterized in our lab and their function has been reported in *Krook-Magnuson et al. (2011)*. Source Data available in *Appendix 1—table 1* - Source Data.

| Cell type | Cell name | # Sections | Dendritic length ($\mu$M)) | | | | Somatic dia-meter ($\mu$m) | Calculated synap-tic Area ($\mu m^2$) |
|---|---|---|---|---|---|---|---|---|
| | | | SO | SP | SR | SLM | | |
| Ivy | 0217–1 DAB 3_2_10 left slice | 1 | 129.2 | 64.5 | 1200.6 | 0 | 38.9 | 1188.5 |
| Ivy | 9 n23-7 DAB 12_16_09 left +middle slice | 2 | 0 | 0 | 2703.2 | 300.3 | 45.2 | 1604.6 |
| Ivy | 9 n23-6 DAB 06_10 left slice | 1 | 75.4 | 133.8 | 2115.4 | 0 | 36.6 | 1052.1 |
| Ivy | 9 n16-3 DAB 12_29_09 left slice | 1 | 0 | 0 | 1015.2 | 0 | 52.5 | 2164.8 |
| Ivy | Average | | 51.15 | 49.575 | 1758.6 | 75.075 | | 1502.5 |
| Neurogliaform | 9n 12–5 DAB 1_06_09 | 1 | 0 | 0 | 2097.7 | 525 | 34.4 | 929.4 |
| Neurogliaform | 91021 DAB 3_18_10 sec-ond,third, fourth from left slice | 3 | 0 | 0 | 1230.7 | 780.1 | 28 | 615.8 |
| Neurogliaform | 9d 8–3 DAB 1_15_10 left and right slice | 2 | 0 | 0 | 2328.2 | 1382.4 | 32.2 | 814.3 |
| Neurogliaform | Average | | 0 | 0 | 1885.5 | 895.8 | | 786.5 |

**Appendix 1—table 64.** Estimated or observed somatic area and dendritic length. Experimental observations of the dendritic length of broad interneuron classes were used as the basis for these estimations. The relative lengths for PV+ basket cells and axo-axonic cells were further differentiated based on experimental observations in region CA3 *Papp et al. (2013)*. The observations published in *Mátyás et al. (2004)* for CCK+ basket cells were also applied to the CCK+ Schaffer Collateral-Associated cells, based on the discussion in *Mátyás et al. (2004)*. The data for ivy and neurogliaform cells were based on measurements from filled cells from slices. Due to the compact nature of their morphology, especially the neurogliaform cells, the dendritic lengths within the slices were assumed to comprise most or all of the dendritic extents of those cells. See section below for raw data. The O-LM cell morphological measurements were taken from *Blasco-Ibáñez and Freund (1995)*.

| Interneuron | Soma area (100 $\mu m^2$) | Dendritic length ($\mu m^2$) | | | | | Reference |
|---|---|---|---|---|---|---|---|
| | | Total | SO | SP | SR | SLM | |
| Ivy | 1502 | 1934.4 | 51.15 | 49.575 | 1758.6 | 75.075 | See below |
| Neurogliaform | 786 | 2781.4 | 0 | 0 | 1885.5 | 895.8 | See below |
| PV+ basket | 3428 | 4359 | 1493 | 697 | 1877 | 292 | (*Papp et al., 2013*) |
| Bistratified | 1006 | 4347.75 | 1074.57 | 248.28 | 2369.24 | 655.66 | (*Gulyás et al., 1999*) |
| Axo-axonic | 2329 | 2825 | 570 | 659 | 1259 | 337 | (*Papp et al., 2013*) |
| CCK+ basket | 966 | 6338.31 | 1213.92 | 310.61 | 3522.6 | 1291.18 | (*Mátyás et al., 2004*) |
| SCA | 966 | 6338.31 | 1213.92 | 310.61 | 3522.6 | 1291.18 | (*Mátyás et al., 2004*) |
| O-LM | 3007.78 | 4165.68 | 4165.68 | 0 | 0 | 0 | (*Blasco-Ibáñez and Freund, 1995*) |

**Appendix 1—table 65.** The synaptic densities (# boutons per 100 $\mu$m of dendritic length, or # boutons per 100 $\mu m^2$ of somatic area) on the soma and dendrites of PV cells, given in *Gulyás et al. (1999)*, were applied to the axo-axonic, PV+ basket, and bistratified cells. The synaptic densities of CCK+ cells *Mátyás et al. (2004)* were applied to the CCK+ basket and the Schaffer Collateral-Associated cells. For the ivy, neurogliaform, and O-LM cells, there were not sufficient experimental data published to constrain the synaptic density, and so an average of all synaptic densities for all cell classes was computed and applied to these cell types.

| | | | Dendritic | | | | | | | | |
|---|---|---|---|---|---|---|---|---|---|---|---|
| | Somatic | | SO | | SP | | SR | | SLM | | |
| Reference | Exc | Inh | Exc | Inh | Exc | Inh | Exc | Inh | Exc | Inh | Ref |
| Ivy | 21.8 | 16.1 | 172.2 | 23.7 | 163.3 | 38.5 | 193.8 | 25.3 | 97.4 | 31.7 | Calc. from average |
| Neurogliaform | 21.8 | 16.1 | 172.2 | 23.7 | 163.3 | 38.5 | 193.8 | 25.3 | 97.4 | 31.7 | Calc. from average |
| PV+ basket | 40.7 | 18.1 | 342.5 | 19.2 | 345 | 16.1 | 371.2 | 18.3 | 132.2 | 28.6 | (*Gulyás et al., 1999*) |
| Bistratified | 40.7 | 18.1 | 342.5 | 19.2 | 345 | 16.1 | 371.2 | 18.3 | 132.2 | 28.6 | (*Gulyás et al., 1999*) |
| Axo-axonic | 40.7 | 18.1 | 342.5 | 19.2 | 345 | 16.1 | 371.2 | 18.3 | 132.2 | 28.6 | (*Gulyás et al., 1999*) |
| CCK+ basket | 3.4 | 16.1 | 84.3 | 32.5 | 52.7 | 87.4 | 82 | 37.8 | 86.5 | 58.8 | (*Mátyás et al., 2004*) |
| SCA | 3.4 | 16.1 | 84.3 | 32.5 | 52.7 | 87.4 | 82 | 37.8 | 86.5 | 58.8 | (*Mátyás et al., 2004*) |
| O-LM | 21.8 | 16.1 | 172.2 | 23.7 | 163.3 | 38.5 | 193.8 | 25.3 | 97.4 | 31.7 | Calc. from average |

**Appendix 1—table 66.** Estimated numbers of excitatory and inhibitory synapses on each cell type, calculated by multiplying the somatic area or dendritic length by the respective synaptic density. About 20% of synapses onto O-LM cells are GABAergic, while at least 60% are from local excitatory collaterals *Kispersky et al. (2012)*. Therefore, we conserved the total (inhibitory + excitatory) synaptic density of O-LM cells as calculated previously, but set 20% of that total to be inhibitory and the rest to be excitatory synapses.

| Ref | Somatic | | SO | | SP | | Dendritic SR | | SLM | | Total | | Ref |
|---|---|---|---|---|---|---|---|---|---|---|---|---|---|
| | Exc | Inh | Exc | Inh | Exc | Inh | Exc | Inh | Exc | Inh | Exc | Inh | |
| Ivy | 326.9 | 242.3 | 88 | 12 | 82 | 19 | 3408 | 445 | 73 | 24 | 3651 | 500 | Calculated |
| Neurogliaform | 171.1 | 126.8 | 0 | 0 | 0 | 0 | 3654 | 477 | 873 | 284 | 4527 | 761 | Calculated |
| PV+ basket | 1395.2 | 620.1 | 4230 | 237 | 866 | 40 | 9449 | 466 | 840 | 182 | 15385 | 925 | Calculated |
| Bistratified | 409.4 | 182 | 3681 | 206 | 856 | 40 | 8796 | 434 | 867 | 188 | 14200 | 868 | Calculated |
| Axo-axonic | 947.9 | 421.3 | 1615 | 90 | 819 | 38 | 6338 | 313 | 969 | 210 | 9741 | 651 | Calculated |
| CCK+ basket | 32.8 | 155.7 | 1024 | 394 | 164 | 271 | 2887 | 1332 | 1117 | 759 | 5192 | 2756 | Calculated |
| SCA | 32.8 | 155.7 | 1024 | 394 | 164 | 271 | 2887 | 1332 | 1117 | 759 | 5192 | 2756 | Calculated |
| O-LM | 654.6 | 485.2 | 6527 | 1632 | 0 | 0 | 0 | 0 | 0 | 0 | 6527 | 1632 | Calculated |

**Appendix 1—table 67.** Ion channels included in the model. GHK: based on Goldman-Hodgkin-Katz equation; Q-O: quasi-ohmic; Hyperpol.-act: Hyperpolarization-activated; Nucleo.-gated: Nucleotide-gated; voltage-act.: voltage activated; voltage-dep.: voltage dependent; Calcium-act.: Calcium-activated; Pyr.: pyramidal; NGF: neurogliaform; dist.: distal; prox.: proximal.

| Ion | | Model | | | | | | | | | |
|---|---|---|---|---|---|---|---|---|---|---|---|
| Channel | Description | Type | Pyramidal | Axo-axonic | Bistratified | CCK+ Basket | Ivy | Neurogliaform | O-LM | PV+ Basket | S.C.-Assoc. |
| $Ca_{v,L}$ | L-type Calcium | GHK | • | • | • | • | • | • | | • | • |
| $Ca_{v,N}$ | N-type Calcium | Q-O | • | • | • | • | • | • | | • | • |
| HCN | Hyperpol.-act, Cyclic Nucleo.-gated | Q-O | | | | • | | | | | • |
| $HCN_{OLM}$ | Hyperpol.-act, Cyclic Nucleo.-gated for O-LM cells | Q-O | | | | | | | • | | |
| $HCN_p$ | Hyperpol.-act, Cyclic Nucleo.-gated for Pyr. cells | Q-O | • | | | | | | | | |
| $K_{Ca,S}$ | Small (SK) Calcium-activated potassium | Q-O | • | • | • | • | • | • | • | • | • |
| $K_{dr,fast}$ | Fast delayed rectifier potassium | Q-O | • | • | • | • | | | • | • | • |
| $K_{dr,fast,ngf}$ | Fast delayed rectifier potassium for NGF-family cells | Q-O | | | | | • | • | | | |
| $K_{dr,p}$ | Delayed rectifier potassium for Pyr. cells | Q-O | • | | | | | | | | |
| $K_{v,A}$ | A-type voltage-act. potassium | Q-O | • | • | • | • | | | | • | • |
| $K_{v,A,dist,p}$ | A-type voltage-act. potassium for dist. Pyr. dendrites | Q-O | • | | | | | | | | |
| $K_{v,A,ngf}$ | A-type voltage-act. potassium for NGF-family cells | Q-O | | | | | • | • | | | |
| $K_{v,A,olm}$ | A-type voltage-act. potassium for O-LM cells | Q-O | | | | | | | • | | |
| $K_{v,A,prox,p}$ | A-type voltage-act. potassium for prox. Pyr. dendrites | Q-O | • | | | | | | | | |
| $K_{v,Ca,B}$ | Big (BK) Calcium-act., voltage-dep. potassium | Q-O | | • | • | • | • | • | | • | |
| $K_{v,Group}$ | Multiple slower voltage-dep. potassium | Q-O | | | • | • | | • | | • | |
| leak | Leak | Q-O | • | • | • | • | • | • | • | • | • |
| $Na_v$ | Voltage-dep. sodium | Q-O | • | • | | | | | • | • | |
| $Na_{v,bis}$ | Voltage-dep. sodium for bistratified cells | Q-O | | | • | | | | | | |
| $Na_{v,cck}$ | Voltage-dep. sodium for CCK+ cells | Q-O | | | | • | | | | | • |
| $Na_{v,ngf}$ | Voltage-dep. sodium for NGF-family cells | Q-O | | | | | • | • | | | |

*Appendix 1—table 67 continued on next page*

*Appendix 1—table 67 continued*

| Ion | | Model | | | | | | | | | | | |
|-----|-------------|------|------|-----------|-----------|-------------|------------|-----|-------------|-----|------|------------|-----------|
| Channel | Description | Type | Pyramidal | Axo-axonic | Bistratified | CCK+ Basket | Ivy | Neurogliaform | O-LM | PV+ Basket | S.C.-Assoc. |
| $Na_{v,p}$ | Voltage-dep. sodium for Pyr. cells | Q-O | • | | | | | | | | |
| pas | Leak | Q-O | • | | | | | | | | |

# Ion channel descriptions

*Appendix—table 67* lists which model ion channels are found in each model cell type. The channels are further described below, summarized from *Bezaire (2015)*, and their equations are included as well. The activation/inactivation curves and current-voltage relations are reproduced here from *Bezaire (2015)*.

## Calcium channels

The calcium channels were adapted from previous Soltesz Lab models (*Santhakumar et al., 2005*; *Dyhrfjeld-Johnsen et al., 2007*; *Morgan and Soltesz, 2008*) and include an L-type and N-type channel (*Appendix 1—figure 23*); the L-type channel does not inactivate and the N-type channel inactivates.

### Ca$_{v,L}$ Channel

*Jaffe et al. (1994)* developed this channel based on activation data from CA1 and CA3 hippocampal neurons in adult guinea pigs, at room temperature. It has been further used in many other models implemented by Migliore. The voltage of half-activation was shifted by $-10$ mV, accounting for ionic differences in the experimental preparation compared to the model condition. It uses the GHK equation to calculate the driving force through the channel, allowing a mild dependence on calcium concentration.

### Ca$_{v,N}$ Channel

*Jaffe et al. (1994)* also developed this channel, using the same preparation as for the Ca$_{v,L}$ channel. *Aradi and Holmes (1999)* then modified the channel code, replacing the GHK calculation with a quasi-ohmic calculation of the driving force. In addition, its behavior was altered somewhat compared to previous implementations such as *Morgan and Soltesz (2008)* and *Santhakumar et al. (2005)*. Their implementations contained a typo in the channel definition that caused its equations to differ from those presented in *Aradi and Holmes (1999)*, and had the effect of reducing the conductance of the channel below its intended magnitude.

At high levels of activation, the channel conductance decreases slightly (*Appendix 1—figure 23*), a behavior that resulted from replacing of the GHK calculation with a quasi-ohmic expression. However, it may not have too large of an effect since it only happens at very depolarized potentials, potentials that are likely to be achieved only at the peak of a spike.

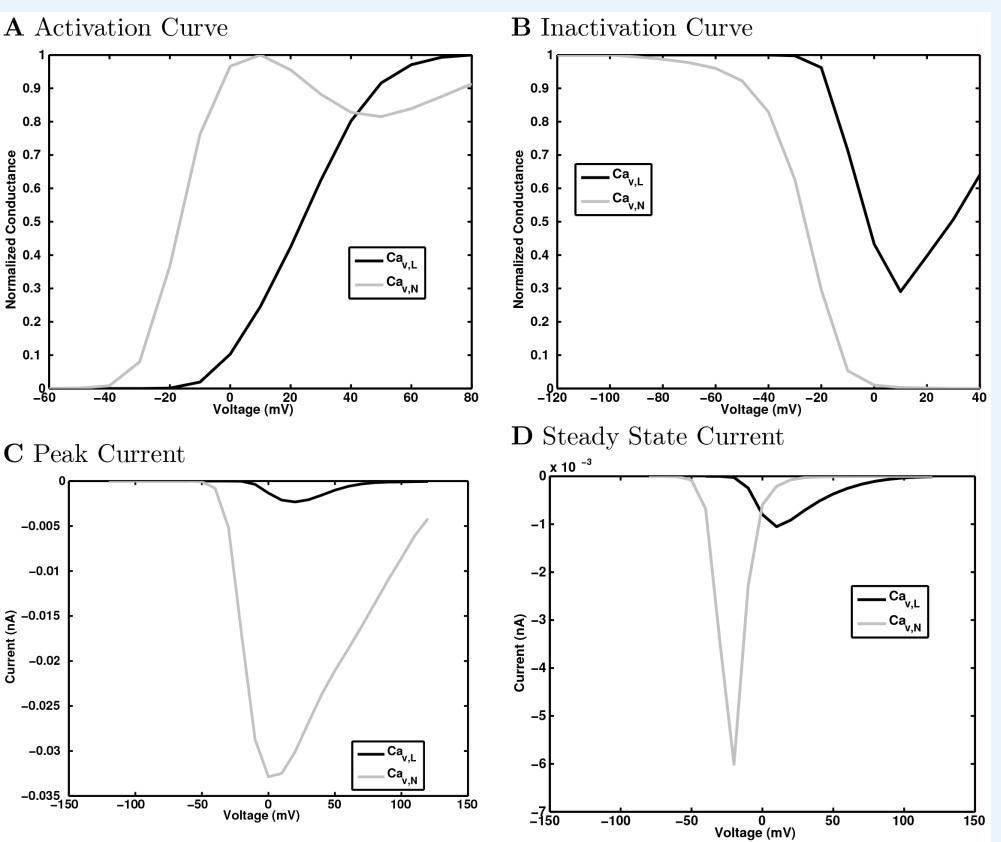

**Appendix 1—figure 23.** Calcium channel currents.

## HCN channels

The characteristic behavior of HCN channels, their hyperpolarized voltage-dependent activation is captured in these models, but not their cyclic-nucleotide gating. Because they are hyperpolarization-activated, the protocol used to characterize the inactivating of the other channels was used to characterize the activation of the HCN channels. In *Appendix 1—figure 24*, the differing behavior of the HCN channels can be seen.

## HCN Channel

The HCN channel model was based on experiments carried out in CA1 pyramidal cells of Sprague-Dawley rats at room temperature *Chen et al. (2001)*. The original channel model included fast and slow components and used separate, artificial ion definitions for each. We only retained the slow component as the inclusion of the fast component caused a non-physiological, oscillating sag when included in cells. We also reduced the voltage dependence of the slow component slightly to further decrease the oscillation of the sag.

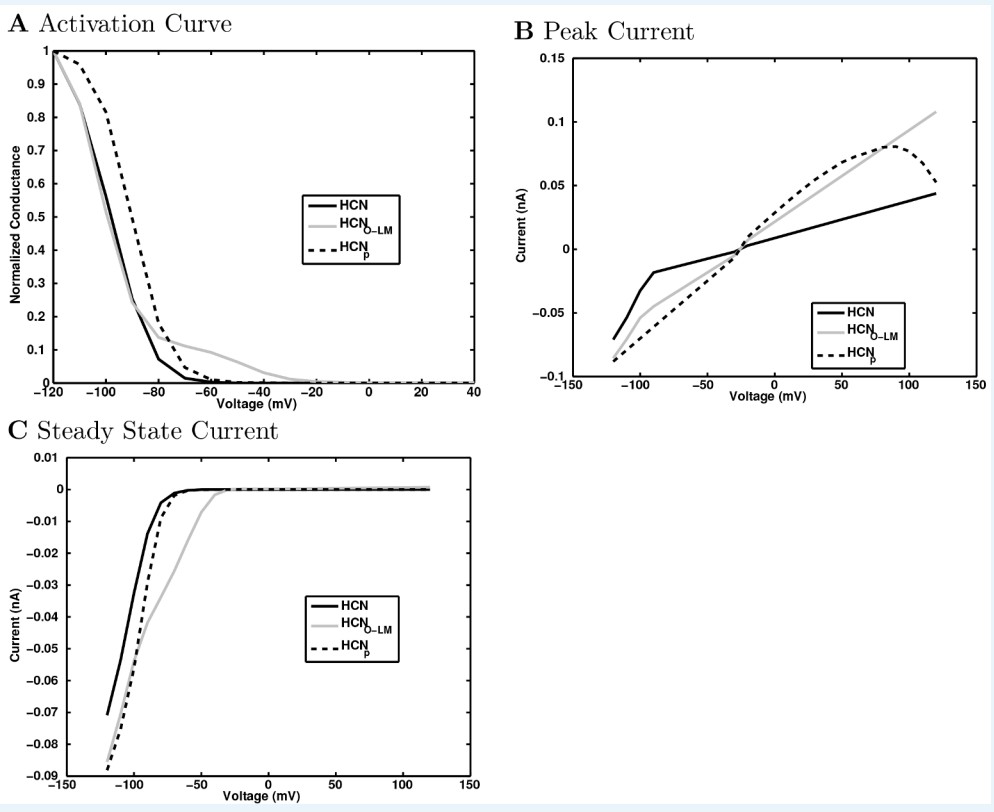

**Appendix 1—figure 24.** HCN channel currents.

## HCN$_{OLM}$ Channel

*Saraga et al. (2003)* developed this channel model based on data from young Sprague-Dawley rats at a warm room temperature, and *Cutsuridis et al. (2010)* included it in a model O-LM cell, which we incorporated into our model.

## HCN$_p$ Channel

*Cutsuridis et al. (2010)* developed this channel model based experimental data from adult Sprague-Dawley rats at $23^o$ or $33^o$ (*Magee, 1998*) and included it in a model pyramidal cell.

## Potassium channels

We included several potassium channels: delayed rectifier, A-type potassium channels, calcium-dependent potassium channels, and leak channels. We developed multiple variations, which enabled us to tune their thresholds and voltage dependence to the voltage-dependent behavior of the cells into which we placed them.

## Delayed rectifier potassium channels

The delayed rectifier models had a voltage-dependent activation component but not an inactivating component.

In *Appendix 1—figure 25*, the differing behavior of the delayed rectifiers can be seen.

### K$_{dr,fast}$ channel

**Yuen and Durand (1991)** originally implemented this model, based on another model of a fast delayed rectifier in a squid axon. They adjusted its parameters so their model cell action potential wave form matched that produced by an experimental mouse cell. **Aradi and Holmes (1999)** later modified the model to shift the voltage dependence slightly.

### K$_{dr,fast,ngf}$ channel

To better fit the behavior of experimental neurogliaform cells, we shifted by $-10$ mV the voltage dependence of the K$_{dr,fast}$ channels within the neurogliaform and ivy cells, which usually have a higher threshold than other cells.

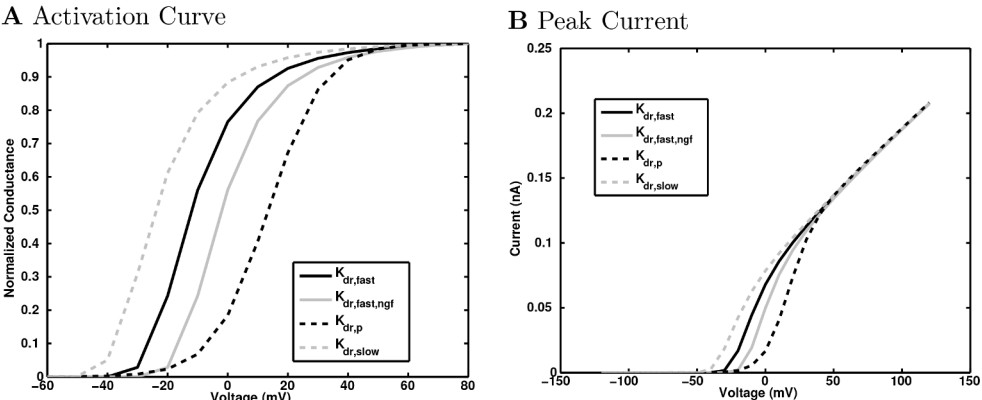

**Appendix 1—figure 25.** Delayed rectifier potassium channel currents.

### K$_{dr,p}$ channel

This channel, implemented by Migliore based on experimental data from hippocampal cells in rat pups at room temperature (**Klee et al., 1995**), was used in the pyramidal cell model we included in our model (**Poolos et al.,2002**).

## A-type potassium channels

A-type potassium channels are transient and quickly-inactivating; they activate near the action potential threshold, delaying action potential onset, increasing the action potential threshold, and even modulating early repolarization after an action potential(**Storm, 1990**).

### K$_{v,A}$ Channel

**Migliore et al. (1995)** developed this channel model, starting with the equations of **Borg-Graham (1991)** and modifying them to account for **Ficker and Heinemann (1992)** and **Numann et al. (1987)** to get the burst behavior they were investigating.

### K$_{v,A,dist,p}$ Channel

**Migliore et al. (1995)** also developed this channel for use in the CA1 pyramidal cell model of **Poolos et al. (2002)**, based on rat hippocampal pyramidal cell data from **Klee et al. (1995)**.

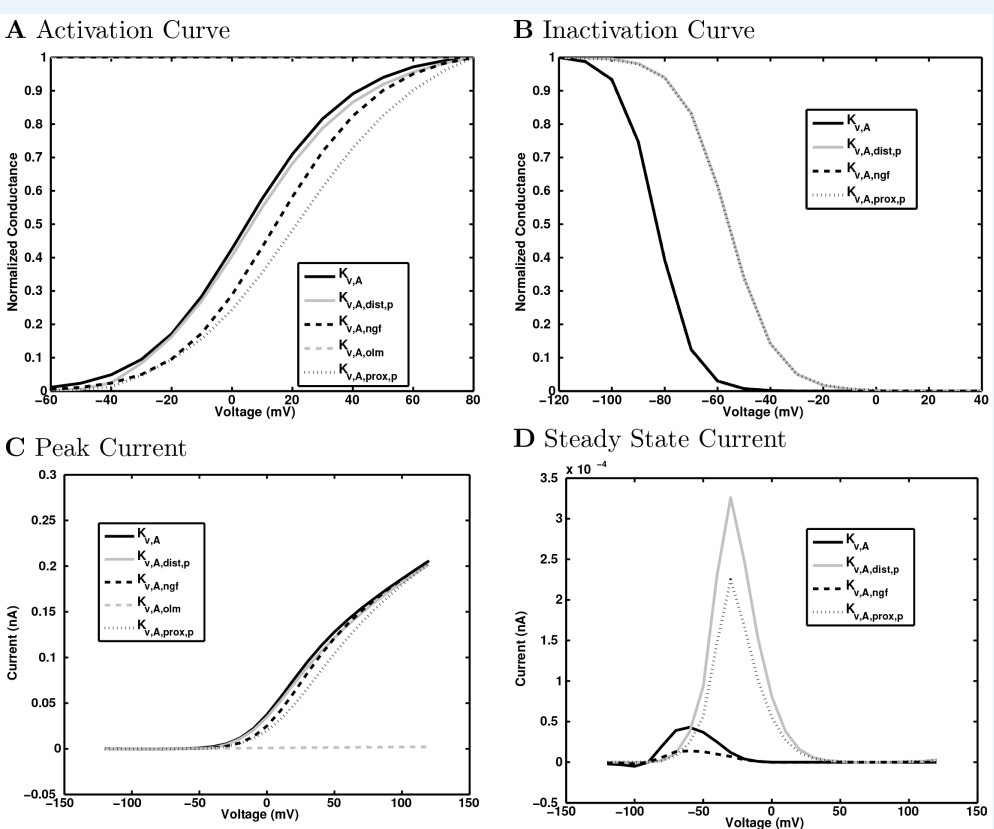

**Appendix 1—figure 26.** A-type potassium channel currents.

### $K_{v,A,prox,p}$ channel

***Migliore et al. (1995)*** also developed this channel model, which differs slightly from the $K_{v,A,dist,p}$ channel model in timing and voltage.

### $K_{v,A,ngf}$ channel

We modified the activation and inactivation equations of the KvA A-type channel developed by ***Migliore et al. (1995)*** for use in neurogliaform and ivy cells. We offset the voltage dependence by + 10 mV to better fit the high-threshold neurogliaform cell family.

### $K_{v,A,olm}$ channel

***Saraga et al. (2003)*** developed this channel based on experimental data from the McBain lab and others, and ***Cutsuridis et al. (2010)*** used it in the O-LM cell model that we included in our model.

## Other potassium channels

Some of the potassium channels included in the model were not voltage dependent in the same way as the above ones. The leak and KCaS channels are not voltage activated nor inactivating. The KvCaB channel was voltage-dependent, but was also $Ca^{2+}$ gated. KvGroup was voltage-dependent activation but had no inactivating component.

## K$_{Ca,S}$ channel

The model of this calcium-activated potassium channel (known as the small or 'SK' channel), was developed by *Yuen and Durand (1991)* and modified by *Aradi and Holmes (1999)* based on data from *Beck et al. (1997)*, *Latorre et al. (1989)*, *Sah (1996)*, and *Lancaster et al. (1991)*.

## K$_{v,Ca,B}$ Channel

This big ('BK') potassium channel, both voltage dependent and calcium-gated, was implemented by *Migliore et al. (1995)* based on a model from *Moczydlowski and Latorre (1983)* and has been modified somewhat.

The behavior of these other channels is shown as a function of voltage (*Appendix 1—figure 27*) and as a function of internal Ca$^{2+}$ concentration for the calcium-gated potassium channels (*Appendix 1—figure 28*).

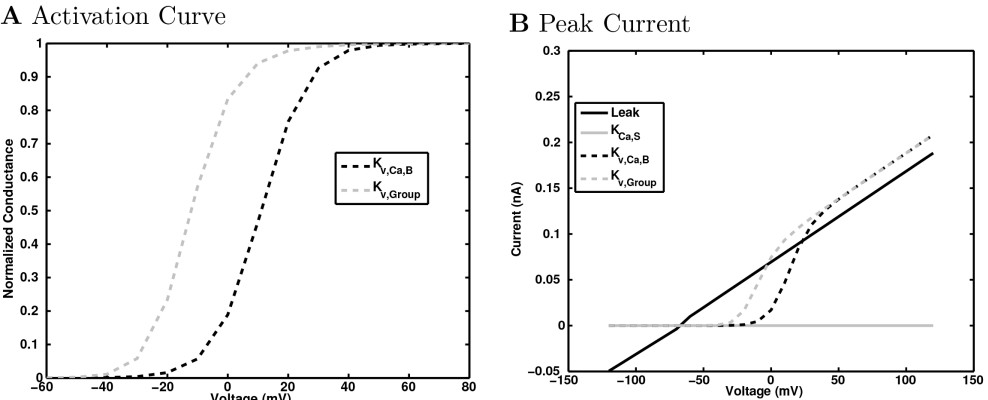

**Appendix 1—figure 27.** Other potassium channel currents. Because they didn't have a voltage-sensitive inactivation component, only the activation curve, which is equivalent to the IV Peak curve, need be shown here.

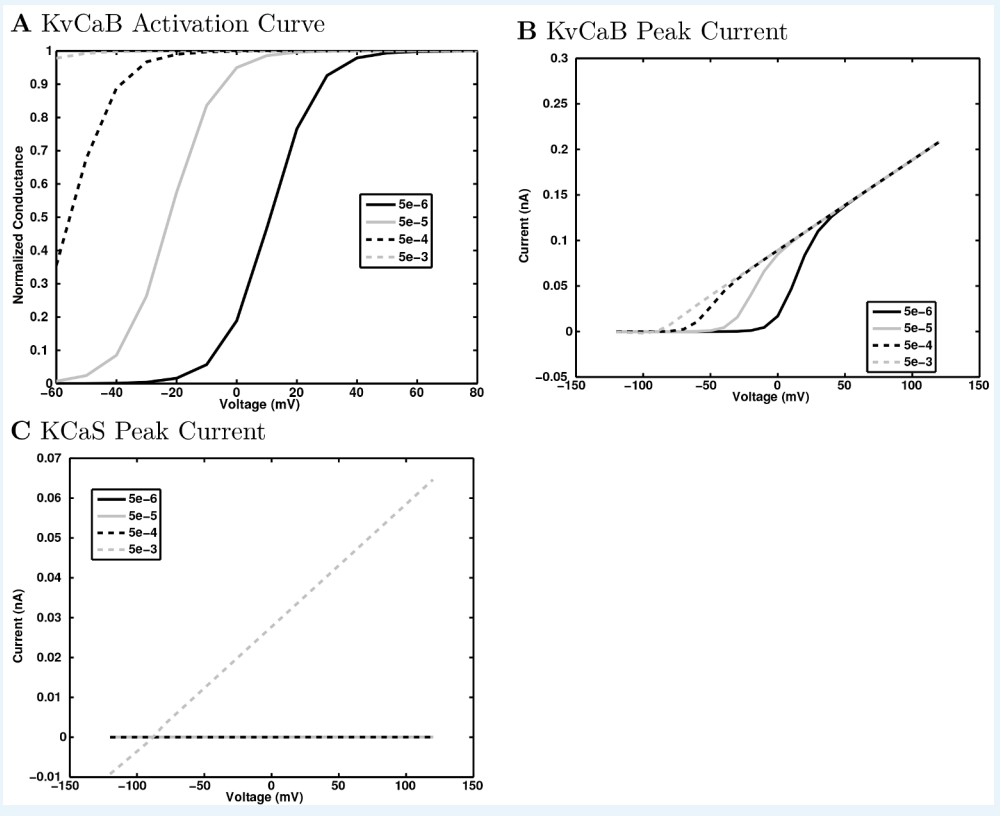

**Appendix 1—figure 28.** Calcium-dependent potassium channel dependence on calcium concentration. (**a**) The normalized conductance of the channels are plotted as a function of test voltage step and calcium concentration. (**b**) and (**c**) The current-voltage relation is shown at several calcium concentrations for (**b**) KvCaB channel and (**c**) KCaS channel. Note that the KCaS channel is only active at the highest calcium concentration and is not dependent on voltage (although the voltage continues to set the driving force) when it is active.

### $K_{v,Group}$ channel

We implemented a new potassium channel model, starting with the fast delayed rectifier potassium channel, and adjusting the parameters to match the channel behavior of *Lien and Jonas (2003)* which was presented as a model of the Kv3.1b channel. However, the methods used in *Lien and Jonas (2003)* are likely to have included multiple potassium channel types in their channel characterization: they added 300 uM of 4-AP, billed as a low enough dose that it would block only Kv3.1b channels, and subtracted the intracellular potential recording of the cell in the presence of that blocker from the control recording and attributed that entire difference to Kv3.1b. However, it is possible to block other potassium channel types with doses of 4-AP as low as 5-10 $\mu$M *Campanac et al. (2013)*, so two or more potassium channel types were likely blocked in *Lien and Jonas (2003)*, meaning that more than one potassium channel type contributes to the dynamics of the model channel fit by *Lien and Jonas (2003)*, so we called it 'KvGroup'.

### Leak channel

The leak channel is a very simple, quasi-ohmic model component that employs a non-specific current with a reversal potential set near but depolarized from the potassium reversal potential. Its conductance is inversely related to the membrane resistance.

## Sodium channels

As with the potassium channels, we implemented variations of the fast activating, fast-inactivating sodium channel to enable each cell type to achieve a physiologically realistic threshold.

In *Appendix 1—figure 29*, the differing behavior of the sodium channels can be seen.

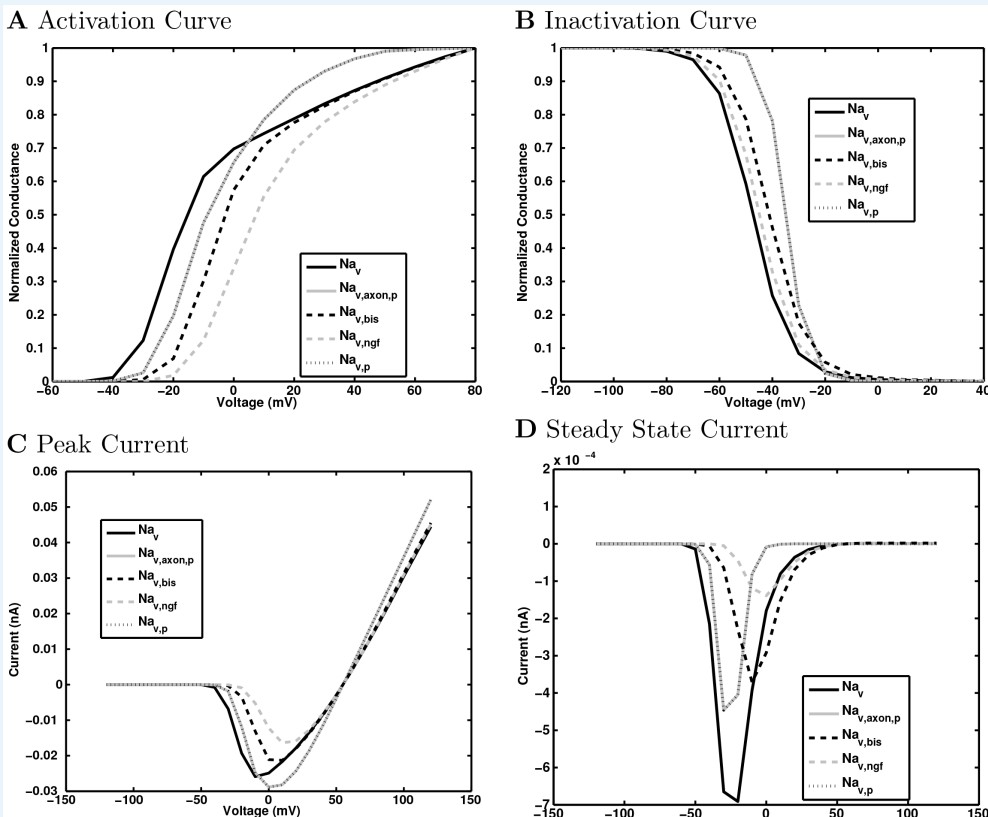

**Appendix 1—figure 29.** Sodium channel voltage dependence. The normalized conductance of the sodium channel is plotted as (**a**) a function of test voltage step to show activation and (**b**) as a function of holding voltage prior to the test step to show inactivation.

## Na$_v$ channel

*Yuen and Durand (1991)* originally developed this channel model, which was then modified by *Aradi and Holmes (1999)*.

## Na$_{v,bis}$ channel

We depolarized the threshold of the ch_Nav channel slightly, to better fit the bistratified cell behavior. Additionally, the voltage dependence of its activation has been shifted by $-5$ mV and inactivation by $-3$ mV. The coefficients have also been adjusted.

## Na$_{v,cck}$ channel

This channel is modified from ch_Nav to have a higher threshold and a slow inactivation component to help realize the spike adaptation observed in experimental CCK+ cells. The voltage dependence of its activation has been shifted by $-1$ mV and its coefficients have been adjusted as well.

### Na$_{v,ngf}$ channel

We modified this channel from ch_Nav, giving it a much higher threshold suitable for neurogliaform family cells. We offset its activation and inactivation voltage dependencies have been offset by about $-19$ mV, and the coefficients have been slightly modified as well.

### Na$_{v,cck}$ channel

We modified this channel from the ch_Nav model to have slightly different kinetics, mainly adjusting the coefficients (with only minor adjustments to the voltage dependence) to achieve more realistic CCK+ cell behavior.

### Na$_{v,p}$ channel

*Migliore et al. (1999)* developed this channel, using it in their pyramidal cell model that we incorporated into our model *Poolos et al. (2002)*. We only implemented the fast inactivating option of this channel, though a slow option was also available in the original implementation *Poolos et al. (2002)*.

## Ion channel equations

For each ion channel, the equations used to compute the conductance are laid out below, with the following conventions:

- $I_{ion}$ gives the specific current/area through the channels (of that type) within a small area of the membrane; it is multiplied by the area of the neuron section it covers to arrive at a total current estimate.
- $E_{ion}$ gives the reversal potential of the channel
- The ions flowing through HCN and leak channels (usually a combination of sodium and potassium) are not explicitly specified in the code, so instead they are denoted with an *H* or *leak*.
- The units of variables and quantities are listed to their right in blue text
- T $^oC$ = temperature of model, $34^oC$ in this network
- T $K$ = temperature of model converted to Kelvins, $307.15K$ in this network (or $307.16K$ in some channels that convert the temperature differently).
- $g_{max\frac{nS}{cm^2}}$ = max. conductance, set when the cell is defined
- $dt$ ms = time step
- $t_{inc}\ ms = -dt\ ms * q_{10}$
- $v$ mV = membrane potential local to the ion channel
- $F$ kC = $96.487 \frac{kC}{mol}$ (Faraday's constant)
- Temperature may be listed with units of Celsius ($^oC$) or units of Kelvin ($K$) depending on what was used in the original code equation. For equations in this appendix section listing units of Kelvin, the equation has been simplified so as not to show the conversion of temperature from units of Celsius to Kelvin, even though this conversion takes place explicitly in the equation of the code file itself. Ex: equations in the code files including the term $273.15 + T$ (or $273.16 + T$ if converting relative to the freezing point of water rather than the more correct triple point of water), representing the conversion from Celsius to Kelvin ($1K - 1^oC = 273.15\mathrm{degrees}$), are instead displayed with $TK$ in the equations of this section.
- The universal gas constant $R$, commonly listed in units of Joules per K*mol, is listed here in the alternative units of Coulomb * Volt per K*mol ($CVK^{-1}mol^{-1}$) to be consistent with the units used for the other terms in the equations.

Unless otherwise specified, for each channel, the temperature dependence can be given by:

$$q_{10} = 3^{\frac{T - 34\,^oC}{10\,^oC}} \tag{4}$$

The equations specific to each channel are given below. For each channel, first any equations for constants, which only need to be solved once, are listed. Next, equations that need to be solved each time step are listed. Because of the way the NEURON simulator works, some equations may even be solved multiple times per time step. Also, at each instance of the ion channel (at each point within each section of the cell where the ion channel is inserted), the equations need to be solved again.

## CavL

The following equations are solved each time step of simulation, as explained above, using the membrane potential $v$ in the section of the membrane where the ion channel is located:

$$g \frac{S}{cm^2} = g_{max} \frac{S}{cm^2} * m^2 * h \tag{5}$$

$$I_L \frac{mA}{cm^2} = g \frac{S}{cm^2} * ghk(v\,mV, [Ca^{2+}]_i\,mM, [Ca^{2+}]_o mM) \tag{6}$$

The activation of channel conductance is represented by $m$ (v mVoltage-dependent activation) and $h$ (calcium-dependent activation). The driving force through the channel, $ghk$, is calculated using the Goldman-Hodgkin-Katz (GHK) equation. The values of ghk, m and h are also solved each time step, by calculating the following equations.

For $ghk$:

$$f\,mV = \frac{\frac{25\,mV}{293.15\,K} * T\,K}{2} \tag{7}$$

$$ghk\,mV = -f\,mV * \left(1 - \left(\frac{[Ca^{2+}]_i\,mM}{[Ca^{2+}]_o\,mM}\right) * exp\left(\frac{v\,mV}{f\,mV}\right)\right) * \frac{\frac{v\,mV}{f\,mV}}{exp(\frac{v\,mV}{f\,mV}) - 1} \tag{8}$$

For $m$:

$$\alpha\,ms^{-1} = \frac{15.69\,ms^{-1} * (-1.0 * v + 81.5\,mV)}{exp\left(\frac{-1.0 * v + 81.5\,mV}{10.0\,mV}\right) - 1.0} \tag{9}$$

$$\beta\,ms^{-1} = 0.29\,ms^{-1} * exp\left(\frac{-v}{10.86\,mV}\right) \tag{10}$$

$$\tau_m\,ms = \frac{1}{(\alpha\,ms^{-1} + \beta\,ms^{-1})} \tag{11}$$

$$m_\infty = \alpha\,ms^{-1} * \tau_m\,ms \tag{12}$$

$$\frac{dm}{dt\,ms} = \frac{m_\infty - m}{\tau_m\,ms} \tag{13}$$

For $h$:

$$h = \frac{0.001\,mM}{0.001mM + [Ca^{2+}]_i\,mM} \tag{14}$$

## CavN

The following equations are solved each time step:

$$g \; \frac{S}{cm^2} = g_{max} \; \frac{S}{cm^2} * c^2 * d \tag{15}$$

$$I_{Ca} \; \frac{mA}{cm^2} = g \; \frac{S}{cm^2} * (v \; mV - E_{Ca} \; mV) \tag{16}$$

The values of $c$ and $d$ are also solved each time step, by calculating the following equations:

$$\alpha_c \; ms^{-1} = -0.19 \; ms^{-1} * \frac{v \; mV - 19.88 \; mV}{exp\left(\frac{v \; mV - 19.88 \; mV}{10 \; mV}\right) - 1} \tag{17}$$

$$\beta_c \; ms^{-1} = 0.046 \; ms^{-1} * exp\left(\frac{-v}{20.73 \; mV}\right) \tag{18}$$

$$\tau_c \; ms = \frac{1}{\alpha_c \; ms^{-1} + \beta_c \; ms^{-1}} \tag{19}$$

$$c_\infty = \frac{\alpha_c \; ms^{-1}}{\alpha_c \; ms^{-1} + \beta_c \; ms^{-1}} \tag{20}$$

$$\alpha_d \; ms^{-1} = 0.00016 \; ms^{-1} * exp\left(\frac{-v}{48.4 \; mV}\right) \tag{21}$$

$$\beta_d \; ms^{-1} = \frac{1}{exp\left(\frac{-v + 39 \; mV}{10 \; mV}\right) + 1} \tag{22}$$

$$\tau_d \; ms = \frac{1}{\alpha_d \; ms^{-1} + \beta_d \; ms^{-1}} \tag{23}$$

$$d_\infty = \frac{\alpha_d \; ms^{-1}}{\alpha_d \; ms^{-1} + \beta_d \; ms^{-1}} \tag{24}$$

$$c_{exp} = 1 - exp\left(\frac{t_{inc} \; ms}{\tau_c \; ms}\right), \qquad d_{exp} = 1 - exp\left(\frac{t_{inc} \; ms}{\tau_d \; ms}\right) \tag{25}$$

$$c = c + c_{exp} * (c_\infty - c), \qquad d = d + d_{exp} * (d_\infty - d) \tag{26}$$

## HCN

Since the HCN channel conducts a mixture of sodium and potassium, the reversal potential $E_H$ is set to lie between their reversal potentials, usually around $-30$ mV. The following equations are solved each time step:

$$g \; \frac{S}{cm^2} = g_{max} \; \frac{S}{cm^2} * h^2 \tag{27}$$

$$I_H \; \frac{mA}{cm^2} = g \; \frac{S}{cm^2} * (v \; mV - E_H \; mV) \tag{28}$$

The value of $h$ is also solved each time step, by calculating the following equations:

$$\tau_{slow} \ ms = \left(80 * 1.5 \ ms + \frac{.75 * 172.7 \ ms}{1 + exp(\frac{-(v \ mV + 59.3 \ mV)}{-0.83 \ mV})}\right) * \frac{1}{q_{10}} \tag{29}$$

$$h_\infty = \frac{1}{1 + exp\left(\frac{v \ mV + 91 \ mV}{10 \ mV}\right)} \tag{30}$$

$$\frac{dh}{dt \ ms} = \frac{h_\infty - h}{\tau_{slow} \ ms} \tag{31}$$

## HCNolm

Since the HCN channel conducts a mixture of sodium and potassium, the reversal potential $E_H$ is set to lie between their reversal potentials, usually around $-30$ mV. The following equations are solved each time step of simulation:

$$g \ \frac{S}{cm^2} = g_{max} \ \frac{S}{cm^2} * r \tag{32}$$

$$I_H \ \frac{mA}{cm^2} = g \ \frac{S}{cm^2} * (v \ mV - E_H \ mV) \tag{33}$$

The value of $r$ is also solved each time, by calculating the following equations:

$$r_\infty = \frac{1}{1 + exp\left(\frac{v \ mV + 84.1 \ mV}{10.2 \ mV}\right)} \tag{34}$$

$$\tau_r = 100 \ ms + \frac{1 \ ms}{exp(-17.9 \ mV - 0.116 * v \ mV) + exp(-1.84 \ mV + 0.09 * v \ mV)} \tag{35}$$

$$r_{exp} = 1 - exp\left(\frac{-dt \ ms}{\tau_r \ ms}\right) \tag{36}$$

$$\frac{dr}{dt \ ms} = \frac{r_\infty - r}{\tau_r \ ms} \tag{37}$$

## HCNp

A different temperature dependence was calculated for this channel than the default calculation specified at the beginning of the section:

$$q_t = 4.5^{\frac{T-33 \ ^oC}{10 \ ^oC}} \tag{38}$$

As with the other HCN channels, the reversal potential $E_H$ was set to lie between the sodium and potassium reversal potentials, around $-30$ mV. Then, the following equations are solved each time step:

$$g \ \frac{S}{cm^2} = g_{max} \ \frac{S}{cm^2} * l \tag{39}$$

$$I_H \ \frac{mA}{cm^2} = g \ \frac{S}{cm^2} * (v \ mV - E_H \ mV) \tag{40}$$

The value of $l$ is also solved each time step, by calculating the following equations:

$$\alpha = exp(0.0378\ mV^{-1} * 2.2 * (v\ mV + 75\ mV)) \tag{41}$$

$$\beta = exp(0.0378\ mV^{-1} * 2.2 * 0.4 * (v\ mV + 75\ mV)) \tag{42}$$

$$l_\infty = \frac{1}{1 + exp(0.0378\ mV^{-1} * 4 * (v\ mV + 90\ mV))} \tag{43}$$

$$\tau_l\ ms = \frac{\beta}{q_t * 0.011\ ms^{-1} * (1 + \alpha)} \tag{44}$$

$$\frac{dl}{dt\ ms} = \frac{l_\infty - l}{\tau_l\ ms} \tag{45}$$

## KCaS

The following equations are solved each time step of simulation:

$$g\ \frac{S}{cm^2} = g_{max}\ \frac{S}{cm^2} * q^2 \tag{46}$$

$$I_K\ \frac{mA}{cm^2} = g\ \frac{S}{cm^2} * (v\ mV - E_K\ mV) \tag{47}$$

The value of $q$ is also solved each time, by calculating the following equations:

$$\alpha_q\ ms^{-1} = 15\ mM^{-2}ms^{-1} * ([Ca^{2+}]_i\ mM)^2 \tag{48}$$

$$\beta_q\ ms^{-1} = 0.00025\ ms^{-1} \tag{49}$$

$$\tau_q\ ms = \frac{1}{q_{10} * (\alpha_q\ ms^{-1} + \beta_q\ ms^{-1})} \tag{50}$$

$$q_\infty = \alpha_q\ ms^{-1} * \tau_q\ ms \tag{51}$$

$$q_{exp} = 1 - exp\left(\frac{-dt\ ms}{\tau_q\ ms}\right) \tag{52}$$

$$q = q + q_{exp} * (q_\infty - q) \tag{53}$$

## Kdrfast

The following equations are solved each time step of simulation:

$$g\ \frac{S}{cm^2} = g_{max}\ \frac{S}{cm^2} * n^4 \tag{54}$$

$$I_K\ \frac{mA}{cm^2} = g\ \frac{S}{cm^2} * (v\ mV - E_K\ mV) \tag{55}$$

The value of $n$ is also solved each time step, by calculating the following equations:

$$\alpha_n \ ms^{-1} = -0.07 \ ms^{-1} * \frac{v \ mV + 18 \ mV}{exp\left(\frac{v \ mV + 18 \ mV}{-6 \ mV}\right) - 1} \tag{56}$$

$$\beta_n \ ms^{-1} = 0.264 \ ms^{-1} * exp\left(\frac{v \ mV + 43 \ mV}{40 \ mV}\right) \tag{57}$$

$$\tau_n \ ms = \frac{1}{\alpha_n \ ms^{-1} + \beta_n \ ms^{-1}} \tag{58}$$

$$n_\infty = \frac{\alpha_n \ ms^{-1}}{\alpha_n \ ms^{-1} + \beta_n \ ms^{-1}} \tag{59}$$

$$n_{exp} = 1 - exp\left(\frac{t_{inc} \ ms}{\tau_n \ ms}\right) \tag{60}$$

$$n = n + n_{exp} * (n_\infty - n) \tag{61}$$

## Kdrfastngf

The following equations are solved each time step of simulation:

$$g \ \frac{S}{cm^2} = g_{max} \ \frac{S}{cm^2} * n^4 \tag{62}$$

$$I_K \ \frac{mA}{cm^2} = g \ \frac{S}{cm^2} * (v \ mV - E_K \ mV) \tag{63}$$

The value of $n$ is also solved each time, by calculating the following equations:

$$\alpha_n \ ms^{-1} = -0.07 \ ms^{-1} * \frac{v \ mV + 8 \ mV}{exp\left(\frac{v \ mV + 8 \ mV}{-6 \ mV}\right) - 1} \tag{64}$$

$$\beta_n \ ms^{-1} = 0.264 \ ms^{-1} * exp\left(\frac{v \ mV + 33 \ mV}{40 \ mV}\right) \tag{65}$$

$$\tau_n \ ms = \frac{1}{\alpha_n \ ms^{-1} + \beta_n \ ms^{-1}} \tag{66}$$

$$n_\infty = \frac{\alpha_n \ ms^{-1}}{\alpha_n \ ms^{-1} + \beta_n \ ms^{-1}} \tag{67}$$

$$n_{exp} = 1 - exp\left(\frac{t_{inc} \ ms}{\tau_n \ ms}\right) \tag{68}$$

$$n = n + n_{exp} * (n_\infty - n) \tag{69}$$

## Kdrp

A different temperature dependence was implemented for this channel:

$$q_t = 1^{\frac{T - 24 \ ^oC}{10 \ ^oC}} \tag{70}$$

Then, the following equations are solved each time step:

$$g \; \frac{S}{cm^2} = g_{max} \; \frac{S}{cm^2} * n \tag{71}$$

$$I_K \; \frac{mA}{cm^2} = g \; \frac{S}{cm^2} * (v \; mV - E_K \; mV) \tag{72}$$

The values of $n$ and $l$ are also solved each time step, by calculating the following equations, where the ideal gas constant $R$ is $8.315 \; CVK^{-1}mol^{-1}$ and the conversion of Volts to milliVolts is given as $1.0e-3 \; VmV^{-1}$:

$$n_\infty = \frac{1}{1 + exp\left(\frac{1.0e-3 \; VmV^{-1} * (-3) * (v \; mV - 13 \; mV) * 9.648e4 \; Cmol^{-1}}{8.315 \; CVK^{-1}mol^{-1} * T \; K}\right)} \tag{73}$$

$$\tau_n \; ms = \frac{exp\left(\frac{1.0e-3 \; VmV^{-1} * (-3) * 0.7 * (v \; mV - 13 \; mV) * 9.648e4 \; Cmol^{-1}}{8.315 \; CVK^{-1}mol^{-1} * T \; K}\right)}{q_t * 0.02 \; ms^{-1} * \left(1 + exp\left(\frac{1.0e-3 \; VmV^{-1} * (-3) * (v \; mV - 13 \; mV) * 9.648e4 \; Cmol^{-1}}{8.315 \; CVK^{-1}mol^{-1} * T \; K}\right)\right)} \tag{74}$$

$$\frac{dn}{dt \; ms} = \frac{n_\infty - n}{\tau_n \; ms} \tag{75}$$

## Kdrslow

The following equations are solved each time step of simulation:

$$g \; \frac{S}{cm^2} = g_{max} \; \frac{S}{cm^2} * n^4 \tag{76}$$

$$I_K \; \frac{mA}{cm^2} = g \; \frac{S}{cm^2} * (v \; mV - E_K \; mV) \tag{77}$$

The value of $n$ is also solved each time, by calculating the following equations:

$$\alpha_n \; ms^{-1} = -0.028 \; ms^{-1}mV^{-1} * \frac{v \; mV + 30 \; mV}{exp\left(\frac{v \; mV + 30 \; mV}{-6 \; mV}\right) - 1} \tag{78}$$

$$\beta_n \; ms^{-1} = 0.1056 \; ms^{-1} * exp\left(\frac{v \; mV + 55 \; mV}{40 \; mV}\right) \tag{79}$$

$$\tau_n \; ms = \frac{1}{\alpha_n \; ms^{-1} + \beta_n \; ms^{-1}} \tag{80}$$

$$n_\infty = \frac{\alpha_n \; ms^{-1}}{\alpha_n \; ms^{-1} + \beta_n \; ms^{-1}} \tag{81}$$

$$n_{exp} = 1 - exp\left(\frac{t_{inc} \; ms}{\tau_n \; ms}\right) \tag{82}$$

$$n = n + n_{exp} * (n_\infty - n) \tag{83}$$

## KvA

A different temperature dependence was implemented for this channel:

$$q_{10} = 3^{\frac{T-30\,°C}{10\,°C}} \tag{84}$$

Then, the following equations are solved each time step of simulation, for every section within the cell, using the membrane potential $v$ in the section of the membrane where the ion channel is located.

$$g\ \frac{S}{cm^2} = g_{max}\ \frac{S}{cm^2} * n * l \tag{85}$$

$$I_A\ \frac{mA}{cm^2} = g\ \frac{S}{cm^2} * (v\ mV - E_K\ mV) \tag{86}$$

The values of $n$ and $l$ are also solved each time step, by calculating the following equations, where the ideal gas constant $R$ is 8.315 $CVK^{-1}mol^{-1}$ and the conversion of Volts to milliVolts is given as $1.0\mathrm{e}-3\ VmV^{-1}$:

$$n_\infty = \frac{1}{1 + exp\left(\frac{1.0\mathrm{e}-3\ VmV^{-1}*(-3)*(v\ mV+33.6\ mV)*9.648\mathrm{e}4\ Cmol^{-1}}{8.315\ CVK^{-1}mol^{-1}*T\ K}\right)} \tag{87}$$

$$\tau_n\ ms = \frac{exp\left(\frac{1.0\mathrm{e}-3\ VmV^{-1}*(-3)*0.6*(v\ mV+33.6\ mV)*9.648\mathrm{e}4\ Cmol^{-1}}{8.315\ CVK^{-1}mol^{-1}*T\ K}\right)}{q_{10} * 0.02\ ms^{-1} * \left(1 + exp\left(\frac{1.0\mathrm{e}-3\ VmV^{-1}*(-3)*(v\ mV+33.6\ mV)*9.648\mathrm{e}4\ Cmol^{-1}}{8.315\ CVK^{-1}mol^{-1}*T\ K}\right)\right)} \tag{88}$$

$$l_\infty = \frac{1}{1 + exp\left(\frac{1.0\mathrm{e}-3\ VmV^{-1}*4*(v\ mV+83\ mV)*9.648\mathrm{e}4\ Cmol^{-1}}{8.315\ CVK^{-1}mol^{-1}*T\ K}\right)} \tag{89}$$

$$\tau_l = \frac{exp\left(\frac{1.0\mathrm{e}-3\ VmV^{-1}*4*(v\ mV+83\ mV)*9.648\mathrm{e}4\ Cmol^{-1}}{8.315\ CVK^{-1}mol^{-1}*T\ K}\right)}{q_{10} * 0.08\ ms^{-1} * \left(1 + exp\left(\frac{1.0\mathrm{e}-3\ VmV^{-1}*4*(v\ mV+83\ mV)*9.648\mathrm{e}4\ Cmol^{-1}}{8.315\ CVK^{-1}mol^{-1}*T\ K}\right)\right)} \tag{90}$$

$$\frac{dn}{dt\ ms} = \frac{n_\infty - n}{\tau_n\ ms} \tag{91}$$

$$\frac{dl}{dt\ ms} = \frac{l_\infty - l}{\tau_l\ ms} \tag{92}$$

## KvAdistp

This channel implemented a different temperature dependence than usual:

$$q_t = 3^{\frac{T-24\,°C}{10\,°C}} \tag{93}$$

Then, the following equations are solved each time step of simulation, for every section within the cell, using the membrane potential $v$ in the section of the membrane where the ion channel is located.

$$g\ \frac{S}{cm^2} = g_{max}\ \frac{S}{cm^2} * n * l \tag{94}$$

$$I_A\ \frac{mA}{cm^2} = g\ \frac{S}{cm^2} * (v\ mV - E_K\ mV) \tag{95}$$

The values of $n$ and $l$ are also solved each time, by calculating the following equations, where the ideal gas constant $R$ is 8.315 $CVK^{-1}mol^{-1}$ and the conversion of Volts to milliVolts is given as $1.0\mathrm{e}-3\ VmV^{-1}$:

$$\frac{dn}{dt\,ms} = \frac{n_\infty - n}{\tau_n\,ms} \tag{96}$$

$$\frac{dl}{dt\,ms} = \frac{l_\infty - l}{\tau_l\,ms} \tag{97}$$

$$\alpha_n = exp\left(\frac{1.0e-3\ VmV^{-1}*\left(\frac{-1}{1+exp\left(\frac{v\,mV+40\,mV}{5\,mV}\right)}-1.8\right)*(v\,mV+1\,mV)*9.648e4\ Cmol^{-1}}{8.315\ CVK^{-1}mol^{-1}*T\ K}\right) \tag{98}$$

$$\beta_n = exp\left(\frac{1.0e-3\ VmV^{-1}*\left(\frac{-1}{1+exp\left(\frac{v\,mV+40\,mV}{5\,mV}\right)}-1.8\right)*0.39*(v\,mV+1\ Cmol^{-1})}{8.315\ CVK^{-1}mol^{-1}*T\ K}\right) \tag{99}$$

$$n_\infty = \frac{1}{1+\alpha_n} \tag{100}$$

$$\tau_n\,ms = \frac{\beta_n}{q_t*0.1\ ms^{-1}*(1+\alpha_n)} \tag{101}$$

$$\alpha_l = exp\left(\frac{1.0e-3\ VmV^{-1}*3*(v\,mV+56\,mV)*9.648e4\ Cmol^{-1}}{8.315\ CVK^{-1}mol^{-1}*T\ K}\right) \tag{102}$$

$$l_\infty = \frac{1}{1+\alpha_l} \tag{103}$$

$$\tau_l\,ms = 0.26\ ms*mV^{-1}*(v\,mV+50\,mV) \tag{104}$$

## KvAngf

This channel used a different form of temperature dependence:

$$q_{10} = 3^{\frac{T-30\,^oC}{10\,^oC}} \tag{105}$$

The following equations are solved each time step of simulation, for every section within the cell, using the membrane potential $v$ in the section of the membrane where the ion channel is located.

$$g\,\frac{S}{cm^2} = g_{max}\,\frac{S}{cm^2}*n*l \tag{106}$$

$$I_A\,\frac{mA}{cm^2} = g\,\frac{S}{cm^2}*(v\,mV - E_K\,mV) \tag{107}$$

The values of $n$ and $l$ are also solved each time, by calculating the following equations:

$$n_\infty = \frac{1}{1 + exp\left(\frac{1.0e-3\ VmV^{-1}*(-3)*(v\ mV+23.6\ mV)*9.648e4\ Cmol^{-1}}{8.315\ CVK^{-1}mol^{-1}*T\ K}\right)} \tag{108}$$

$$\tau_n\ ms = \frac{exp\left(\frac{1.0e-3\ VmV^{-1}*(-3)*0.6*(v\ mV+23.6\ mV)*9.648e4\ Cmol^{-1}}{8.315\ CVK^{-1}mol^{-1}*T\ K}\right)}{q_{10}*0.02\ ms^{-1}*\left(1 + exp\left(\frac{1.0e-3\ VmV^{-1}*(-3)*(v\ mV+23.6\ mV)*9.648e4\ Cmol^{-1}}{8.315\ CVK^{-1}mol^{-1}*T\ K}\right)\right)} \tag{109}$$

$$l_\infty = \frac{1}{1 + exp\left(\frac{1.0e-3\ VmV^{-1}*4*(v\ mV+83\ mV)*9.648e4\ Cmol^{-1}}{8.315\ CVK^{-1}mol^{-1}*T\ K}\right)} \tag{110}$$

$$\tau_l\ ms = \frac{exp\left(\frac{1.0e-3\ VmV^{-1}*4*(v\ mV+83\ mV)*9.648e4\ Cmol^{-1}}{8.315\ CVK^{-1}mol^{-1}*T\ K}\right)}{q_{10}*0.08\ ms^{-1}*\left(1 + exp\left(\frac{1.0e-3\ VmV^{-1}*4*(v\ mV+83\ mV)*9.648e4\ Cmol^{-1}}{8.315\ CVK^{-1}mol^{-1}*T\ K}\right)\right)} \tag{111}$$

$$\frac{dn}{dt\ ms} = \frac{n_\infty - n}{\tau_n\ ms} \tag{112}$$

$$\frac{dl}{dt\ ms} = \frac{l_\infty - l}{\tau_l\ ms} \tag{113}$$

## KvAolm

The following equations are solved each time step of simulation, for every section within the cell, using the membrane potential $v$ in the section of the membrane where the ion channel is located:

$$g\ \frac{S}{cm^2} = g_{max}\ \frac{S}{cm^2}*a*b \tag{114}$$

$$I_A\ \frac{mA}{cm^2} = g\ \frac{S}{cm^2}*(v\ mV - E_K\ mV) \tag{115}$$

The values of $a$ and $b$ are also solved each time, by calculating the following equations:

$$\tau_a\ ms = 5\ ms \tag{116}$$

$$a_\infty = \frac{1}{1 + exp\left(\frac{-(v\ mV+14\ mV)}{16.6\ mV}\right)} \tag{117}$$

$$\alpha_b\ ms^{-1} = \frac{0.000009\ ms^{-1}}{exp\left(\frac{v\ mV-26\ mV}{18.5\ mV}\right)} \tag{118}$$

$$\beta_b\ ms^{-1} = \frac{0.014\ ms^{-1}}{exp\left(\frac{v\ mV+70\ mV}{-11\ mV}\right) + 0.2} \tag{119}$$

$$\tau_b\ ms = \frac{1}{\alpha_b\ ms^{-1} + \beta_b\ ms^{-1}} \tag{120}$$

$$b_\infty = \frac{1}{1 + exp\left(\frac{v\ mv+71\ mv}{7.3\ mv}\right)} \tag{121}$$

$$a_{exp} = 1 - exp\left(\frac{-dt\ ms}{\tau_a\ ms}\right), \qquad b_{exp} = 1 - exp\left(\frac{-dt\ ms}{\tau_b\ ms}\right) \tag{122}$$

$$a = a + a_{exp} * (a_\infty - a), \qquad b = b + b_{exp} * (b_\infty - b) \tag{123}$$

## KvAproxp

This channel implemented a different temperature dependence:

$$q_{10} = 3^{\frac{T-24\,°C}{10\,°C}} \tag{124}$$

Then, the following equations are solved each time step:

$$g\,\frac{S}{cm^2} = g_{max}\,\frac{S}{cm^2} * n * l \tag{125}$$

$$I_A\,\frac{mA}{cm^2} = g\,\frac{S}{cm^2} * (v\,mV - E_K\,mV) \tag{126}$$

The values of $n$ and $l$ are also solved each time step, by calculating the following equations, where the ideal gas constant $R$ is $8.315\ CVK^{-1}mol^{-1}$ and the conversion of Volts to milliVolts is given as $1.0\mathrm{e}-3\ VmV^{-1}$:

$$\frac{dn}{dt\,ms} = \frac{n_\infty - n}{\tau_n\,ms} \tag{127}$$

$$\frac{dl}{dt\,ms} = \frac{l_\infty - l}{\tau_l\,ms} \tag{128}$$

$$\alpha_n = exp\left(\frac{1.0\mathrm{e}-3V\,mV^{-1} * \left(\frac{-1}{1+exp\left(\frac{v\,mV+40\,mV}{5\,mV}\right)} - 1.5\right) * (v\,mV - 11\,mV) * 9.648e4\ Cmol^{-1}}{8.315\ CVK^{-1}mol^{-1} * T\,K}\right) \tag{129}$$

$$\beta_n = exp\left(\frac{1.0\mathrm{e}-3\ VmV^{-1} * \left(\frac{-1}{1+exp\left(\frac{v\,mV+40\,mV}{5\,mV}\right)} - 1.5\right) * 0.55 * (v\,mV - 11\,mV) * 9.648e4\ Cmol^{-1}}{8.315\ CVK^{-1}mol^{-1} * T\,K}\right) \tag{130}$$

$$n_\infty = \frac{1}{1+\alpha_n} \tag{131}$$

$$\tau_n\,ms = \frac{\beta_n}{q_t * 0.05\,ms^{-1} * (1+\alpha_n)} \tag{132}$$

$$\alpha_l = exp\left(\frac{1.0\mathrm{e}-3\ VmV^{-1} * 3 * (v\,mV + 56\,mV) * 9.648e4\ Cmol^{-1}}{8.315\ CVK^{-1}mol^{-1} * T\,K}\right) \tag{133}$$

$$l_\infty = \frac{1}{1+\alpha_l} \tag{134}$$

$$\tau_l\,ms = 0.26\,ms * mV^{-1} * (v\,mV + 50\,mV) \tag{135}$$

## KvCaB

The following equations are solved each time step, where the ideal gas constant $R$ is $8.313424\ CVK^{-1}mol^{-1}$ and $F$ is Faraday's constant, $9.648e4\ Cmol^{-1}$. In the code for this channel, $F$ is reported in units of $kC$ rather than $C$, but it is always multiplied by the voltage in units of $mV$ instead of $V$, so the conversion is implicit ($mV * kC = V * C$).

$$g \ \frac{S}{cm^2} = g_{max} \ \frac{S}{cm^2} * n \tag{136}$$

$$I_K \ \frac{mA}{cm^2} = g \ \frac{S}{cm^2} * (v \ mV - E_K \ mV) \tag{137}$$

The value of $n$ is also solved each time, by calculating the following equations that depend on voltage and internal calcium concentration:

$$\alpha_n \ ms^{-1} = [Ca^{2+}]_i \ mM * \frac{0.28 \ ms^{-1}}{[Ca^{2+}]_i \ mM + \left(0.48e-3mM * exp\left(\frac{-2*0.84*F \ Cmol^{-1}*v \ mV}{R \ CVK^{-1}mol^{-1}*T \ K}\right)\right)} \tag{138}$$

$$\beta_n \ ms^{-1} = \frac{0.48 \ ms^{-1}}{1 + \frac{[Ca^{2+}]_i \ mM}{0.13e-6 \ mM*exp\left(\frac{-2*F \ Cmol^{-1}*v \ mV}{R \ CVK^{-1}mol^{-1}*T \ K}\right)}} \tag{139}$$

$$\tau_n \ ms = \frac{1}{\alpha_n \ ms^{-1} + \beta_n \ ms^{-1}} \tag{140}$$

$$n_\infty = \alpha_n \ ms^{-1} * \tau_n \ ms \tag{141}$$

$$n_{exp} = 1 - exp\left(\frac{t_{inc} \ ms}{\tau_n \ ms}\right) \tag{142}$$

$$n = n + n_{exp} * (n_\infty - n) \tag{143}$$

## KvGroup

The following equations are solved each time step of simulation:

$$g \ \frac{S}{cm^2} = g_{max} \ \frac{S}{cm^2} * n \tag{144}$$

$$I_{Group} \ \frac{mA}{cm^2} = g \ \frac{S}{cm^2} * (v \ mV - E_K \ mV) \tag{145}$$

The value of $n$ is also solved each time, by calculating the following equations:

$$\alpha_n \ ms^{-1} = 0.0189324 \ ms^{-1} * \frac{-(v \ mV - 4.18371 \ mV)}{exp\left(\frac{-(v \ mV - 4.18371 \ mV)}{6.42606 \ mV}\right) - 1} \tag{146}$$

$$\beta_n \ ms^{-1} = 0.015857 \ ms^{-1} \ * \ exp\left(\frac{-v \ mV}{25.4834 \ mV}\right) \tag{147}$$

$$\tau_n \ ms = \frac{1}{\alpha_n \ ms^{-1} + \beta_n \ ms^{-1}} \tag{148}$$

$$n_\infty = \frac{\alpha_n \ ms^{-1}}{\alpha_n \ ms^{-1} + \beta_n \ ms^{-1}} \tag{149}$$

$$n_{exp} = 1 - exp\left(\frac{t_{inc} \ ms}{\tau_n \ ms}\right) \tag{150}$$

$$n = n + n_{exp} * (n_\infty - n) \tag{151}$$

## KvM

This channel used a different temperature dependence than usual:

$$q_{10} = 5^{\frac{T-35\,^\circ C}{10\,^\circ C}} \tag{152}$$

Then, the following equations are solved each time step of simulation, for every section within the cell, using the membrane potential $v$ in the section of the membrane where the ion channel is located.

$$g\,\frac{S}{cm^2} = g_{max}\,\frac{S}{cm^2} * m \tag{153}$$

$$I_M\,\frac{mA}{cm^2} = g\,\frac{S}{cm^2} * (v\,mV - E_K\,mV) \tag{154}$$

The value of $m$ is also solved each time, by calculating the following equations:

$$m_\infty = \frac{1}{1 + exp\left(\frac{v\,mV + 40\,mV}{-10\,mV}\right)} \tag{155}$$

$$\tau_m\,ms = 120\,ms + \frac{exp(0.0378 * 7 * .4 * (v\,mV + 42\,mV))}{0.009\,ms^{-1} * (1 + exp(0.0378 * 7 * (v\,mV + 42\,mV)))} \tag{156}$$

$$\frac{dm}{dt\,ms} = \frac{m_\infty - m}{\tau_m\,ms} \tag{157}$$

## Leak

The following equations are solved each time step of simulation, for every section within the cell, using the membrane potential $v$ in the section of the membrane where the ion channel is located. Since the leak channel conducts mostly potassium, the reversal potential $E_{leak}$ is set close to the potassium reversal potential $E_K$.

$$g\,\frac{S}{cm^2} = g_{max}\,\frac{S}{cm^2} \tag{158}$$

$$I_{leak}\,\frac{mA}{cm^2} = g\,\frac{S}{cm^2} * (v\,mV - E_{leak}\,mV) \tag{159}$$

## Nav

The following equations are solved each time step:

$$g\,\frac{S}{cm^2} = g_{max}\,\frac{S}{cm^2} * m^3 * h \tag{160}$$

$$I_{Na}\,\frac{mA}{cm^2} = g\,\frac{S}{cm^2} * (v\,mV - E_{Na}\,mV) \tag{161}$$

The values of $m$ and $h$ are also solved each time, by calculating the following equations:

$$\alpha_m \; ms^{-1} = -0.3 \; ms^{-1} * \frac{v \; mV + 43 \; mV}{exp\left(\frac{v \; mV + 43 \; mV}{-5 \; mV}\right) - 1} \tag{162}$$

$$\beta_m \; ms^{-1} = 0.3 \; ms^{-1} * \frac{v \; mV + 15 \; mV}{exp\left(\frac{v \; mV + 15 \; mV}{5 \; mV}\right) - 1} \tag{163}$$

$$\tau_m \; ms = \frac{1}{\alpha_m \; ms^{-1} + \beta_m \; ms^{-1}} \tag{164}$$

$$m_\infty = \frac{\alpha_m \; ms^{-1}}{\alpha_m \; ms^{-1} + \beta_m \; ms^{-1}} \tag{165}$$

$$\alpha_h \; ms^{-1} = \frac{0.23 \; ms^{-1}}{exp\left(\frac{v \; mV + 65 \; mV}{20 \; mV}\right)} \tag{166}$$

$$\beta_h \; .ms^{-1} = \frac{3.33 \; ms^{-1}}{1 + exp\left(\frac{v \; mV + 12.5 \; mV}{-10 \; mV}\right)} \tag{167}$$

$$\tau_h \; ms = \frac{1}{\alpha_h \; ms^{-1} + \beta_h \; ms^{-1}} \tag{168}$$

$$h_\infty = \frac{\alpha_h \; ms^{-1}}{\alpha_h \; ms^{-1} + \beta_h \; ms^{-1}} \tag{169}$$

$$m_{exp} = 1 - exp\left(\frac{t_{inc} \; ms}{\tau_m \; ms}\right), \qquad h_{exp} = 1 - exp\left(\frac{t_{inc} \; ms}{\tau_h \; ms}\right) \tag{170}$$

$$m = m + m_{exp} * (m_\infty - m), \qquad h = h + h_{exp} * (h_\infty - h) \tag{171}$$

## Navaxonp

Within the ion channel, these equations solve to constants:

$$q_t = 1^{\frac{T - 24 \; °C}{10 \; °C}} \tag{172}$$

Then, the following equations are solved each time step:

$$g \; \frac{S}{cm^2} = g_{max} \; \frac{S}{cm^2} * m^3 * h \tag{173}$$

$$I_{Na} \; \frac{mA}{cm^2} = g \; \frac{S}{cm^2} * (v \; mV - E_{Na} \; mV) \tag{174}$$

The values of $m$ and $h$ are also solved each time step, by calculating the following equations:

$$InAct = 1 \tag{175}$$

$$\frac{dm}{dt \; ms} = \frac{m_\infty - m}{\tau_m \; ms} \tag{176}$$

$$\frac{dh}{dt \; ms} = \frac{h_\infty - h}{\tau_h \; ms} \tag{177}$$

$$\alpha_m \ ms^{-1} = \frac{Ra \ ms^{-1} * (v \ mV + 15 \ mV)}{1 - exp\left(\frac{-(v \ mV + 15 \ mV)}{7.2 \ mV}\right)} \tag{178}$$

$$beta_m = \frac{Rb \ ms^{-1} * (-v - 15 \ mV)}{1 - exp\left(\frac{-(-v - 15 \ mV)}{7.2 \ mV}\right)} \tag{179}$$

$$\tau_m \ ms = \frac{1}{(\alpha_m \ ms^{-1} + \beta_m \ ms^{-1}) * q_t} \tag{180}$$

$$m_\infty = \frac{\alpha_m \ ms^{-1}}{\alpha_m \ ms^{-1} + \beta_m \ ms^{-1}} \tag{181}$$

$$\alpha_h = \frac{Rd \ ms^{-1} * (v \ mV + 30 \ mV)}{1 - exp\left(\frac{-(v \ mV + 30 \ mV)}{1.5 \ mV}\right)} \tag{182}$$

$$\beta_h = \frac{Rg \ ms^{-1} * (-v - 30 \ mV)}{1 - exp\left(\frac{-(-v - 30 \ mV)}{1.5 \ mV}\right)} \tag{183}$$

$$\tau_h \ ms = \frac{1}{(\alpha_h \ ms^{-1} + \beta_h \ ms^{-1}) * q_t} \tag{184}$$

$$h_\infty = \frac{1}{1 + exp\left(\frac{v \ mV + 35 \ mV}{4 \ mV}\right)} \tag{185}$$

## Navbis

The following equations are solved each time step:

$$g \ \frac{S}{cm^2} = g_{max} \ \frac{S}{cm^2} * m^3 * h \tag{187}$$

$$I_{Na} \ \frac{mA}{cm^2} = g \ \frac{S}{cm^2} * (v \ mV - E_{Na} \ mV) \tag{188}$$

The values of $m$ and $h$ are also solved each time, by calculating the following equations:

$$\alpha_m \ ms^{-1} = -0.2 \ ms^{-1} * \frac{v \ mV + 38 \ mV}{exp\left(\frac{v \ mV + 38 \ mV}{-5 \ mV}\right) - 1} \tag{189}$$

$$\beta_m \ ms^{-1} = 0.5 \ ms^{-1} * \frac{v \ mV + 10 \ mV}{exp\left(\frac{v \ mV + 10 \ mV}{5 \ mV}\right) - 1} \tag{190}$$

$$\tau_m \ ms = \frac{1}{\alpha_m \ ms^{-1} + \beta_m \ ms^{-1}} \tag{191}$$

$$m_\infty = \frac{\alpha_m \ ms^{-1}}{\alpha_m \ ms^{-1} + \beta_m \ ms^{-1}} \tag{192}$$

$$\alpha_h \, ms^{-1} = \frac{0.23 \, ms^{-1}}{exp\left(\frac{v \, mV + 62 \, mV}{20 \, mV}\right)} \tag{193}$$

$$\beta_h \, ms^{-1} = \frac{2 \, ms^{-1}}{1 + exp\left(\frac{v \, mV + 9.5 \, mV}{-10 \, mV}\right)} \tag{194}$$

$$\tau_h \, ms = \frac{1}{\alpha_h \, ms^{-1} + \beta_h \, ms^{-1}} \tag{195}$$

$$h_\infty = \frac{\alpha_h \, ms^{-1}}{\alpha_h \, ms^{-1} + \beta_h \, ms^{-1}} \tag{196}$$

$$m_{exp} = 1 - exp\left(\frac{t_{inc} \, ms}{\tau_m \, ms}\right), \qquad h_{exp} = 1 - exp\left(\frac{t_{inc} \, ms}{\tau_h \, ms}\right) \tag{197}$$

$$m = m + m_{exp} * (m_\infty - m), \qquad h = h + h_{exp} * (h_\infty - h) \tag{198}$$

## Navcck

The following equations are solved each time step:

$$g \, \frac{S}{cm^2} = g_{max} \, \frac{S}{cm^2} * m^3 * h * s \tag{199}$$

$$I_{Na} \, \frac{mA}{cm^2} = g \, \frac{S}{cm^2} * (v \, mV - E_{Na} \, mV) \tag{200}$$

The values of $m$, $h$ (fast inactivation) and $s$ (slow inactivation) are also solved each time, by calculating the following equations:

$$\alpha_m \, ms^{-1} = -0.5 \, ms^{-1} * \frac{v \, mV + 42 \, mV}{exp\left(\frac{v \, mV + 42 \, mV}{-5 \, mV}\right) - 1} \tag{201}$$

$$\beta_m \, ms^{-1} = 0.3 \, ms^{-1} * \frac{v \, mV + 13 \, mV}{exp\left(\frac{v \, mV + 13 \, mV}{5 \, mV}\right) - 1} \tag{202}$$

$$\tau_m \, ms = \frac{1}{\alpha_m \, ms^{-1} + \beta_m \, ms^{-1}} \tag{203}$$

$$m_\infty = \frac{\alpha_m \, ms^{-1}}{\alpha_m \, ms^{-1} + \beta_m \, ms^{-1}} \tag{204}$$

$$\alpha_h \, ms^{-1} = \frac{0.6 \, ms^{-1}}{exp\left(\frac{v \, mV + 65 \, mV}{20 \, mV}\right)} \tag{205}$$

$$\beta_h \, ms^{-1} = \frac{1.31 \, ms^{-1}}{1 + exp\left(\frac{v \, mV + 12.5 \, mV}{-10 \, mV}\right)} \tag{206}$$

$$\tau_h \, ms = \frac{1}{\alpha_h \, ms^{-1} + \beta_h \, ms^{-1}} \tag{207}$$

$$h_\infty = \frac{\alpha_h \, ms^{-1}}{\alpha_h \, ms^{-1} + \beta_h \, ms^{-1}} \tag{208}$$

$$\alpha_s \ ms^{-1} = \frac{0.003 \ ms^{-1}}{exp\left(\frac{v + 45 \ mV}{6 \ mV}\right)} \tag{209}$$

$$\beta_s \ ms^{-1} = \frac{0.005 \ ms^{-1}}{1 + exp\left(\frac{v \ mV + 35 \ mV}{-20 \ mV}\right)} \tag{210}$$

$$\tau_s \ ms = \frac{1}{\alpha_s \ ms^{-1} + \beta_s \ ms^{-1}} \tag{211}$$

$$s_\infty = \frac{\alpha_s \ ms^{-1}}{\alpha_s \ ms^{-1} + \beta_s \ ms^{-1}} \tag{212}$$

$$m_{exp} = 1 - exp\left(\frac{t_{inc} \ ms}{\tau_m \ ms}\right), \qquad h_{exp} = 1 - exp\left(\frac{t_{inc} \ ms}{\tau_h \ ms}\right), \qquad s_{exp} = 1 - exp\left(\frac{t_{inc} \ ms}{\tau_s \ ms}\right) \tag{213}$$

$$m = m + m_{exp} * (m_\infty - m), \qquad h = h + h_{exp} * (h_\infty - h), \qquad s = s + s_{exp} * (s_\infty - s) \tag{214}$$

## Navngf

The following equations are solved each time step:

$$g \ \frac{S}{cm2} = g_{max} \ \frac{S}{cm2} * m^3 * h \tag{215}$$

$$I_{Na} \ \frac{mA}{cm2} = g \ \frac{S}{cm2} * (v \ mV - E_{Na} \ mV) \tag{216}$$

The values of $m$ and $h$ are also solved each time, by calculating the following equations:

$$\alpha_m \ ms^{-1} = -0.34133 \ ms^{-1} * \frac{v \ mV + 24 \ mV}{exp\left(\frac{v \ mV + 24 \ mV}{-5 \ mV}\right) - 1} \tag{217}$$

$$\beta_m \ ms^{-1} = 0.28483 \ ms^{-1} * \frac{v \ mV - 4 \ mV}{exp\left(\frac{v \ mV - 4 \ mV}{5 \ mV}\right) - 1} \tag{218}$$

$$\tau_m \ ms = \frac{1}{\alpha_m \ ms^{-1} + \beta_m \ ms^{-1}} \tag{219}$$

$$m_\infty = \frac{\alpha_m \ ms^{-1}}{\alpha_m \ ms^{-1} + \beta_m \ ms^{-1}} \tag{220}$$

$$\alpha_h \ ms^{-1} = \frac{0.29648 \ ms^{-1}}{exp\left(\frac{v \ mV + 64.4184 \ mV}{20 \ mV}\right)} \tag{221}$$

$$\beta_h \ ms^{-1} = \frac{3.0931 \ ms^{-1}}{1 + exp\left(\frac{v \ mV + 12.1463 \ mV}{-10 \ mV}\right)} \tag{222}$$

$$\tau_h \ ms = \frac{1}{\alpha_h \ ms^{-1} + \beta_h \ ms^{-1}} \tag{223}$$

$$h_\infty = \frac{\alpha_h \ ms^{-1}}{\alpha_h \ ms^{-1} + \beta_h \ ms^{-1}} \tag{224}$$

$$m_{exp} = 1 - exp\left(\frac{t_{inc} \ ms}{\tau_m \ ms}\right), \qquad h_{exp} = 1 - exp\left(\frac{t_{inc} \ ms}{\tau_h \ ms}\right) \tag{225}$$

$$m = m + m_{exp} * (m_\infty - m), \qquad h = h + h_{exp} * (h_\infty - h) \tag{226}$$

## Navp

This channel model has a different dependence on temperature:

$$q_t = 1^{\frac{T - 24\,°C}{10\,°C}} \tag{227}$$

Then, the following equations are solved each time step:

$$g\ \frac{S}{cm^2} = g_{max}\ \frac{S}{cm^2} * m^3 * h * s \tag{228}$$

$$I_{Na}\ \frac{mA}{cm^2} = g\ \frac{S}{cm^2} * (v\ mV - E_{Na}\ mV) \tag{229}$$

The values of $m,\ h,$ and $s$ are also solved each time step, by calculating the following equations, where the ideal gas constant $R$ is $8.315\ CVK^{-1}mol^{-1}$ and the conversion of Volts to milliVolts is given as $1.0e - 3\ VmV^{-1}$:

$$InAct = 1 \tag{230}$$

$$\frac{dm}{dt\ ms} = \frac{m_\infty - m}{\tau_m\ ms} \tag{231}$$

$$\frac{dh}{dt\ ms} = \frac{h_\infty - h}{\tau_h\ ms} \tag{232}$$

$$\frac{ds}{dt\ ms} = \frac{s_\infty - s}{\tau_s\ ms} \tag{233}$$

$$\alpha_m\ ms^{-1} = \frac{Ra\ ms^{-1} * (v\ mV - (-15)\ mV)}{1 - exp\left(\frac{-(v\ mV - (-15)\ mV)}{7.2\ mV}\right)} \tag{234}$$

$$\beta_m\ ms^{-1} = \frac{Rb\ ms^{-1} * (-v - (15)\ mV)}{1 - exp\left(\frac{-(-v - (15)\ mV)}{7.2\ mV}\right)} \tag{235}$$

$$\tau_m\ ms = \frac{1}{(\alpha_m\ ms^{-1} + \beta_m\ ms^{-1}) * q_t} \tag{236}$$

$$m_\infty = \frac{\alpha_m}{\alpha_m + \beta_m} \tag{237}$$

$$\alpha_h\ ms^{-1} = \frac{Rd\ ms^{-1} * (v\ mV - (-30)\ mV)}{1 - exp\left(\frac{-(v\ mV - (-30)\ mV)}{1.5\ mV}\right)} \tag{238}$$

$$\beta_h\ ms^{-1} = \frac{Rg\ ms^{-1} * (-v - (30)\ mV)}{1 - exp\left(\frac{-(-v - (30)\ mV)}{1.5\ mV}\right)} \tag{239}$$

$$\tau_h = \frac{1}{(\alpha_h\ ms^{-1} + \beta_h\ ms^{-1}) * q_t} \tag{240}$$

$$h_\infty = \frac{1}{1 + exp\left(\frac{v\ mV + 35\ mV}{4\ mV}\right)} \tag{241}$$

$$s_\infty = \frac{1}{\left(1 + exp\left(\frac{v\,mV + 43\,mV}{2\,mV}\right)\right)} + InAct * \left(1 - \frac{1}{\left(1 + exp\left(\frac{v\,mV + 43\,mV}{2\,mV}\right)\right)}\right) \tag{242}$$

$$\tau_s\,ms = \frac{exp\left(\frac{1.0e-3\,VmV^{-1}*12*0.2*(v\,mV+45\,mV)*9.648e4\,Cmol^{-1}}{8.315\,CVK^{-1}mol^{-1}*T\,K}\right)}{0.0003\,ms^{-1} * \left(1 + exp\left(\frac{1.0e-3\,VmV^{-1}*12*(v\,mV+45\,mV)*9.648e4\,Cmol^{-1}}{8.315\,CVK^{-1}mol^{-1}*T\,K}\right)\right)} \tag{243}$$

