## [Decision Letter]

Thank you for submitting your article "Interneuronal mechanisms of hippocampal theta oscillation in full-scale and rationally reduced models of the CA1 circuit" for consideration by *eLife*. Your article has been favorably evaluated by Eve Marder (Senior Editor) and three reviewers, one of whom, Frances K Skinner (Reviewer #1), is a member of our Board of Reviewing Editors.

The reviewers have discussed the reviews with one another and the Reviewing Editor has drafted this decision to help you prepare a revised submission.

Summary:

The authors have developed a large-scale model of CA1 microcircuitry, with conductance-based models of nine distinct cell types. It is a treasure trove of highly intertwined model and experimental details, an impressive tour de force of bringing together much experimental detail in a cohesive, carefully constructed model that is open and accessible with available. The authors propose this as a first full-scale CA1 model capable of generating theta. Perturbation experiments show that some, but not all, of the neuron types are necessary to generate the theta rhythm, and conclude that the diversity of interneuron types is necessary to generate theta.

All reviewers recognized that putting this together was a herculean task and the commitment to open access was appreciated. However, there were also various concerns raised by all the reviewers. We truly understand the difficulty of writing and preparing these papers, with all of the concomitant decisions about what to include, so we hope that the following comments will give you a sense of how well the decisions you have made are "working" for the reader.

Essential revisions:

1) Coordination with companion paper and network clamp contribution to present paper

All the reviewers have had a chance to view the companion paper and it was felt that inclusion of the 'network clamp' in the present form in this manuscript was not a good fit as the present paper is a research article type. Rather, it was felt that the authors should focus on the scientific results of the work in the present paper. The authors should consider what important things have been demonstrated about the network by using the NetworkClamp tool. And even though it requires more full-scale simulations, they could make those changes to the full network model itself for the present paper.

i) For example, Panel 7E seems to be the essential one: show how theta is affected by reducing separately PV+ or SOM+ input by 90% – do these simulations for the full model here, and state them here. Then in the companion paper, they could show the current Figure 7 and Figure 8 whereby the NetworkClamp tool achieves analogous results, but much more quickly. Also, then here they could make a reference to the SimTracker paper as a place where readers could learn more.

ii) As it stands, the description and proposed use of 'Network Clamp' is somewhat confusing. Comparison is made with being similar to in vivo (from perspective of a single pyramidal cell) – but the spontaneous theta is presented in reference to a whole hippocampus preparation (Goutagny et al. 2009, Amilhon et al.). Also, it is the case that one needs the full-scale simulation in the first place (to have the appropriate inputs), before using networkclamp, right? – Thus, the advantage in the present paper is not quite clear. Also, the CA3 and EC afferent inputs are still there – so that inhibitory sculpting and cross-validation comments unclear. Could they expand and explain use intentions more specifically?

iii) Subsection “Interneuronal contributions to theta oscillations in simplified models derived from the full-scale virtual CA1 network”, first paragraph: implies that Methods would provide greater detail on the implementation of the Network Clamp, but there is not any mention of it there. Was a section in Methods omitted by mistake? Is the "network clamp" just a theoretical concept that the authors implemented entirely by hand, or is it a mechanism that NEURON users could just add to their own network simulations? Unclear. Also, it sounds like presynaptic spike trains that are inputs to the pyramidal cell are entrained to theta, but that could be better articulated in a Methods subsection.

2) Clarity about overall goal/theta generation mechanism/experimental comparisons

i) Subsection “An accessible approach to modeling that balances detail, scale, flexibility and performance”, second paragraph: In commenting on the overall approach/strategy, the authors mention answering particular questions. The question presumably is about how theta is generated in CA1 microcircuitry (such as a whole hippocampus preparation of Goutagny et al.). If this is the case, the authors should state this explicitly.

Along these same lines, are the authors concerned about rat and mouse differences? (Goutagny et al. is rat, but Amilhon et al. is mouse, and model is rat, and both experimental papers are referred to and compared).

Model details are from rat CA1 (Bezaire and Soltesz), and in consideration with whole hippocampus prep of Goutagny et al. However, model comparisons are done with in vivo (e.g., subsection “Emergence of spontaneous theta and γ oscillations in the full-scale model in the absence of rhythmic external inputs”, fourth paragraph) with inhibition aspects, and the higher firing pyramidal cells to in vivo (rather than in whole hippocampus prep). Please be clear about what is used and being compared to and why, i.e., rationales given.

ii) If possible, the authors should provide a clear summary of the theta generation mechanism (assuming this is the question being asked). 'Factors' of diversity, etc. are given, but not really a generation mechanism it seems although it clearly seems to revolve around the 0.65Hz-based inputs. In essence, the authors should edit their paper to be clear about intentions and interpretation.

iii) The authors show that some interneuron diversity is necessary to generate theta. In particular, perturbing all interneurons to behave/look progressively more like one type of interneuron (PV+B) did not allow for generation of theta. Presumably in these simulations, all interneurons had identical passive/conductance/kinetic parameters? How much variance is there in firing patterns among PV+B cells characterized empirically? Might this variance in physiological firing properties be sufficient to generate the theta rhythm you observe within your CA1 model? This could be tested with one more round of simulations, similar to the network configuration in Figure 6 except the one constant set of intrinsic PV+B-like model parameters applied to all interneurons could be varied randomly within some set range.

iv) Subsection “Phase-preferential firing of interneurons in the full-scale model of the isolated CA1”, last paragraph and Figure 5: Model vs. experiment differences occur in PV+B – which you then perturb to show how it related to the generation of theta. How do you explain that your model PV+B cells fire at the trough of theta, while experiments show them nearly in phase with the peaks? Is that what is meant by the "note" about the recording site in Ferguson et al. in these lines? (Similar question for the Ivy cells – though that is less important here Ivy's were not essential to the model's theta generation.) Otherwise – isn't this problematic since your exploration of PV+B was so essential to the Figure 6 content?

Subsection “Stimulation”, last paragraph: Figure 5—figure supplement 1 indicates large discrepancies between model and empirical data. Authors comment very little on this, but shouldn't this be a big deal?

v) Even though the network models were constrained by biological data, the models still failed to reproduce the experimentally observed theta modulated neuronal firing phases of principal and inhibitory neurons (e.g. see Figure 5). For instance, according to the Klausberger data the basket cell profile is that of a neural accumulator, that is it linearly rises till a max value and abruptly ends. However, the model BC response is gaussian/Bell shape! Can the authors comment as why this is case even though their circuit models are faithful representations of the real CA1 circuit in terms of cell numbers, connectivities, etc.? Are they still missing important components which were not included in their models? Can they comment (even speculate) which are these components?

vi) Subsection “Perturbation experiments indicate a key role for interneuronal diversity in the emergence of spontaneous theta”, second paragraph: – Data not shown for what happens when pyramidal cell connectivity removed – this is quite interesting as it is not many connections as authors point out, and may help the authors and others understand the mechanistic essence. Could more specifics be provided (or relevant Figure 6 and supplement be expanded?) when it is said theta rhythm collapses – does that mean no rhythm or higher frequency or unclear output or what?

vii) Subsection “Perturbation experiments indicate a key role for interneuronal diversity in the emergence of spontaneous theta”, last paragraph: It is said that goes from silent to hyperactive, but this does not seem to be the case in Figure 6 or supplement where the non-theta all seems to be around 3 Hz?

viii) Subsection “Emergence of theta oscillations from a biological data-driven, full-scale model of the CA1 network”, third paragraph: 33,500 spikes obtained in model – how was that determined, and presumably, with the 0.65 Hz input, how different is the number of spikes with larger or smaller frequency inputs (where they have shown the theta rhythm collapses)? This could be helpful in getting at the theta rhythm essence.

3) Model details

i) The description of intrinsic properties of individual neurons (passive and active parameters of the conductance-based models) is entirely missing from the paper, and inadequate on the model website. These model details need to be accessible forever, directly from the paper (or as supplementary information), not from an external website that could be discontinued. Yet even on the website, the information about individual neurons is incomplete/inconvenient to access. For example, a row of graphs about each channel in the model is included on the website (http://mariannebezaire.com/ca1_graphic/mymodel.html), but the axes of the graphs are not labeled. Do these show voltage activation curves? Distributions throughout the somato-dendritic axis? Are all ion channels included in each neuron type? Also, I found out by accident (website had no instructions) that manually clicking on each morphology sends you to a page with a set of tables and images with drop-down menus. It would have been helpful to provide text/tables summarizing channels included in each neuron type, with graphs of model vs. experimental traces and references to the exact equations for channel. It is essential for such information to be accessible from supplemental information, not an external site, unless it is a well-monitored, maintained, and curated site such as model DB.

ii) Such detailed models suffer from too much detail, which translates mathematically into thousands of parameters that need to be tweaked to simulate accurately and quantitatively the responses of cells in the networks. Did the authors use any optimization technique to constrain their model's responses other than calculated guises of biological detail?

The network models contained 1 excitatory principal cell type and 8 types inhibitory interneurons. These interneurons targeted specific compartments of the principal cells. But what about other types of excitatory cells? Also, what about inhibitory cells that targeted CA1 inhibitory cells (e.g. Chamberland, S., and Topolnik, L. (2012))?

---

## [Author Response]

*Essential revisions:*

*1) Coordination with companion paper and network clamp contribution to present paper*

*All the reviewers have had a chance to view the companion paper and it was felt that inclusion of the 'network clamp' in the present form in this manuscript was not a good fit as the present paper is a research article type. Rather, it was felt that the authors should focus on the scientific results of the work in the present paper. The authors should consider what important things have been demonstrated about the network by using the NetworkClamp tool. And even though it requires more full-scale simulations, they could make those changes to the full network model itself for the present paper.*

The Network Clamp results have been removed from the current paper.

*i) For example, Panel 7E seems to be the essential one: show how theta is affected by reducing separately PV+ or SOM+ input by 90% – do these simulations for the full model here, and state them here. Then in the companion paper, they could show the current Figure 7 and Figure 8 whereby the NetworkClamp tool achieves analogous results, but much more quickly. Also, then here they could make a reference to the SimTracker paper as a place where readers could learn more.*

Done.As suggested,we ran simulations of partial PV or SOM disinhibition, and these results are now included as new Figure panel 6D. In addition, we removed all Network Clamp results, keeping only a brief one-sentence reference to it as a pointer to our other manuscript. We also briefly mention it in the Methods section, as it played a role in tuning and developing our full-scale model.

*ii) As it stands, the description and proposed use of 'Network Clamp' is somewhat confusing. Comparison is made with being similar to in vivo (from perspective of a single pyramidal cell) – but the spontaneous theta is presented in reference to a whole hippocampus preparation (Goutagny et al. 2009, Amilhon et al.). Also, it is the case that one needs the full-scale simulation in the first place (to have the appropriate inputs), before using networkclamp, right? Thus, the advantage in the present paper is not quite clear. Also, the CA3 and EC afferent inputs are still there – so that inhibitory sculpting and cross-validation comments unclear. Could they expand and explain use intentions more specifically?*

Done, section removed. The criticisms within this point will be addressed in our Simtracker manuscript that will be submitted elsewhere (and already available on BioRXiv as Bezaire et al., 2016, see http://biorxiv.org/content/early/2016/10/19/081927).

*iii) Subsection “Interneuronal contributions to theta oscillations in simplified models derived from the full-scale virtual CA1 network”, first paragraph: implies that Methods would provide greater detail on the implementation of the Network Clamp, but there is not any mention of it there. Was a section in Methods omitted by mistake? Is the "network clamp" just a theoretical concept that the authors implemented entirely by hand, or is it a mechanism that NEURON users could just add to their own network simulations? Unclear. Also, it sounds like presynaptic spike trains that are inputs to the pyramidal cell are entrained to theta, but that could be better articulated in a Methods subsection.*

Done, section removed. The criticisms within this point will be addressed in our Simtracker manuscript.

*2) Clarity about overall goal/theta generation mechanism/experimental comparisons*

*i) Subsection “An accessible approach to modeling that balances detail, scale, flexibility and performance”, second paragraph: In commenting on the overall approach/strategy, the authors mention answering particular questions. The question presumably is about how theta is generated in CA1 microcircuitry (such as a whole hippocampus preparation of Goutagny et al.). If this is the case, the authors should state this explicitly.*

Done. We have clarified (in Introduction, first paragraph; In Discussion, seventh paragraph) that our particular question in this manuscript is how the isolated CA1 circuitry (Goutagny et al., 2009) can generate theta oscillations.

*Along these same lines, are the authors concerned about rat and mouse differences? (Goutagny et al. is rat, but Amilhon et al. is mouse, and model is rat, and both experimental papers are referred to and compared).*

Done. We have addressed this issue in the Discussion in a new section under the subheading “Rationale for bases of comparison between modeling results with experimental data”.

*Model details are from rat CA1 (Bezaire and Soltesz), and in consideration with whole hippocampus prep of Goutagny et al. However, model comparisons are done with in vivo (e.g., subsection “Emergence of spontaneous theta and γ oscillations in the full-scale model in the absence of rhythmic external inputs”, fourth paragraph) with inhibition aspects, and the higher firing pyramidal cells to in vivo (rather than in whole hippocampus prep). Please be clear about what is used and being compared to and why, i.e., rationales given.*

Done. We have addressed this issue in the Discussion in a new section under the subheading “Rationale for bases of comparison between modeling results with experimental data”.

*ii) If possible, the authors should provide a clear summary of the theta generation mechanism (assuming this is the question being asked). 'Factors' of diversity, etc. are given, but not really a generation mechanism it seems although it clearly seems to revolve around the 0.65Hz-based inputs. In essence, the authors should edit their paper to be clear about intentions and interpretation.*

Done.We have added an integrated summary of the mechanisms underlying the emergence of theta in our model (Discussion, subsection “Emergence of theta oscillations from a biological data-driven, full-scale model of the CA1 network”).

*iii) The authors show that some interneuron diversity is necessary to generate theta. In particular, perturbing all interneurons to behave/look progressively more like one type of interneuron (PV+B) did not allow for generation of theta. Presumably in these simulations, all interneurons had identical passive/conductance/kinetic parameters? How much variance is there in firing patterns among PV+B cells characterized empirically? Might this variance in physiological firing properties be sufficient to generate the theta rhythm you observe within your CA1 model? This could be tested with one more round of simulations, similar to the network configuration in Figure 6 except the one constant set of intrinsic PV+B-like model parameters applied to all interneurons could be varied randomly within some set range.*

Done. We have run additional simulations with added variability. The results are described in the fourth paragraph of the subsection “Perturbation experiments indicate a key role for interneuronal diversity in the emergence of spontaneous theta”. Briefly, we ran simulations using the network configuration of “All PV+B” but allowed the resting membrane potential of the (PV+B) interneurons to vary with a standard deviation of 2 mV, 5 mV, or 8 mV (based on experimental variability reported in Tricoire et al., 2011 and Mercer et al. 2012). None of these configurations were able to rescue the spontaneous theta rhythm, nor could the theta rhythm in the control network be disrupted by introducing variability in RMP in the control network.

*iv) Subsection “Phase-preferential firing of interneurons in the full-scale model of the isolated CA1”, last paragraph and Figure 5: Model vs. experiment differences occur in PV+B – which you then perturb to show how it related to the generation of theta. How do you explain that your model PV+B cells fire at the trough of theta, while experiments show them nearly in phase with the peaks? Is that what is meant by the "note" about the recording site in Ferguson et al. in these lines? (Similar question for the Ivy cells – though that is less important here Ivy's were not essential to the model's theta generation.) Otherwise – isn't this problematic since your exploration of PV+B was so essential to the Figure 6 content?*

*Subsection “Stimulation”, last paragraph: Figure 5—figure supplement 1 indicates large discrepancies between model and empirical data. Authors comment very little on this, but shouldn't this be a big deal?*

Done. Yes, there is a 180˚ shift in LFP phase when recording from stratum radiatum (as in Ferguson et al) versus stratum pyramidale (current study, and most of the experimental data we cite).

Regarding Figure 5—figure supplement 1, we agree, there are some discrepancies for certain cell types, especially for CCK^+^ cells. We addressed this issue in the Methods (under subheading “Stimulation”, last paragraph). Briefly, it is not generally known yet whether firing rates in the isolated preparation, for given cell types, are closer to anesthetized or awake in vivo rates. In general, it is likely that the in vivo network receives additional inhibition from extra-CA1 sources, and some of these are specifically target interneurons (e.g., the lateral entorhinal GABAergic input targets CCK^+^ cells).

*v) Even though the network models were constrained by biological data, the models still failed to reproduce the experimentally observed theta modulated neuronal firing phases of principal and inhibitory neurons (e.g. see Figure 5). For instance, according to the Klausberger data the basket cell profile is that of a neural accumulator, that is it linearly rises till a max value and abruptly ends. However, the model BC response is gaussian/Bell shape! Can the authors comment as why this is case even though their circuit models are faithful representations of the real CA1 circuit in terms of cell numbers, connectivities, etc.? Are they still missing important components which were not included in their models? Can they comment (even speculate) which are these components?*

Yes, we believe that major extra-CA1 phasically active projections, such as the CA3 inputs to PV+ basket cells (and others, such as the septal GABAergic input), likely contribute to the phase preference of PV+ basket cells during theta. This possibility is discussed in the last paragraph of the subsection “Mechanism of theta generation and phase-preferential firing of interneurons in the full-scale model of the isolated CA1”.

*vi) Subsection “Perturbation experiments indicate a key role for interneuronal diversity in the emergence of spontaneous theta”, second paragraph: Data not shown for what happens when pyramidal cell connectivity removed – this is quite interesting as it is not many connections as authors point out, and may help the authors and others understand the mechanistic essence. Could more specifics be provided (or relevant Figure 6 and supplement be expanded?) when it is said theta rhythm collapses – does that mean no rhythm or higher frequency or unclear output or what?*

Done. We have carried out additional simulations to show the result of doubling or halving the number of connections between CA1 pyramidal cells, and the results are shown in Figure 6 along with the results of removing CA1 pyramidal- pyramidal connectivity. We have included a discussion of these results in our section on the mechanisms of theta generation in our model (Discussion, seventh paragraph; see also Results subsection “Perturbation experiments indicate a key role for interneuronal diversity in the emergence of spontaneous theta”, third paragraph). The revised supplemental figure Figure 6—figure supplement 1 illustrates how the oscillation frequencies change when the theta rhythm collapses in response to various manipulations.

*vii) Subsection “Perturbation experiments indicate a key role for interneuronal diversity in the emergence of spontaneous theta”, last paragraph: It is said that goes from silent to hyperactive, but this does not seem to be the case in Figure 6 or supplement where the non-theta all seems to be around 3 Hz?*

Done.We revised the wording in the description of these results (subsection “Perturbation experiments indicate a key role for interneuronal diversity in the emergence of spontaneous theta”).

*viii) Subsection “Emergence of theta oscillations from a biological data-driven, full-scale model of the CA1 network”, third paragraph: 33,500 spikes obtained in model – how was that determined, and presumably, with the 0.65 Hz input, how different is the number of spikes with larger or smaller frequency inputs (where they have shown the theta rhythm collapses)? This could be helpful in getting at the theta rhythm essence.*

Done.We have added text and a new equation (as Equation 1) explaining the calculation (subsection “Emergence of theta oscillations from a biological data-driven, full-scale model of the CA1 network”, second paragraph). Equation 1 shows the calculation for the case of 0.65Hz input as a template, but more generally it also illustrates that the number of afferent spikes/ cycle for other stimulation frequencies changes not just with the simulation frequency but also with the resulting network oscillation frequency (for example, at 0.5Hz input, where the network no longer displays robust theta (Figure 6), if the network had oscillated at the control theta frequency of 7.8 Hz, it would have received 29,100 spikes/cycle; if it had been able to sustain a slower theta oscillation, it could have received up to 45,500 spikes/cycle for a 5 Hz oscillation; but it was not able to react to the stimulation in this way, instead displaying a strong 19.5 Hz oscillation (Figure 6—figure supplement 1) that corresponds to 11,700 spikes/cycle).

*3) Model details*

*i) The description of intrinsic properties of individual neurons (passive and active parameters of the conductance-based models) is entirely missing from the paper, and inadequate on the model website. These model details need to be accessible forever, directly from the paper (or as supplementary information), not from an external website that could be discontinued. Yet even on the website, the information about individual neurons is incomplete/inconvenient to access. For example, a row of graphs about each channel in the model is included on the website (http://mariannebezaire.com/ca1_graphic/mymodel.html), but the axes of the graphs are not labeled. Do these show voltage activation curves? Distributions throughout the somato-dendritic axis? Are all ion channels included in each neuron type? Also, I found out by accident (website had no instructions) that manually clicking on each morphology sends you to a page with a set of tables and images with drop-down menus. It would have been helpful to provide text/tables summarizing channels included in each neuron type, with graphs of model vs. experimental traces and references to the exact equations for channel. It is essential for such information to be accessible from supplemental information, not an external site, unless it is a well-monitored, maintained, and curated site such as model DB.*

Done. We have included the information as a new supplementary material to the manuscript. This supplementary material, therefore, contains a detailed characterization of each cell type in the model accessible from this paper. In addition, the supplementary material contains a table for properties of the ion channels used within the model. Data from the experimental cells featured in the supplementary material has been submitted and is available in the Open Science Framework repository.

We have also worked with the developers of Open Source Brain to convert our model into NeuroML and characterize it thoroughly. Open Source Brain is similar to ModelDB in that it is a curated, monitored, and well-maintained website that contains a large number of models, and it also allows for users to navigate through models, applying various standard and custom protocols to the network and its components. See the in-progress entry at: http://opensourcebrain.org/projects/nc_ca1

Further, we have added the necessary code to support a “Model View” tab of our code entry in ModelDB, so that some characteristics of the model can be viewed directly from NEURON within the ModelDB site (the curator informed us that this feature would be activated upon making our model entry public, which would occur upon acceptance of our manuscript by a peer-reviewed journal): https://senselab.med.yale.edu/ModelDB/showModel.cshtml?model=187604

*ii) Such detailed models suffer from too much detail, which translates mathematically into thousands of parameters that need to be tweaked to simulate accurately and quantitatively the responses of cells in the networks. Did the authors use any optimization technique to constrain their model's responses other than calculated guises of biological detail?*

We have added more detail about the model tuning in the methods. Mainly, the cell intrinsic properties and incoming synaptic amplitudes and kinetics were optimized and hand-tuned to experimental data where available, and the goodness of fit is reported in the new supplementary material. As explained in the methods, we used a combination of optimization and hand-tuning at the level of individual cell type electrophysiological profiles. We employed an optimization tool built into NEURON (the Multiple Run Fitter, or MRF) to tune the ion channel conductances and other electrophysiological properties to produce each cell type model. Connections between cells were hand-tuned to fit the rise and decay times, as well as the synaptic amplitudes.

*The network models contained 1 excitatory principal cell type and 8 types inhibitory interneurons. These interneurons targeted specific compartments of the principal cells. But what about other types of excitatory cells? Also, what about inhibitory cells that targeted CA1 inhibitory cells (e.g. Chamberland, S., and Topolnik, L. (2012))?*

We agree with the reviewer that it will be interesting to include these cell types in the future. However, as explained in the manuscript, we refrained from including cell types for which insufficient biological data were available, as is the case for the inhibitory cell specific interneurons and the different types of pyramidal cells (see also the list of “explicit, forced assumptions” in Table 2 of Bezaire & Soltesz, 2013).